# Precise Asymptotic Generalization for Multiclass Classification with Overparameterized Linear Models

**David X. Wu**
Department of EECS
UC Berkeley
Berkeley, CA 94720
david_wu@berkeley.edu

**Anant Sahai**
Department of EECS
UC Berkeley
Berkeley, CA 94720
sahai@eecs.berkeley.edu

## Abstract

We study the asymptotic generalization of an overparameterized linear model for multiclass classification under the Gaussian covariates bi-level model introduced in Subramanian et al. (2022), where the number of data points, features, and classes all grow together. We fully resolve the conjecture posed in Subramanian et al. (2022), matching the predicted regimes for generalization. Furthermore, our new lower bounds are akin to an information-theoretic strong converse: they establish that the misclassification rate goes to 0 or 1 asymptotically. One surprising consequence of our tight results is that the min-norm interpolating classifier can be asymptotically suboptimal relative to noninterpolating classifiers in the regime where the min-norm interpolating regressor is known to be optimal.

The key to our tight analysis is a new variant of the Hanson-Wright inequality which is broadly useful for multiclass problems with sparse labels. As an application, we show that the same type of analysis can be used to analyze the related multilabel classification problem under the same bi-level ensemble.

## 1  Introduction

In this paper, we directly follow up on a specific line of work initiated by Subramanian et al. (2022); Wu and Sahai (2023). For the sake of self-containedness, we briefly reiterate the context, directing the reader to Subramanian et al. (2022) and the references cited therein for more. A broader story can be found in Bartlett et al. (2021); Belkin (2021); Dar et al. (2021); Oneto et al. (2023).

Classical statistical learning theory intuition predicts that highly expressive models, which can interpolate random labels (Zhang et al., 2016; 2021), ought not to generalize well. However, deep learning practice has seen such models performing well when trained with good labels. Resolving this apparent contradiction has recently been the focus of a multitude of works, and this paper builds on one particular thread of investigation that can be rooted in Bartlett et al. (2020); Muthukumar et al. (2020) where the concept of benign/harmless interpolation was crystallized in the context of overparameterized linear regression problems and conditions given for when this can happen. In Muthukumar et al. (2021), a specific toy "bi-level model" with Gaussian features was introduced to study overparameterized binary classification and show that successful generalization could happen even beyond the conditions for benign interpolation for regression. Following the introduction of the corresponding multi-class problem in Wang et al. (2021) with a constant number of classes, an asymptotic setting where the number of classes can grow with the number of training examples was introduced in Subramanian et al. (2022) where a conjecture was presented for when minimum-norm interpolating classifiers will generalize. We are now in a position to state our main contributions; afterwards, we expand on the related works.

37th Conference on Neural Information Processing Systems (NeurIPS 2023).

**Our contributions**

Our main contribution is crisply identifying the asymptotic regimes where an overparameterized linear model which performs minimum-norm interpolation does and does not generalize for multiclass classification under a Gaussian features assumption, thus resolving the main conjecture posed by (Subramanian et al., 2022). We improve on the analysis of Subramanian et al. (2022); Wu and Sahai (2023), covering all regimes with the asymptotically optimal misclassification rate. When the model generalizes, it does so with a misclassification rate $o(1)$, and we show a matching "strong converse" establishing when it misclassifies, it does so with rate $1 - o(1)$, where the explicit rate is nearly identical to that of random guessing. The critical component of our analysis is a new variant of the Hanson-Wright inequality, which applies to bilinear forms between a vector with subgaussian entries and a vector that is bounded and has *soft sparsity*, a notion we will define in Section 4.2. We show how this tool can be used to analyze other multiclass problems, such as multilabel classification.

## 1.1 Brief treatment of related work

Our thread begins with a recent line of work that analyzes the generalization behavior of overparameterized linear models for regression (Hastie et al., 2022; Mei and Montanari, 2022; Bartlett et al., 2020; Belkin et al., 2020; Muthukumar et al., 2020). These simple models demonstrate how the capacity to interpolate noise can actually aid in generalization: training noise can be harmlessly absorbed by the overparameterized model without contaminating predictions on test points. In effect, extra features can be regularizing (in the context of descent algorithms' implicit regularization (Soudry et al., 2018; Ji and Telgarsky, 2019; Engl et al., 1996; Gunasekar et al., 2018)), but an excessive amount of such regularization causes regression to fail because even the true signal will not survive the training process. Although works in this thread focus on very shallow networks, Chatterji and Long (2023) established that deeper networks can behave similarly. Note that recently, Mallinar et al. (2022) called-out an alternative regime (behaving like 1-nearest-neighbor learning) called "tempered" overfitting in which training noise is not completely absorbed but the true signal does survive training.

The thread continues in a line of work that studies binary classification (Muthukumar et al., 2021; Chatterji and Long, 2021; Wang and Thrampoulidis, 2021) in similar overparameterized linear models. While confirming that the basic story is similar to regression, these works identify a further surprise: binary classification can work in some regimes where the corresponding regression problem would not work[1] due to the regularizing effect of overparameterization being too strong. Just as in the regression case, the results here are sharp in toy models: we can exactly characterize where binary classification using an interpolating classifier asymptotically generalizes.

With binary classification better understood, the thread continues to multiclass classification. After all, the current wave of deep learning enthusiasm originated in breakthrough performance in multiclass classification, and we have seen a decade of ever larger networks trained on ever larger datasets with ever more classes Kaplan et al. (2020). Using similar toy models (Muthukumar et al., 2020; 2021; Wang et al., 2022), the constant number of classes case was studied in Wang et al. (2021) to recover results similar to binary classification and subsequently generalized to general convex losses with regularization in Loureiro et al. (2021) and student-teacher networks in Cornacchia et al. (2023).

Subramanian et al. (2022) further introduced a model where the number of classes grows with the number of training points and proved an achievability result on how fast the number of classes can grow while still allowing the interpolating classifier to asymptotically generalize. While Subramanian et al. (2022) gave a conjecture for what the full region should be, there was no converse proof, and they could not show generalization in entire conjectured region. Wu and Sahai (2023) proved a partial weak converse; they showed that the misclassification rate is bounded away from $0$ — rather than tending to $1$ — in some of the predicted regimes.

The model, formally defined in the following section, is a stylized version of the well-known spiked covariance model (Johnstone, 2001; Donoho et al., 2018). On the theoretical front, it is related to several problems such as PCA variants (Montanari and Richard, 2015; Richard and Montanari, 2014)

---

[1]Regression failing in the overparameterized regime is linked to the empirical covariance of the limited data not revealing the spiked reality of the underlying covariance (Wang and Fan, 2017). See Appendix J of Subramanian et al. (2022). When regression doesn't generalize, we also get "support-vector proliferation" in classification problems (Muthukumar et al., 2021; Hsu et al., 2021) which is also intimately related to the phenomenon of "neural collapse" (Papyan et al., 2020) as discussed, for example, in Xu et al. (2023).

and community detection in the stochastic block model (Abbe, 2017). These models have also been applied in practice for climate studies and functional data analysis (Johnstone, 2001). At a high level, spiked covariance models can be interpreted as a linearized version of the manifold hypothesis.

## 2 Problem setup

The following exposition is lifted from Subramanian et al. (2022), which we include for the sake of staying consistent and self-contained. In Appendix K, we include an alternative high level framing of the problem which readers may find helpful for intuition.

We consider the multiclass classification problem with $k$ classes. The training data consists of $n$ pairs $\{\boldsymbol{x}_i, \ell_i\}_{i=1}^n$ where $\boldsymbol{x}_i \in \mathbb{R}^d$ are i.i.d standard Gaussian vectors[2]. We assume that the labels $\ell_i \in [k]$ are generated as follows.

**Assumption 1** (1-sparse noiseless model). *The class labels $\ell_i$ are generated based on which of the first $k$ dimensions of a point $\boldsymbol{x}_i$ has the largest value,*

$$\ell_i = \arg\max_{m \in [k]} \boldsymbol{x}_i[m]. \tag{1}$$

Let us emphasize at this point that the classifier that we analyze only observes the training data, and does not use any of the data-generating assumptions.

For a vector $\boldsymbol{x}$, we index its $j$th entry with $\boldsymbol{x}[j]$. Hence, under Assumption 1, $\boldsymbol{x}_i[m]$ can be interpreted as how representative of class $m$ the $i$th training point is.

For clarity of exposition in the analysis, we make explicit a feature weighting that transforms the training points:

$$\boldsymbol{x}_i^w[j] = \sqrt{\lambda_j} \boldsymbol{x}_i[j] \quad \forall j \in [d]. \tag{2}$$

Here $\boldsymbol{\lambda} \in \mathbb{R}^d$ contains the squared feature weights. The feature weighting serves the role of favoring the true pattern, something that is essential for good generalization. Again, we emphasize that the classifier does not do any reweighting of features; this explicit step is purely syntactic. [3]

The weighted feature matrix $\boldsymbol{X}^w \in \mathbb{R}^{n \times d}$ is given by

$$\boldsymbol{X}^w = [\boldsymbol{x}_1^w \quad \cdots \quad \boldsymbol{x}_n^w]^\top = [\sqrt{\lambda_1} \boldsymbol{z}_1 \quad \cdots \quad \sqrt{\lambda_d} \boldsymbol{z}_d] \tag{3}$$

where we introduce the notation $\boldsymbol{z}_j \in \mathbb{R}^n$ to contain the $j^{th}$ feature from the $n$ training points. Note that $\boldsymbol{z}_j \sim N(0, \boldsymbol{I}_n)$ are i.i.d Gaussians. We use a one-hot encoding for representing the labels as the matrix $\boldsymbol{Y}^{\mathsf{oh}} \in \mathbb{R}^{n \times k}$

$$\boldsymbol{Y}^{\mathsf{oh}} = [\boldsymbol{y}_1^{\mathsf{oh}} \quad \cdots \quad \boldsymbol{y}_k^{\mathsf{oh}}], \qquad \text{where} \qquad \boldsymbol{y}_m^{\mathsf{oh}}[i] = \begin{cases} 1, & \text{if } \ell_i = m \\ 0, & \text{otherwise} \end{cases}. \tag{4}$$

Since we consider linear models, we center the one-hot encodings and define

$$\boldsymbol{y}_m \triangleq \boldsymbol{y}_m^{\mathsf{oh}} - \frac{1}{k}\boldsymbol{1}. \tag{5}$$

---

[2]Following previous work, we are staying within a Gaussian features framework. However, recent developments have confirmed that these models are actually predictive when the features arise from nonlinearities in a lifting, as long as there is enough randomness underneath (Hu and Lu, 2022; Lu and Yau, 2022; Goldt et al., 2022; Misiakiewicz, 2022; McRae et al., 2022; Pesce et al., 2023; Kaushik et al., 2023).

[3]Our weighted feature model is equivalent to other works (e.g. Muthukumar et al. (2021)) that assume that the covariates come from a $d-$dimensional anisotropic Gaussian with a covariance matrix $\Sigma$ that favors the truly important directions (Wei et al., 2022). These directions do not have to be axis-aligned — we make that assumption only for notational convenience. In reality, the optimizer will never know these directions *a priori*.

Our classifier consists of $k$ coefficient vectors $\widehat{\boldsymbol{f}}_m$ for $m \in [k]$ that are learned by minimum-norm interpolation (MNI) of the zero-mean one-hot variants using the weighted features:[4]

$$\widehat{\boldsymbol{f}}_m = \arg \min_{\boldsymbol{f}} \|\boldsymbol{f}\|_2 \tag{6}$$

$$\text{s.t. } \boldsymbol{X}^w \boldsymbol{f} = \boldsymbol{y}_m. \tag{7}$$

We can express these coefficients in closed form as

$$\widehat{\boldsymbol{f}}_m = (\boldsymbol{X}^w)^\top \left(\boldsymbol{X}^w (\boldsymbol{X}^w)^\top\right)^{-1} \boldsymbol{y}_m. \tag{8}$$

On a test point $\boldsymbol{x}_{\text{test}} \sim N(0, \boldsymbol{I}_d)$ we predict a label as follows: First, we transform the test point into the weighted feature space to obtain $\boldsymbol{x}_{\text{test}}^w$ where $\boldsymbol{x}_{\text{test}}^w[j] = \sqrt{\lambda_j} \boldsymbol{x}_{\text{test}}[j]$ for $j \in [d]$. Then we compute $k$ scalar "scores" and assign the class based on the largest score as follows:

$$\hat{\ell} = \arg \max_{1 \le m \le k} \widehat{\boldsymbol{f}}_m^\top \boldsymbol{x}_{\text{test}}^w. \tag{9}$$

By assumption, a misclassification event $\mathcal{E}_{\text{err}}$ occurs whenever

$$\arg \max_{1 \le m \le k} \boldsymbol{x}_{\text{test}}[m] \ne \arg \max_{1 \le m \le k} \widehat{\boldsymbol{f}}_m^\top \boldsymbol{x}_{\text{test}}^w. \tag{10}$$

We study where the MNI generalizes in an asymptotic regime where the number of training points, features, classes, and feature weights all scale according to the bi-level ensemble model[5]:

**Definition 1** (Bi-level ensemble). *The bi-level ensemble is parameterized by $p, q, r$ and $t$ where $p > 1$, $0 \le r < 1$, $0 < q < (p - r)$ and $0 \le t < r$. Here, parameter $p$ controls the extent of overparameterization, $r$ determines the number of favored features, $q$ controls the weights on favored features and $t$ controls the number of classes. The number of features ($d$), number of favored features ($s$), and number of classes ($k$) all scale with the number of training points ($n$) as follows:*

$$d = \lfloor n^p \rfloor, s = \lfloor n^r \rfloor, a = n^{-q}, k = c_k \lfloor n^t \rfloor, \tag{11}$$

*where $c_k$ is a positive integer. Define the feature weights by*

$$\sqrt{\lambda_j} = \begin{cases} \sqrt{\frac{ad}{s}}, & 1 \le j \le s \\ \sqrt{\frac{(1-a)d}{d-s}}, & \text{otherwise} \end{cases}. \tag{12}$$

*We introduce the notation $\lambda_F \triangleq \frac{ad}{s}$ and $\lambda_U \triangleq \frac{(1-a)d}{d-s}$ to distinguish between the (squared) favored and unfavored weights, respectively.*

We visualize the bi-level model in Fig. 1, reproduced from Subramanian et al. (2022). Intuitively, the bi-level ensemble captures a simple family of overparameterized problems where learning can succeed. Although there are $d = n^p$ features where $d \gg n$, there is a low dimensional subspace of favored, higher weight features of dimension $s = n^r$, and $s \ll n$. From this perspective, the bi-level model can be viewed as a parameterized version of an approximate linear manifold hypothesis. Depending on the signal strength, the noise added from the $d - s$ unfavored features can either help generalization ("benign overfitting") or overwhelm the true signal and cause the classifier to fail.

## 3   Main results

In this section we state our main results and compare them to what was known and conjectured previously. Subramanian et al. (2022) use heuristic calculations to conjecture necessary and sufficient conditions for the bi-level model to generalize; we restate the conjecture here for reference.

---

[4]The classifier learned via this method is equivalent to those obtained by other natural training methods (SVMs or gradient-descent with exponential tailed losses like cross-entropy) under sufficient overparameterization (Wang et al., 2021; Kaushik et al., 2023). Recently, Lai and Muthukumar (2023) showed via an extension of Ji and Telgarsky (2021) that a much broader category of losses also asymptotically result in convergence to the same MNI solution for sufficiently overparameterized classification problems.

[5]Such models are widely used to study learning even beyond this particular thread of work. For example, Tan et al. (2023) uses this to understand the privacy/generalization tradeoff of overparameterized learning.

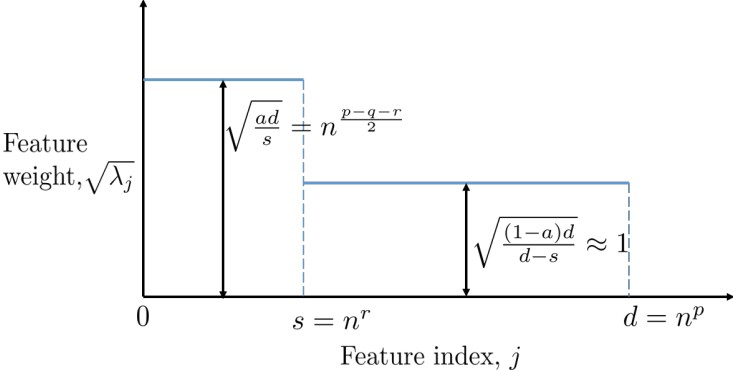

Figure 1: Bi-level feature weighting model. The first $s$ features have a higher weight and are favored during minimum-norm interpolation. These can be thought of as the square-roots of the eigenvalues of the feature covariance matrix $\Sigma$ in a Gaussian model for the covariates as in Bartlett et al. (2020).

**Conjecture 3.1** (Conjectured bi-level regions)**.** *Under the bi-level ensemble model (Definition 1), when the true data generating process is 1-sparse (Assumption 1), as $n \to \infty$, the probability of misclassification $\Pr[\mathcal{E}_{\text{err}}]$ for MNI as described in Eq. (6) satisfies*

$$\Pr[\mathcal{E}_{\text{err}}] \to \begin{cases} 0, & \text{if } t < \min\{1 - r, p + 1 - 2\max\{1, q + r\}\} \\ 1, & \text{if } t > \min\{1 - r, p + 1 - 2\max\{1, q + r\}\} \end{cases}. \tag{13}$$

Our main theorem establishes that Conjecture 3.1 indeed captures the correct generalization behavior of the overparameterized linear model.

**Theorem 3.2** (Generalization for bi-level-model)**.** *Under the bi-level ensemble model (Definition 1), when the true data generating process is 1-sparse (Assumption 1), Conjecture 3.1 holds.*

For comparison, we quote the best known previous positive and negative results for the bi-level model, which only hold in the restricted regime where regression fails ($q + r > 1$).

**Theorem 3.3** (Generalization for bi-level model (Subramanian et al., 2022))**.** *In the same setting as Conjecture 3.1, in the regime where regression fails ($q + r > 1$), as $n \to \infty$ we have $\Pr[\mathcal{E}_{\text{err}}] \to 0$ if*

$$t < \min\{1 - r, p + 1 - 2(q + r), p - 2, 2q + r - 2\}. \tag{14}$$

**Theorem 3.4** (Misclassification in bi-level model (Wu and Sahai, 2023))**.** *In the same setting as Conjecture 3.1, in the regime where regression fails ($q + r > 1$), as $n \to \infty$ we have $\Pr[\mathcal{E}_{\text{err}}] \geq \frac{1}{2}$ if*

$$t > \min\{1 - r, p + 1 - 2(q + r)\}. \tag{15}$$

Let us interpret the different conditions in Conjecture 3.1. To interpret the condition $t < 1 - r$, first rearrange it to $t + r < 1$. Recall that the parameter $r$ controls the number of favored features, and hence is a proxy for the "effective dimension" of the problem. On the other hand, the parameter $t$ controls the number of classes, so in a loose sense there are $n^{t+r}$ parameters being learned. From this perspective, the condition $t + r < 1$ says that the problem is "effectively underparameterized".

The other condition on $p + 1 - 2\max\{1, q + r\}$ comes from looking at the noise from the unfavored features. To see why, recall that the squared favored feature weighting is $\lambda_F = n^{p-q-r}$. So for fixed $p$, the quantity $q + r$ controls the level of favored feature weighting. When $q + r > 1$, the favored feature weighting is small enough that regression fails, and the empirical covariance becomes flat. In this case, the condition becomes $t < p + 1 - 2(q + r) = (p - q - r) + 1 - (q + r)$. As $q + r$ increases, the amount of favoring decreases, making it harder to generalize.

For ease of comparison between our main result and Theorems 3.3 and 3.4, we visualize the regimes in Fig. 2, as in Subramanian et al. (2022); Wu and Sahai (2023). In particular, the blue starred and dashed regions in Fig. 2 indicate how Theorem 3.3 only applies where regression fails. In contrast, our new result holds regardless of whether regression fails or not, as in the the green diamond region and light blue triangle regions. The regions are also completely tight; the looseness between the prior Theorem 3.3 and our result can be seen in the light blue square region.

The weak converse in the prior Theorem 3.4 captures some of the correct conditions for misclassification, but again only when $q + r > 1$. As depicted in the maroon X region for $r < 0.25$ in Fig. 2b, our main theorem gives a strong converse, whereas Theorem 3.4 has nothing to say because $q + r < 1$. Theorem 3.4 also only proves that the misclassification rate is asymptotically at least $\frac{1}{2}$. In the red circle and maroon X regions, we illustrate how our result pushes the misclassification rate to $1 - o(1)$, which requires a more refined analysis. We elaborate on this further in Section 4.

We remark that it is simpler to analyze the case where regression fails, as the random matrices that arise in the analysis are *flat*, i.e. approximately equal to a scaled identity matrix. However, in the regime where regression works, the same matrices have a *spiked* spectrum, which complicates the analysis. To smoothly handle both cases, we leverage a new variant of the Hanson-Wright inequality to show concentration of certain sparse bilinear forms; see Section 4.1 for more details.

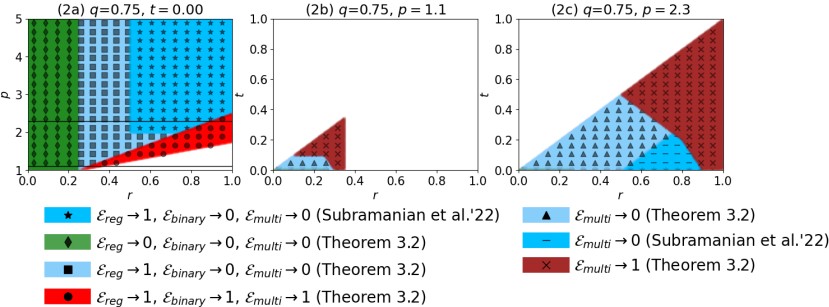

Figure 2: Example of regimes for multiclass/binary classification and regression. The white regions correspond to invalid regimes under the bi-level model. The entirety of 2b and all the light blue regions are new to this paper, as is showing that the error tends to 1 in the maroon regions.

## 4 Technical overview

We now sketch out the proof for our main theorem. As in Subramanian et al. (2022); Wu and Sahai (2023), the starting point is writing out the necessary and sufficient conditions for misclassification.

Assume without loss of generality that the test point $x_{\text{test}} \sim N(0, I_d)$ has true label $\alpha$ for some $\alpha \in [k]$. Let $x_{\text{test}}^w$ be the weighted version of this test point. From (10), an equivalent condition for misclassification is that for some $\beta \neq \alpha, \beta \in [k]$, we have $\widehat{f}_\alpha^\top x_{\text{test}}^w < \widehat{f}_\beta^\top x_{\text{test}}^w$, i.e. the score for $\beta$ outcompetes the score for $\alpha$. Define the Gram matrix $A \triangleq X^w (X^w)^\top$, the relative label vector $\Delta y \triangleq y_\alpha - y_\beta \in \{-1, 0, 1\}^n$, and the relative survival vector $\widehat{h}_{\alpha,\beta} \in \mathbb{R}^d$ which compares the signal from $\alpha$ and $\beta$:

$$\widehat{h}_{\alpha,\beta}[j] \triangleq \lambda_j^{-1/2}(\widehat{f}_\alpha[j] - \widehat{f}_\beta[j]) \tag{16}$$

$$= z_j^\top A^{-1} \Delta y, \tag{17}$$

where to obtain the last line we have used (8). By converting the misclassification condition into the unweighted feature space we see that we will have errors when

$$\lambda_\alpha \widehat{h}_{\alpha,\beta}[\alpha] x_{\text{test}}[\alpha] - \lambda_\beta \widehat{h}_{\beta,\alpha}[\beta] x_{\text{test}}[\beta] < \sum_{j \notin \{\alpha,\beta\}} \lambda_j \widehat{h}_{\beta,\alpha}[j] x_{\text{test}}[j]. \tag{18}$$

Define the contamination term $\mathsf{CN}_{\alpha,\beta}$:

$$\mathsf{CN}_{\alpha,\beta} \triangleq \sqrt{\sum_{j \notin \{\alpha,\beta\}} \lambda_j^2 (\widehat{h}_{\beta,\alpha}[j])^2}. \tag{19}$$

Note that $\mathsf{CN}_{\alpha,\beta}$ normalizes the RHS of (18) into a standard Gaussian. Indeed, define

$$Z^{(\beta)} \triangleq \frac{1}{\mathsf{CN}_{\alpha,\beta}} \sum_{j \notin \{\alpha,\beta\}} \lambda_j \widehat{h}_{\beta,\alpha}[j] x_{\text{test}}[j] \sim N(0,1). \tag{20}$$

Since $\alpha, \beta \in [k]$ are favored, we have $\lambda_\alpha = \lambda_\beta = \lambda_F$. Hence, an equivalent condition for misclassification is that there exists some $\beta \neq \alpha$, $\beta \in [k]$ such that

$$\frac{\lambda_F}{\mathsf{CN}_{\alpha,\beta}}(\widehat{\boldsymbol{h}}_{\alpha,\beta}[\alpha]\boldsymbol{x}_{\mathsf{test}}[\alpha] - \widehat{\boldsymbol{h}}_{\beta,\alpha}[\beta]\boldsymbol{x}_{\mathsf{test}}[\beta]) < Z^{(\beta)}. \tag{21}$$

We now translate the above criterion into *sufficient* conditions for correct classification and misclassification and analyze these two cases separately.

**Correct classification:** For correct classification, it suffices for the minimum value of the LHS of Eq. (21) to outcompete the maximum value of the RHS, where the max is taken over $\beta \in [k], \beta \neq \alpha$. Some algebra, as in Subramanian et al. (2022), shows that we correctly classify if

$$\underbrace{\frac{\min_\beta \lambda_F \widehat{\boldsymbol{h}}_{\alpha,\beta}[\alpha]}{\max_\beta \mathsf{CN}_{\alpha,\beta}}}_{\mathsf{SU}/\mathsf{CN}\ \text{ratio}} \left( \underbrace{\min_\beta (\boldsymbol{x}_{\mathsf{test}}[\alpha] - \boldsymbol{x}_{\mathsf{test}}[\beta])}_{\text{closest feature margin}} - \underbrace{\max_\beta |\boldsymbol{x}_{\mathsf{test}}[\beta]|}_{\text{largest competing feature}} \cdot \underbrace{\max_\beta \left| \frac{\widehat{\boldsymbol{h}}_{\alpha,\beta}[\alpha] - \widehat{\boldsymbol{h}}_{\beta,\alpha}[\beta]}{\widehat{\boldsymbol{h}}_{\alpha,\beta}[\alpha]} \right|}_{\text{survival variation}} \right)$$
$$> \underbrace{\max_\beta Z^{(\beta)}}_{\text{normalized contamination}}. \tag{22}$$

We will show that under the conditions specified in Conjecture 3.1, with high probability, the relevant survival to contamination ratio $\mathsf{SU}/\mathsf{CN}$ grows at a polynomial rate $n^v$ for some $v > 0$, whereas the term in the parentheses shrinks at a subpolynomial rate $\omega(n^{-\delta})$ for any $\delta > 0$. Further, by standard subgaussian maximal inequalities, the magnitudes of the *normalized contamination* is no more than $O(\sqrt{\log(nk)})$ with high probability. Thus, with high probability the LHS outcompetes the RHS, leading to correct classification. See Section 4.1 for more discussion on how we prove tight bounds on the survival-to-contamination ratios.

**Misclassification:** On the other hand, for misclassification it suffices for the maximum *absolute* value of the LHS of Eq. (21) to be outcompeted by the maximum value of the RHS. Some manipulations yield the following sufficient condition for misclassification:

$$\underbrace{\frac{\max_\beta \lambda_F \left( \left| \widehat{\boldsymbol{h}}_{\alpha,\beta}[\alpha] \right| + \left| \widehat{\boldsymbol{h}}_{\beta,\alpha}[\beta] \right| \right)}{\min_\beta \mathsf{CN}_{\alpha,\beta}}}_{\mathsf{SU}/\mathsf{CN}\ \text{ratio}} \cdot \underbrace{\max_{\gamma \in [k]} |\boldsymbol{x}_{\mathsf{test}}[\gamma]|}_{\text{largest label-defining feature}} < \underbrace{\max_\beta Z^{(\beta)}}_{\text{normalized contamination}}. \tag{23}$$

We show that within the misclassification regimes in Conjecture 3.1, the survival-to-contamination ratio $\mathsf{SU}/\mathsf{CN}$ *shrinks* at a polynomial rate $n^{-w}$ for some $w > 0$. By standard subgaussian maximal inequalities, the largest label-defining feature is $O(\sqrt{\log(nk)})$ with high probability. Gaussian anticoncentration implies that for some $\beta \neq \alpha, \beta \in [k]$, $Z^{(\beta)}$ outcompetes the LHS with probability at least $\frac{1}{2} - o(1)$. Hence, we conclude that the model will misclassify with rate at least $\frac{1}{2}$ asymptotically.

Let us now describe how to boost the misclassification rate to $1 - o(1)$. Notice that the above argument only considered the competition between the LHS of Eq. (23) and one of the $Z^{(\beta)}$'s on the RHS instead of the maximum $Z^{(\beta)}$. It's not hard to see from the definition of $Z^{(\beta)}$ in Eq. (20) that the $Z^{(\beta)}$ are jointly Gaussian. For intuition's sake, assuming the $Z^{(\beta)}$ were *independent*, then $\max_\beta Z^{(\beta)}$ would outcompete with probability $(\frac{1}{2} - o(1))^{k-1}$.

In reality, the $Z^{(\beta)}$ are correlated, but we are able to show that the maximum correlation between the $Z^{(\beta)}$ is $\frac{1}{2} + o(1)$ with high probability. An application of Slepian's lemma (Slepian (1962)) and some explicit bounds on orthant probabilities (Pinasco et al. (2021)) implies that $\max_\beta Z^{(\beta)} > 0$ with probability at least $1 - \frac{1}{k^{1+o(1)}}$. Another application of anticoncentration implies that $\max_\beta Z^{(\beta)} > n^{-w}$ with probability $1 - o(1)$, which finishes off the proof.

## 4.1 Bounding the survival-to-contamination ratio

Note that the critical *survival-to-contamination* ratio appears in both Eqs. (22) and (23). The most involved part of the proof is nailing down the correct order of growth of the survival to contamination ratio; a similar analysis tightly bounds the survival variation and the correlation structure of the $Z^{(\beta)}$.

To understand the relative survival and contamination, we must analyze the bilinear forms $\widehat{h}_{\alpha,\beta}[j] = z_j^\top A^{-1} \Delta y$. Similarly, to control the correlation of the $Z^{(\beta)}$, we must understand the correlation between the $\widehat{h}_{\alpha,\beta}$ vectors, which reduces to understanding the bilinear forms $z_j^\top A^{-1} y_\alpha$ for $j \in [d], \alpha \in [k]$. The main source of inspiration for bounding these bilinear forms is the heuristic style of calculation carried out in Appendix K of Subramanian et al. (2022) that leads to Conjecture 3.1.

To simplify the discussion, we temporarily restrict to the regime where regression fails ($q + r > 1$). However, our main technical tool seamlessly generalizes to the regime where regression works ($q + r < 1$). In the regime where regression fails, $A^{-1}$ turns out to have a *flat* spectrum: $A^{-1} \approx \alpha I$ for some constant $\alpha > 0$. Assume for now that $A^{-1}$ is *exactly* equal to a scaled identity matrix. Then the survival is proportional to $z_\alpha^\top \Delta y$, which is a random inner product. Similarly, to bound the contamination terms we must control the random inner product $z_j^\top \Delta y$ for $j \notin \{\alpha, \beta\}$.

Since $\Delta y$ is a sparse vector — it only has $\frac{2n}{k}$ nonzero entries in expectation — a quick computation reveals that $\mathbb{E}[z_\alpha^\top \Delta y] = \tilde{O}(\frac{n}{k})$ and $\mathbb{E}[z_j^\top \Delta y] = 0$. The deciding factor, then, is how tightly these quantities concentrate around their means. A naïve application of Hoeffding implies a concentration radius of order $\tilde{O}(\sqrt{n})$, which would lead to looseness in the overall result. The hope is to exploit sparsity to get a concentration radius of order $\tilde{O}(\sqrt{n/k})$. This is where our new technical tool Theorem 4.1 comes in, which may be of independent interest; we present it in the following section.

## 4.2 A new variant of the Hanson-Wright inequality

In reality, even in the regime where regression fails, $A^{-1}$ is not actually perfectly flat. Even worse, in the regime where regression works, $A^{-1}$ is actually spiked. Thus, we cannot simply reduce the bilinear form $z_j^\top A^{-1} \Delta y$ to an inner product. Instead, we turn to the well-known Hanson-Wright inequality (Rudelson and Vershynin, 2013), which tells us that quadratic forms of random vectors with independent, mean zero, subgaussian entries concentrate around their mean. It was used extensively to study binary classification (Muthukumar et al., 2021), and multiclass classification (Subramanian et al., 2022; Wu and Sahai, 2023).

However, just as Hoeffding is loose, so too is the standard form of Hanson-Wright, because it also does not exploit sparsity. This motivates a new variant of Hanson-Wright which fully leverages the (soft) sparsity inherent to multiclass problems with an increasing number of classes. We now formally define the notions of soft and hard sparsity.

**Definition 2** (Soft and hard sparsity). *For $\pi \leq 1$, we say that random vector $y = (Y_i)_{i=1}^n$ has soft sparsity at level $\pi$ if $|Y_i| \leq 1$ almost surely and $\mathrm{Var}(Y_i) \leq \pi$ for all $i$. On the other hand, we say that $y$ has hard sparsity at level $\pi$ if at most a $\pi$ fraction of the $Y_i$ are nonzero.*

In particular, our variant Theorem 4.1 below requires that one of the vectors in the bilinear form has *soft sparsity* at level $\pi$. Throughout, one should think of $\pi = o(1)$, and for us indeed $\pi = O(\frac{1}{k})$. One can check that a bounded random vector $y$ with *hard sparsity* level $\pi$ must also have soft sparsity at level $O(\pi)$, so soft sparsity is more general for bounded random vectors. In Table 1 we compare our variant with several variants of Hanson-Wright which have appeared in the literature, some of which involve hard sparsity.

Define the subgaussian norm $\|\xi\|_{\psi_2}$ (Vershynin, 2018) as

$$\|\xi\|_{\psi_2} = \inf_{K > 0} \left\{ K : \mathbb{E} \exp\left(\xi^2 / K^2\right) \leq 2 \right\}, \tag{24}$$

**Theorem 4.1** (Hanson-Wright for bilinear forms with soft sparsity). *Let $x = (X_1, \ldots, X_n) \in \mathbb{R}^n$ and $y \in (Y_1, \ldots, Y_n) \in \mathbb{R}^n$ be random vectors such that $(X_i, Y_i)$ are independent pairs of (possibly correlated) centered random variables such that $\|X_i\|_{\psi_2} \leq K$ and $Y_i$ has soft sparsity at level $\pi$, i.e. $|Y_i| \leq 1$ almost surely, and $\mathbb{E}[Y_i^2] \leq \pi$. Assume that conditioned on $Y_j$, $\|X_j\|_{\psi_2} \leq K$. Then there exists an absolute constant $c > 0$ such that for all $M \in \mathbb{R}^{n \times n}$ and $\epsilon \geq 0$ we have*

$$\Pr\left[ |x^\top M y - \mathbb{E}[x^\top M y]| > \epsilon \right] \leq 2 \exp\left( -c \min\left\{ \frac{\epsilon^2}{K^2 \pi \|M\|_F^2}, \frac{\epsilon}{K \|M\|_2} \right\} \right). \tag{25}$$

The full proof of Theorem 4.1 is deferred to Appendix G. The main proof techniques are heavily inspired by those of Rudelson and Vershynin (2013); Zhou (2019); Park et al. (2022). However, the

| Variant | Assumptions on $\boldsymbol{y}$ | Concentration radius |
|---|---|---|
| Classic quadratic[a]: $\boldsymbol{x}^\top \boldsymbol{M} \boldsymbol{x}$ | same as $\boldsymbol{x}$ | $\tilde{O}(\|\boldsymbol{M}\|_F)$ |
| Sparse bilinear[b]: $\boldsymbol{x}^\top \boldsymbol{M}(\boldsymbol{\gamma} \circ \boldsymbol{y})$ | $\gamma_i \sim \mathsf{Ber}(\pi)$, indep. of $X_i$ but not $Y_i$ | $\tilde{O}(\sqrt{\pi}\|\boldsymbol{M}\|_F)$ |
| Sparse bilinear[c]: $\boldsymbol{x}^\top \boldsymbol{M}(\boldsymbol{\gamma} \circ \boldsymbol{y})$ | $\gamma_i \sim \mathsf{Ber}(\pi)$, indep. of $Y_i$ but not $X_i$ | $\tilde{O}(\sqrt{\pi}\|\boldsymbol{M}\|_F)$ |
| Theorem 4.1: $\boldsymbol{x}^\top \boldsymbol{M} \boldsymbol{y}$ | $|Y_i| \leq 1$ a.s., $\mathbb{E} Y_i^2 \leq \pi$ | $\tilde{O}(\sqrt{\pi}\|\boldsymbol{M}\|_F)$ |

Table 1: Comparison of different variants of the Hanson-Wright inequality. In all variants, we assume that $(\boldsymbol{x}, \boldsymbol{y}) = (X_i, Y_i)_{i=1}^n$ are subgaussian, centered, and the pairs $(X_i, Y_i)$ are independent across $i$. We use $\circ$ to denote elementwise multiplication, which allows us to express hard sparsity with the sparsity mask $\boldsymbol{\gamma} \in \{0, 1\}^n$. The concentration radius corresponds to the size of typical fluctuations guaranteed by the concentration inequality, i.e. the $\epsilon$ needed for high probability guarantees.
[a] (Rudelson and Vershynin, 2013, Theorem 1.1); [b] (Park et al., 2022, Theorem 1); [c] (Wu and Sahai, 2023, Theorem 4)

proof of Theorem 4.1 is actually simpler than in Park et al. (2022); Wu and Sahai (2023), as bounded with soft sparsity turns out to be easier to work with than subgaussian with hard sparsity. We refer readers to Wu and Sahai (2023) for a more in-depth discussion of how these new "sparse" variants overcome the limitations of previous proof techniques used to study classification problems.

We briefly illustrate how Theorem 4.1 can be used to get tighter results throughout our analysis. A quick calculation reveals that the label vectors $\boldsymbol{\Delta} y$ and $\boldsymbol{y}_\alpha$ both have soft sparsity at level $\pi = O(1/k)$. However, $\boldsymbol{y}_\alpha$ does not have hard sparsity as required by the variants in Park et al. (2022); Wu and Sahai (2023). Since $\|\boldsymbol{M}\|_F^2 \leq n\|\boldsymbol{M}\|_2^2$, we obtain a concentration radius $\epsilon$ which scales like $\sqrt{n/k}$ rather than $\sqrt{n}$ (obtained via vanilla Hanson-Wright) or $n/\sqrt{k}$ (obtained via Cauchy-Schwarz). This gain is crucial to tightly analyzing the survival, contamination, and correlation structure.

### 4.3 Completing the proof sketch

Theorem 4.1 and the above insights about sparsity and independence allow us to prove the following bounds on the relative survival and contamination terms which are tight up to log factors; see the Appendix for more details. For brevity's sake, we introduce the notation $\mu \triangleq n^{q+r-1}$.

**Proposition 4.2** (Bounds on relative survival). *Suppose we are in the bi-level model. With probability at least $1 - O(1/n)$,*

$$\lambda_F \widehat{\boldsymbol{h}}_{\alpha,\beta}[\alpha] = \min\left\{\mu^{-1}, 1\right\} \Theta(n^{-\min\left\{t, \frac{1}{2}\right\}}) \sqrt{\log k}.$$

Next, we state our bounds on contamination.

**Proposition 4.3** (Bounds on contamination). *Suppose we are in the bi-level model. Then with probability at least $1 - O(1/n)$, the contamination satisfies*

$$\mathsf{CN}_{\alpha,\beta} = \underbrace{\min\left\{\mu^{-1}, 1\right\} \Theta(n^{\frac{r-t-1}{2}})}_{\text{favored features}} + \underbrace{\Theta(n^{\frac{1-t-p}{2}})}_{\text{unfavored features}}. \tag{26}$$

Translating the parameters in Propositions 4.2 and 4.3 we see that (i) the relative survival is diminished by a factor $1/k$ as long as $k = o(\sqrt{n})$, and a factor $1/\sqrt{n}$ for $k = \Omega(\sqrt{n})$ (this looseness ends up being negligible for the final result) and (ii) the contamination is diminished by a factor of $1/\sqrt{k}$. This essentially matches the expected behavior from the heuristic calculation in Subramanian et al. (2022). Together with some straightforward algebra, Propositions 4.2 and 4.3 allow us to compute the regimes where the survival-to-contamination ratio $\mathsf{SU}/\mathsf{CN}$ grows or decays polynomially. This yields the stated regimes in Conjecture 3.1; see Appendix A.1 for more details.

For technical reasons, the analogous bounds in Subramanian et al. (2022) are loose, giving rise to unnecessary conditions for good generalization such as $t < p - 2$ and $t < 2q + r - 2$. Moreover, we are able to give both upper and lower bounds on the survival and contamination terms, whereas they only give one-sided inequalities for each quantity.

# 5 Discussion

In this paper we resolve the main conjecture of Subramanian et al. (2022), identifying the exact regimes where an overparameterized linear model succeeds at multiclass classification. Our techniques also lay the foundation for investigating related generalization for other multiclass tasks and nonlinear algorithms. We hope that by bringing the rigorous proofs closer to the heuristic style of calculation, we open the path for analyzing more complicated and realistic models.

An important next step is to extend our results to more realistic spiked covariance models. For example, one typically observes power-law decay for the extreme eigenvalues in applications. We expect that the bi-level model can be relaxed to allow for constant deviations in the weightings for the favored, non-label defining features and power law decay for the unfavored features. The former change would likely only affect constants in certain areas of the argument that do not crucially depend on the exact constants involved, whereas the latter would likely just change the effective degree of overparameterization (Bartlett et al., 2020). However, even constant fluctuations in weighting for the label-defining features can lead to significant subtleties, as these constant deviations manifest as polynomially large variations in the number of examples of each class. Such heterogeneity between label-defining directions would likely lead to significantly messier conditions for generalization.

Another future direction is to move beyond Gaussian features. It is plausible that similar results would hold for vector subgaussian features which are rotationally invariant, allowing us to rotate into the basis where the features have diagonal covariance. One place where Gaussianity is crucially used is to obtain an explicit lower bound on the margin between the features.

As an example application, we sketch out how our proof techniques imply precise conditions for a variant of the learning task called multilabel classification. In a simple model for multilabel classification, each datapoint can have several of $k$ possible labels — corresponding to the positive valued features — but in the training set only one such correct label is provided at random for each datapoint. We deem that the model generalizes if for any queried label it successfully labels test inputs as positive or negative. We can use the MNI approach here to learn classifiers.

Some thought reveals that the main difference between multilabel classification and multiclass classification from a survival and contamination perspective is that positive features no longer need to outcompete other features. Thus, the main object of study would be the bilinear forms $z_j^\top A^{-1} y_\alpha$, which is possible thanks to Theorem 4.1. The survival and contamination terms are only affected by the expected values of these bilinear forms, but the expected values match the multiclass behavior up to log factors, which do not affect the regimes where SU/CN will grow or shrink polynomially. A similar analysis thus reveals that MNI will generalize in exactly the same regimes as in Conjecture 3.1. Here, the model generalizes in the sense that with high probability over the labels the model will correctly classify, and failure to generalize means that the model will do no better than a coin toss.

Perhaps surprisingly, resolving Conjecture 3.1 also implies that MNI is asymptotically *suboptimal* compared to a natural *non-interpolative* approach: simply make $\hat{f}_m$ equal to the average[6] of all positive training examples of class $m$. A straightforward analysis, detailed in the supplementary material, reveals this scheme fails to generalize exactly when $t < \min\{1 - r, p + 1 - 2(q + r)\}$, even in the regime where regression succeeds ($q + r < 1$). This is particularly interesting because we have shown that in the regime where regression succeeds, MNI generalizes only when $t < \min\{1 - r, p - 1\}$, which is a smaller region. In light of this gap, it would be interesting to identify the *information-theoretic* barrier for multiclass classification, especially within the broader context of statistical-computation gaps (see e.g. Wu and Xu (2021); Brennan and Bresler (2020)).

## Acknowledgments and Disclosure of Funding

DW acknowledges support from NSF Graduate Research Fellowship DGE-2146752. We acknowledge funding support from NSF AST-2132700 for this work. DW appreciates helpful discussions with Sidhanth Mohanty and Prasad Raghavendra.

---

[6]Note that Frei et al. (2023) point out that even leaky ReLU networks trained with a gradient flow can behave like averages of training examples.

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

# Contents

# A  Preliminaries and notation

For positive integers $n$, we use the shorthand $[n] \triangleq \{1, \ldots, n\}$. For a vector $\boldsymbol{v} \in \mathbb{R}^n$, $\|\boldsymbol{v}\|_2$ always denotes the Euclidean norm. We index entries by using square brackets, so $\boldsymbol{v}[j]$ denotes the $j$th entry of $\boldsymbol{v}$. For any matrix $\boldsymbol{M} \in \mathbb{R}^{m \times n}$, we denote its $ij$th entry by $m_{ij}$, $\|\boldsymbol{M}\|_2$ denotes the spectral norm, and $\|\boldsymbol{M}\|_F = \mathrm{Tr}(\boldsymbol{M}^\top \boldsymbol{M})$ denotes the Frobenius norm. We use $\sigma_{\max}(\boldsymbol{M})$ and $\sigma_{\min}(\boldsymbol{M})$ to denote the maximum and minimum singular values of $\boldsymbol{M}$, respectively. If $\boldsymbol{M} \in \mathbb{R}^{n \times n}$ is symmetric, we write $\mu_1(\boldsymbol{M}) \geq \mu_2(\boldsymbol{M}) \geq \ldots \geq \mu_n(\boldsymbol{M})$ to denote the ordered eigenvalues of $\boldsymbol{M}$. Given two vectors $\boldsymbol{v}, \boldsymbol{u} \in \mathbb{R}^n$, we write $\boldsymbol{v} \circ \boldsymbol{u} \in \mathbb{R}^n$ to denote the entrywise product of $\boldsymbol{v}$ and $\boldsymbol{u}$.

We make extensive use of big-$O$ notation. In this paragraph, $c$ refers to a positive constant which does not depend on $n$, and all statements hold for sufficiently large $n$. If $f(n) = O(g(n))$, then $f(n) \leq cg(n)$ for some $c$. If $f(n) = \tilde{O}(g(n))$, then $f(n) \leq cg(n) \operatorname{poly} \log(n)$ for some $c$. If $f(n) = o(g(n))$, then for all $c > 0$ we have $f(n) \leq cg(n)$. We write $f(n) = \Omega(g(n))$ if $f(n) \geq cg(n)$ for some $c$. Finally, we write $f(n) = \Theta(g(n))$ if there exists positive constants $c_1$ and $c_2$ such that $c_1 g(n) \leq f(n) \leq c_2 g(n)$.

Table 2: Notation

| Symbol | Definition | Dimension | Source |
|---|---|---|---|
| $k$ | Number of classes | Scalar | Sec. 2 |
| $n$ | Number of training points | Scalar | Sec. 2 |
| $d$ | Dimension of each point — the total number of features | Scalar | Sec. 2 |
| $s$ | The number of favored features | Scalar | Def. 1 |
| $a$ | The constant controlling the favored weights | Scalar | Def. 1 |
| $p$ | Parameter controlling overparameterization ($d = n^p$) | Scalar | Def. 1 |
| $r$ | Parameter controlling the number of favored features ($s = n^r$) | Scalar | Def. 1 |
| $q$ | Parameter controlling the favored weights ($a = n^{-q}$) | Scalar | Def. 1 |
| $t$ | Parameter controlling the number of classes ($k = c_k n^t$) | Scalar | Def. 1 |
| $c_k$ | The number of classes when $t = 0$ ($k = c_k n^t$) | Scalar | Def. 1 |
| $\lambda_j$ | Squared weight of the $j$th feature | Scalar | Def. 1 |
| $\boldsymbol{x}_i$ | $i$th training point (unweighted) | Length-$d$ vector | Sec. 2 |
| $\ell_i$ | Class label of $i$th training point | Scalar | Eqn. 1 |
| $\boldsymbol{w}_i$ | $i$th training point (weighted) | Length-$d$ vector | Eqn. 2 |
| $\boldsymbol{X}^w$ | Weighted feature matrix | ($n \times d$)-matrix | Eqn. 3 |
| $\boldsymbol{z}_j$ | The collected $j$th features of all training points | Length-$n$ vector | Eqn. 3 |
| $\boldsymbol{y}_m^{\mathrm{oh}}$ | One-hot encoding of all the training points for label $m$ | Length-$n$ vector | Eqn. 4 |
| $\boldsymbol{Y}^{\mathrm{oh}}$ | One-hot label matrix | ($n \times k$)-matrix | Eqn. 4 |
| $\boldsymbol{y}_m$ | Zero-mean encoding of the training points for label $m$ | Length-$n$ vector | Eqn. 5 |
| $\hat{\boldsymbol{f}}_m$ | Learned coefficients for label $m$ using min-norm interpolation | Length-$d$ vector | Eqn. 8 |
| $\boldsymbol{x}_{\mathrm{test}}$ | A single test point | Length-$d$ vector | Sec. 2 |
| $\boldsymbol{x}_{\mathrm{test}}^w$ | A single weighted test point | Length-$d$ vector | Sec. 2 |
| $\boldsymbol{A}$ | Gram matrix $\boldsymbol{A} = \boldsymbol{X}^w (\boldsymbol{X}^w)^\top$ | ($n \times n$)-matrix | Sec. 4 |
| $\mu_i(\boldsymbol{A})$ | The $i$th eigenvalue of matrix $\boldsymbol{A}$, sorted in descending order | Scalar | App. A |
| $\lambda_F$ | Squared favored feature weights: $\lambda_F = \frac{ad}{s}$ | Scalar | Def. 1 |
| $\lambda_U$ | Squared unfavored feature weights: $\lambda_F = \frac{(1-a)d}{d-s}$ | Scalar | Def. 1 |
| $\widehat{\boldsymbol{h}}_{\alpha,\beta}$ | Relative survival $\widehat{\boldsymbol{h}}_{\alpha,\beta}[j] = \lambda_j^{-1/2}(\hat{f}_\alpha[j] - \hat{f}_\beta[j])$ | Length-$d$ vector | Eqn. 16 |
| $\mathsf{CN}_{\alpha,\beta}$ | Normalizing factor $\mathsf{CN}_{\alpha,\beta} = \sqrt{\left( \sum_{j \notin \{\alpha,\beta\}} \lambda_j^2 (\widehat{\boldsymbol{h}}_{\beta,\alpha}[j])^2 \right)}$ | Scalar | Eqn. 19 |
| $\|\cdot\|_{\psi_2}$ | The subgaussian norm of a scalar random variable | Scalar | Eqn. 24 |
| $\|\cdot\|_{\psi_1}$ | The subexponential norm of a scalar random variable | Scalar | Eqn. 51 |
| $\mu$ | Factor controlling whether regression works, $\mu \triangleq n^{q+r-1}$ | Scalar | App. A.1 |
| $\boldsymbol{Z}_T$ | Unweighted subset of favored features, where $T \subseteq [s]$ | ($n \times |T|$)-matrix | App. B.2 |
| $\boldsymbol{W}_T$ | Weighted subset of favored features, $\boldsymbol{W}_T = \sqrt{\lambda_F} \boldsymbol{Z}_T$ | ($n \times |T|$)-matrix | App. B.2 |
| $\boldsymbol{A}_{-T}$ | Leave-$T$-out Gram matrix, where $T \subseteq [s]$, $\boldsymbol{A}_{-T} = \boldsymbol{A} - \boldsymbol{W}_T \boldsymbol{W}_T^\top$ | ($n \times n$)-matrix | Eqn. 74 |
| $\boldsymbol{H}_k$ | Hat matrix, $\boldsymbol{H}_k = \boldsymbol{W}_k \boldsymbol{A}_{-k}^{-1} \boldsymbol{W}_k$ | ($k \times k$)-matrix | Eqn. 53 |

Let us now describe the organization of the appendix. In Appendix A.1, we give a more detailed proof sketch and introduce the main propositions that complete the proof of Theorem 3.2. In Appendix B, we introduce the main tools that allow us to prove that the critical bilinear forms $\boldsymbol{z}_j^\top \boldsymbol{A}^{-1}\Delta y$ concentrate: our new variant of the Hanson-Wright inequality, the Woodbury inversion formula, and Wishart concentration to bound the spectra of the relevant random matrices that appear. In Appendix C we apply these tools to bound some useful quantities that repeatedly appear in the rest of the proofs. After that, we proceed to bound the survival, contamination, and correlation structure in Appendices D to F. In Appendix J, we present the analysis for the averaging scheme described in Section 5. Finally, we prove our new variant of Hanson-Wright (Theorem 4.1) in Appendix G.

## A.1 Proof of Theorem 3.2

In this section, we fill in some of the details of the proof sketch of Theorem 3.2. After recalling the beginning of the proof, we will split up the proof into two subtheorems: one for the positive result where MNI generalizes (Theorem A.2), and another for the negative result where MNI misclassifies (Theorem A.4).

Assume without loss of generality that the test point $\boldsymbol{x}_{\text{test}} \sim N(0, \boldsymbol{I}_d)$ has true label $\alpha$ for some $\alpha \in [k]$. Let $\boldsymbol{x}_{\text{test}}^w$ be the weighted version of this test point. From (10), an equivalent condition for misclassification is that for some $\beta \neq \alpha, \beta \in [k]$, we have $\widehat{\boldsymbol{f}}_\alpha^\top \boldsymbol{x}_{\text{test}}^w < \widehat{\boldsymbol{f}}_\beta^\top \boldsymbol{x}_{\text{test}}^w$, i.e. the score for $\beta$ outcompetes the score for $\alpha$. Define the Gram matrix $\boldsymbol{A} \triangleq \boldsymbol{X}^w(\boldsymbol{X}^w)^\top$, the relative label vector $\Delta y \triangleq \boldsymbol{y}_\alpha - \boldsymbol{y}_\beta \in \{-1, 0, 1\}^n$, and the relative survival vector $\widehat{\boldsymbol{h}}_{\alpha,\beta} \in \mathbb{R}^d$ which compares the signal from $\alpha$ and $\beta$:

$$\widehat{\boldsymbol{h}}_{\alpha,\beta}[j] \triangleq \lambda_j^{-1/2}(\widehat{\boldsymbol{f}}_\alpha[j] - \widehat{\boldsymbol{f}}_\beta[j]) \tag{27}$$

$$= \boldsymbol{z}_j^\top \boldsymbol{A}^{-1}\Delta y, \tag{28}$$

where to obtain the last line we have used the explicit formula for the MNI classifiers (8). By converting the misclassification condition into the unweighted feature space we see that we will have errors when

$$\lambda_\alpha \widehat{\boldsymbol{h}}_{\alpha,\beta}[\alpha]\boldsymbol{x}_{\text{test}}[\alpha] - \lambda_\beta \widehat{\boldsymbol{h}}_{\beta,\alpha}[\beta]\boldsymbol{x}_{\text{test}}[\beta] < \sum_{j \notin \{\alpha,\beta\}} \lambda_j \widehat{\boldsymbol{h}}_{\beta,\alpha}[j]\boldsymbol{x}_{\text{test}}[j]. \tag{29}$$

Define the contamination term $\mathsf{CN}_{\alpha,\beta}$:

$$\mathsf{CN}_{\alpha,\beta} \triangleq \sqrt{\sum_{j \notin \{\alpha,\beta\}} \lambda_j^2 (\widehat{\boldsymbol{h}}_{\beta,\alpha}[j])^2}. \tag{30}$$

Note that $\mathsf{CN}_{\alpha,\beta}$ normalizes the RHS of (29) into a standard Gaussian. Indeed, define

$$Z^{(\beta)} \triangleq \frac{1}{\mathsf{CN}_{\alpha,\beta}} \sum_{j \notin \{\alpha,\beta\}} \lambda_j \widehat{\boldsymbol{h}}_{\beta,\alpha}[j]\boldsymbol{x}_{\text{test}}[j] \sim N(0, 1). \tag{31}$$

Since $\alpha, \beta \in [k]$ are favored, we have $\lambda_\alpha = \lambda_\beta = \lambda_F$. Hence an equivalent condition for misclassification is that there exists some $\beta \neq \alpha, \beta \in [k]$ such that

$$\frac{\lambda_F}{\mathsf{CN}_{\alpha,\beta}}(\widehat{\boldsymbol{h}}_{\alpha,\beta}[\alpha]\boldsymbol{x}_{\text{test}}[\alpha] - \widehat{\boldsymbol{h}}_{\beta,\alpha}[\beta]\boldsymbol{x}_{\text{test}}[\beta]) < Z^{(\beta)}. \tag{32}$$

We will translate the above criterion into *sufficient* conditions for correct classification and misclassification and analyze these two cases separately.

First, let us present our tight characterization of the survival and contamination terms, which will be useful for both sides of the theorem. Recall our definition of $\mu \triangleq n^{q+r-1}$; whether this quantity polynomially shrinks or decays directly determines if regression works or fails.

**Proposition 4.2** (Bounds on relative survival). *Suppose we are in the bi-level model. With probability at least $1 - O(1/n)$,*

$$\lambda_F \widehat{\boldsymbol{h}}_{\alpha,\beta}[\alpha] = \min\left\{\mu^{-1}, 1\right\}\Theta(n^{-\min\left\{t, \frac{1}{2}\right\}})\sqrt{\log k}.$$

**Proposition 4.3** (Bounds on contamination). *Suppose we are in the bi-level model. Then with probability at least $1 - O(1/n)$, the contamination satisfies*

$$\mathsf{CN}_{\alpha,\beta} = \underbrace{\min\left\{\mu^{-1}, 1\right\}\Theta(n^{\frac{r-t-1}{2}})}_{\text{favored features}} + \underbrace{\Theta(n^{\frac{1-t-p}{2}})}_{\text{unfavored features}}. \tag{26}$$

We defer the proof of Proposition 4.2 to Appendix D and the proof of Proposition 4.3 to Appendix E. Combining Propositions 4.2 and 4.3 yields the following sufficient conditions for when the $\mathsf{SU}/\mathsf{CN}$ ratio grows or shrinks polynomially.

**Proposition A.1** (Regimes for survival-to-contamination). *Under the bi-level ensemble model (Definition 1), when the true data generating process is 1-sparse (Assumption 1), as $n \to \infty$, with probability at least $1 - O(1/n)$, the survival-to-contamination ratio satisfies*

$$\frac{\min_\beta \lambda_F \widehat{\boldsymbol{h}}_{\alpha,\beta}[\alpha]}{\max_\beta \mathsf{CN}_{\alpha,\beta}} \geq n^v \text{ for some } v > 0 \text{ if } t < \min\left\{1 - r, p + 1 - 2\max\left\{1, q + r\right\}\right\} \tag{33}$$

$$\frac{\max_\beta \lambda_F \left|\widehat{\boldsymbol{h}}_{\alpha,\beta}[\alpha]\right|}{\min_\beta \mathsf{CN}_{\alpha,\beta}} \leq n^{-w} \text{ for some } w > 0 \text{ if } t > \min\left\{1 - r, p + 1 - 2\max\left\{1, q + r\right\}\right\} \tag{34}$$

*Here, the max and min are being taken over $\beta \neq \alpha, \beta \in [k]$.*

*Proof.* We do casework on whether we want to prove an upper bound or lower bound on $\mathsf{SU}/\mathsf{CN}$. First, suppose we want to prove the lower bound, so assume $t < \min\left\{1 - r, p + 1 - 2\max\left\{1, q + r\right\}\right\}$. Since $t < r$ by the definition of the bi-level ensemble (Definition 1), we have that $t < \frac{1}{2}$. So by union bounding over $\beta$, Proposition 4.2 implies that with probability $1 - O(1/n)$

$$\min_\beta \lambda_F \widehat{\boldsymbol{h}}_{\alpha,\beta}[\alpha] \geq \min\left\{\mu^{-1}, 1\right\}\Omega(n^{-t})\sqrt{\log k}. \tag{35}$$

Then from Proposition 4.3, by union bounding over $\beta$ we see that with probability $1 - O(1/n)$,

$$\max_\beta \mathsf{CN}_{\alpha,\beta} \leq \underbrace{\min\left\{\mu^{-1}, 1\right\}\tilde{O}(n^{\frac{r-t-1}{2}})}_{\text{favored features}} + \underbrace{\tilde{O}(n^{\frac{1-t-p}{2}})}_{\text{unfavored features}}.$$

Let us combine these two bounds. If we compare the survival to the contamination coming from favored features, we obtain

$$\frac{\min\left\{\mu^{-1}, 1\right\}n^{-t}\sqrt{\log k}}{\min\left\{\mu^{-1}, 1\right\}\tilde{O}(n^{\frac{r-t-1}{2}})} \geq \frac{n^{-t - \frac{r-t-1}{2}}}{\log(nsk)} \tag{36}$$

$$\geq \frac{n^{\frac{1-r-t}{2}}}{\log(nsk)}, \tag{37}$$

where we have included the explicit poly log factors for precision. Hence, if $t < 1 - r$, the numerator grows polynomially and dominates the denominator. Now let's compare the survival to the contamination coming from unfavored features. This yields

$$\frac{\min\left\{\mu^{-1}, 1\right\}n^{-t}\sqrt{\log k}}{\tilde{O}(n^{\frac{1-t-p}{2}})} \geq \frac{\min\left\{\mu^{-1}, 1\right\}n^{-t - \frac{1-t-p}{2}}}{\log(nsk)} \tag{38}$$

$$\geq \frac{n^{-\max\left\{q+r-1, 0\right\}} \cdot n^{\frac{p-t-1}{2}}}{\log(nsk)} \tag{39}$$

$$\geq \frac{n^{\frac{p+1-2\max\left\{1, q+r\right\}-t}{2}}}{\log(nsk)}. \tag{40}$$

Hence, by union bounding, we see that with probability $1 - O(1/n)$,

$$\min_\beta \frac{\lambda_F \widehat{\boldsymbol{h}}_{\alpha,\beta}[\alpha]}{\mathsf{CN}_{\alpha,\beta}} \geq n^v, \tag{41}$$

where $v \triangleq \frac{1}{4}(\min\{1 - r, p + 1 - 2\max\{1, q + r\}\} - t) > 0$ by assumption.

For the upper bound, suppose $t > \min\{1 - r, p + 1 - 2\max\{1, q + r\}\}$. Hence $t > 0$, and by union bounding we conclude that with probability at least $1 - O(1/n)$,

$$\max_\beta \lambda_F \left|\widehat{\boldsymbol{h}}_{\alpha,\beta}[\alpha]\right| \leq \min\{\mu^{-1}, 1\} O(n^{-\frac{1}{2}})\sqrt{\log k} \tag{42}$$

and

$$\min_\beta \mathsf{CN}_{\alpha,\beta} \geq \min\{\mu^{-1}, 1\}\Omega(n^{\frac{r-t-1}{2}}) + \Omega(n^{\frac{1-t-p}{2}}). \tag{43}$$

Combining these and union bounding yields that with probability $1 - O(1/n)$,

$$\min_\beta \frac{\lambda_F \widehat{\boldsymbol{h}}_{\alpha,\beta}[\alpha]}{\mathsf{CN}_{\alpha,\beta}} \leq n^{-w}, \tag{44}$$

where $w \triangleq \frac{1}{4}(t - \min\{1 - r, p + 1 - 2\max\{1, q + r\}\}) > 0$ by asssumption. $\square$

We now sketch out a proof of both the positive and negative sides of Theorem 3.2. We point out that the regimes for generalization and misclassification exactly match the regimes above for where the SU/CN ratio grows or shrinks polynomially.

**Theorem A.2** (Positive side of Theorem 3.2). *Under the bi-level ensemble model (Definition 1), when the true data generating process is 1-sparse (Assumption 1), as $n \to \infty$, the probability of misclassification for MNI satisfies $\Pr[\mathcal{E}_{\mathsf{err}}] \to 0$ if*

$$t < \min\{1 - r, p + 1 - 2\max\{1, q + r\}\}.$$

*Proof sketch.* For correct classification, it suffices for the minimum value of the LHS of Eq. (32) to outcompete the maximum value of the RHS, where the max is taken over $\beta \in [k], \beta \neq \alpha$. Some algebra, as in Subramanian et al. (2022), shows that we correctly classify if

$$\underbrace{\frac{\min_\beta \lambda_F \widehat{\boldsymbol{h}}_{\alpha,\beta}[\alpha]}{\max_\beta \mathsf{CN}_{\alpha,\beta}}}_{\text{SU/CN ratio}} \left( \underbrace{\min_\beta (\boldsymbol{x}_{\mathsf{test}}[\alpha] - \boldsymbol{x}_{\mathsf{test}}[\beta])}_{\text{closest feature margin}} - \underbrace{\max_\beta |\boldsymbol{x}_{\mathsf{test}}[\beta]|}_{\text{largest competing feature}} \cdot \underbrace{\max_\beta \left| \frac{\widehat{\boldsymbol{h}}_{\alpha,\beta}[\alpha] - \widehat{\boldsymbol{h}}_{\beta,\alpha}[\beta]}{\widehat{\boldsymbol{h}}_{\alpha,\beta}[\alpha]} \right|}_{\text{survival variation}} \right)$$
$$> \underbrace{\max_\beta Z^{(\beta)}}_{\text{normalized contamination}}. \tag{45}$$

By our lower bound on the survival to contamination ratio (Proposition A.1), assuming $t < \min\{1 - r, p + 1 - 2(q + r)\}$, then with probability at least $1 - O(1/n)$ we have that $\frac{\lambda_F \widehat{\boldsymbol{h}}_{\alpha,\beta}[\alpha]}{\mathsf{CN}_{\alpha,\beta}} \geq n^u$ for some constant $u > 0$. By Lemmas B.2 and B.3 in Subramanian et al. (2022) for every $\epsilon > 0$, with probability at least $1 - \epsilon$, we have $\min_\beta \boldsymbol{x}_{\mathsf{test}}[\alpha] - \boldsymbol{x}_{\mathsf{test}}[\beta] \geq \Omega(\frac{1}{\sqrt{\log k}})$.

Next, by standard subgaussian maxima tail bounds we have that $|\boldsymbol{x}_{\mathsf{test}}[\beta]| \leq 2\sqrt{\log(nk)}$ and $Z^{(\beta)} \leq 2\sqrt{\log(nk)}$ with probability at least $1 - O(1/nk)$. Finally, applying our upper bound on the relative survival variance (Proposition A.3, which we prove below), the survival variation is at most a polynomially decaying $n^{-w}$ with probability at least $1 - O(1/nk)$.

By union bounding, we see that with probability at least $1 - O(1/n) - \epsilon$, the LHS outcompetes the RHS, implying that the model correctly classifies.

$\square$

In fact, given Proposition 4.2, it is straightforward to bound the survival variation.

**Proposition A.3** (Upper bound on the survival variation). *Suppose that $t < 1 - r$. With probability at least $1 - 2/n$, we have*

$$\left| \frac{\widehat{\boldsymbol{h}}_{\alpha,\beta}[\alpha] - \widehat{\boldsymbol{h}}_{\beta,\alpha}[\beta]}{\widehat{\boldsymbol{h}}_{\alpha,\beta}[\alpha]} \right| \leq c_1 n^{-w}, \tag{46}$$

*where $c_1$ and $w$ are both positive constants.*

*Proof.* Since we have $\widehat{h}_{\alpha,\beta}[\alpha] = z_\alpha^\top A^{-1} \Delta y$, the survival variation is

$$\frac{\widehat{h}_{\alpha,\beta}[\alpha] - \widehat{h}_{\beta,\alpha}[\beta]}{\widehat{h}_{\alpha,\beta}[\alpha]} = \frac{z_\alpha^\top A^{-1} \Delta y + z_\beta^\top A^{-1} \Delta y}{z_\alpha^\top A^{-1} \Delta y}$$

Since $t < 1 - r$ and $t < r$ by definition, we know that $t < \frac{1}{2}$, and we can apply Proposition 4.2 to see that with probability at least $1 - 2/n$ we have

$$z_\alpha^\top A^{-1} \Delta y = \max\left\{\mu^{-1}, 1\right\} n^{-t}(1 \pm O(n^{-\kappa_5}))\sqrt{\log k} = -z_\beta^\top A^{-1} \Delta y$$

Hence we have

$$\left|\frac{\widehat{h}_{\alpha,\beta}[\alpha] - \widehat{h}_{\beta,\alpha}[\beta]}{\widehat{h}_{\alpha,\beta}[\alpha]}\right| \le c_1 n^{-\kappa_5} \tag{47}$$

where $c_1$ is an appropriately defined positive constant. $\qquad\square$

**Theorem A.4** (Negative side of Theorem 3.2). *Under the bi-level ensemble model (Definition 1), when the true data generating process is 1-sparse (Assumption 1), as $n \to \infty$, the probability of misclassification for MNI satisfies $\Pr[\mathcal{E}_{\mathsf{err}}] \to 1$ if*

$$t > \min\left\{1 - r, p + 1 - 2\max\left\{1, q + r\right\}\right\}.$$

*Proof sketch.* On the other hand, for misclassification it suffices for the maximum *absolute* value of the LHS of Eq. (32) to be outcompeted by the maximum value of the RHS. Some manipulations yield the following sufficient condition for misclassification:

$$\underbrace{\frac{\max_\beta \lambda_F\left(\left|\widehat{h}_{\alpha,\beta}[\alpha]\right| + \left|\widehat{h}_{\beta,\alpha}[\beta]\right|\right)}{\min_\beta \mathsf{CN}_{\alpha,\beta}}}_{\mathsf{SU/CN}\text{ ratio}} \cdot \underbrace{\max_{\gamma \in [k]} |x_{\mathsf{test}}[\gamma]|}_{\text{largest label-defining feature}} < \underbrace{\max_\beta Z^{(\beta)}}_{\text{normalized contamination}}. \tag{48}$$

Within the misclassification regimes in Conjecture 3.1, Proposition A.1 implies that the survival-to-contamination ratio $\mathsf{SU/CN}$ *shrinks* at a polynomial rate $n^{-w}$ for some $w > 0$. By standard subgaussian maximal inequalities, the largest label-defining feature is $O(\sqrt{\log(nk)})$ with high probability. Gaussian anticoncentration implies that for some $\beta \ne \alpha, \beta \in [k]$, $Z^{(\beta)}$ outcompetes the LHS, which is bounded above by $n^{-w}$, with probability at least $\frac{1}{2} - o(1)$. Hence, we conclude that the model will misclassify with rate at least $\frac{1}{2}$ asymptotically.

Let us now describe how to boost the misclassification rate to $1 - o(1)$. Notice that the above argument only considered the competition between the LHS of Eq. (48) and one of the $Z^{(\beta)}$'s on the RHS instead of the maximum $Z^{(\beta)}$. It's not hard to see from the definition of $Z^{(\beta)}$ in Eq. (31) that the $Z^{(\beta)}$ are jointly Gaussian. For intuition's sake, assuming the $Z^{(\beta)}$ were *independent*, then $\max_\beta Z^{(\beta)}$ would outcompete with probability $(\frac{1}{2} - o(1))^{k-1}$.

In reality, the $Z^{(\beta)}$ are correlated, but we are able to show that the maximum correlation between the $Z^{(\beta)}$ is $\frac{1}{2} + o(1)$ with high probability. An application of Slepian's lemma (Slepian (1962)) and some explicit bounds on orthant probabilities (Pinasco et al. (2021)) implies that $\max_\beta Z^{(\beta)} > 0$ with probability at least $1 - \frac{1}{k^{1+o(1)}}$. An application of anticoncentration for Gaussian maxima (Chernozhukov et al., 2015) implies that $\max_\beta Z^{(\beta)} > n^{-w}$ with probability $1 - o(1)$, which finishes off the proof. $\qquad\square$

To fill in the details of the above proof sketch, we will prove the following proposition in Appendix F.

**Proposition A.5** (Correlation bound). *Assume we are in the bi-level ensemble model (Definition 1), the true data generating process is 1-sparse (Assumption 1), and the number of classes scales with $n$ (i.e. $t > 0$). Then for every $\epsilon > 0$, we have*

$$\Pr\left[\max_{\beta \in [k], \beta \ne \alpha} Z^{(\beta)} > n^{-u}\right] \ge 1 - \Theta\left(\frac{1}{k^{1+o(1)}}\right) - \epsilon \tag{49}$$

*for sufficiently large $n$ and any $u > 0$.*

# B  Main tools

In this section we introduce our suite of technical tools that allow us to prove the desired rates of growth for survival, contamination, and correlation.

## B.1  Hanson-Wright Inequality

As established in Section 4, we need to use the Hanson-Wright inequality to prove our tight characterization of generalization. For the sake of precision, we explicitly state our definitions of subgaussian and subexponential which we use throughout the rest of the paper.

The subgaussian norm $\|\xi\|_{\psi_2}$ of a random variable $\xi$ is defined as in Rudelson and Vershynin (2013),

$$\|\xi\|_{\psi_2} = \inf_{K>0} \left\{ K : \mathbb{E}\exp\left(\xi^2/K^2\right) \leq 2 \right\}. \tag{50}$$

The sub-exponential norm $\|\xi\|_{\psi_1}$ is defined as in Vershynin (2018, Definition 2.7.5):

$$\|\xi\|_{\psi_1} = \inf_{K>0} \left\{ K : \mathbb{E}\exp(|\xi|/K) \leq 2 \right\}. \tag{51}$$

We will occassionally need to use the following variant of Hanson-Wright for nonsparse bilinear forms, first proved in Park et al. (2021).

**Theorem B.1** (Hanson-Wright for bilinear forms without sparsity). *Let $\boldsymbol{x} = (X_1, \ldots, X_n) \in \mathbb{R}^n$ and $\boldsymbol{y} \in (Y_1, \ldots, Y_n)$ be random vectors such that the pairs $(X_i, Y_i)$ are all independent of each other (however $X_i$ and $Y_i$ can be correlated). Assume also that $\mathbb{E}[X_i] = \mathbb{E}[Y_i] = 0$ and $\max\left\{\|X_i\|_{\psi_2}, \|Y_i\|_{\psi_2}\right\} \leq K$. Then there exists an absolute constant $c > 0$ such that for all $\boldsymbol{M} \in \mathbb{R}^{n \times n}$ and $\epsilon \geq 0$ we have*

$$\Pr\left[|\boldsymbol{x}^\top \boldsymbol{M} \boldsymbol{y} - \mathbb{E}[\boldsymbol{x}^\top \boldsymbol{M} \boldsymbol{y}]| > \epsilon\right] \leq 2\exp\left(-c\min\left\{\frac{\epsilon^2}{K^4\|\boldsymbol{M}\|_F^2}, \frac{\epsilon}{K^2\|\boldsymbol{M}\|_2}\right\}\right). \tag{52}$$

Finally, we restate our new version of Hanson-Wright for bilinear forms with soft sparsity, which we prove in Appendix G.

**Theorem 4.1** (Hanson-Wright for bilinear forms with soft sparsity). *Let $\boldsymbol{x} = (X_1, \ldots, X_n) \in \mathbb{R}^n$ and $\boldsymbol{y} \in (Y_1, \ldots, Y_n) \in \mathbb{R}^n$ be random vectors such that $(X_i, Y_i)$ are independent pairs of (possibly correlated) centered random variables such that $\|X_i\|_{\psi_2} \leq K$ and $Y_i$ has soft sparsity at level $\pi$, i.e. $|Y_i| \leq 1$ almost surely, and $\mathbb{E}[Y_i^2] \leq \pi$. Assume that conditioned on $Y_j$, $\|X_j\|_{\psi_2} \leq K$. Then there exists an absolute constant $c > 0$ such that for all $\boldsymbol{M} \in \mathbb{R}^{n \times n}$ and $\epsilon \geq 0$ we have*

$$\Pr\left[|\boldsymbol{x}^\top \boldsymbol{M} \boldsymbol{y} - \mathbb{E}[\boldsymbol{x}^\top \boldsymbol{M} \boldsymbol{y}]| > \epsilon\right] \leq 2\exp\left(-c\min\left\{\frac{\epsilon^2}{K^2\pi\|\boldsymbol{M}\|_F^2}, \frac{\epsilon}{K\|\boldsymbol{M}\|_2}\right\}\right). \tag{25}$$

## B.2  Gram matrices and the Woodbury formula

In order to apply Hanson-Wright to the bilinear form $\boldsymbol{x}^\top \boldsymbol{M} \boldsymbol{y}$, we need to have a deterministic matrix $\boldsymbol{M}$ such that the hypotheses are satisfied. However, in our setting we study bilinear forms such as $\boldsymbol{z}_j^\top \boldsymbol{A}^{-1} \boldsymbol{\Delta} y$. Here, the inverse Gram matrix $\boldsymbol{A}^{-1}$ is not independent of $\boldsymbol{z}_j$ or $\boldsymbol{\Delta} y$, so we cannot simply condition on $\boldsymbol{A}^{-1}$. The way around this is to cleverly decompose $\boldsymbol{A}^{-1}$ using the so-called Woodbury inversion formula (stated formally below), which generalizes the leave-one-out trick and Sherman-Morrison used to study binary classification in Muthukumar et al. (2021). To that end, we will explicitly decompose the Gram matrix $\boldsymbol{A} \triangleq \sum_{j \in [d]} \lambda_j \boldsymbol{z}_j \boldsymbol{z}_j^\top$ based on whether the features $\boldsymbol{z}_j$ are favored or not.

We now introduce some notation to keep track of which matrices contain or leave out which indices. In general, we use subscripts to denote which sets of features we preserve or leave out; we use a minus sign to signify leaving out. The $k$ label-defining features are represented with a subscript $k$, whereas the $s - k$ favored but not label defining features are represented with a subscript $F$. The rest of the $d - s$ unfavored features are represented with a subscript $U$.

For notational convenience, we introduce some new notation for the weighted features, as the superscript $w$ to denote weighted features is rather cumbersome. We denote the weighted label-defining feature matrix by $\boldsymbol{W}_k \triangleq [\boldsymbol{w}_1 \quad \cdots \quad \boldsymbol{w}_k] \in \mathbb{R}^{n \times k}$ , where the vectors $\boldsymbol{w}_i \triangleq \sqrt{\lambda_i} \boldsymbol{z}_i \in \mathbb{R}^n$ denote the weighted observations for feature $i$. Define the unweighted label-defining feature matrix $\boldsymbol{Z}_k \triangleq [\boldsymbol{z}_1 \quad \cdots \quad \boldsymbol{z}_k] \in \mathbb{R}^{n \times k}$. Similarly, define $\boldsymbol{W}_F \triangleq [\boldsymbol{w}_{k+1} \quad \cdots \quad \boldsymbol{w}_s] \in \mathbb{R}^{n \times (s-k)}$, which contains the rest of the weighted favored features and the corresponding unweighted version $\boldsymbol{Z}_F$.

Let $\boldsymbol{A}_{-k} \triangleq \sum_{i \notin [k]} \boldsymbol{w}_i \boldsymbol{w}_i^\top$ denote the leave-$k$-out Gram matrix which removes the $k$ label-defining features. Similarly, let $\boldsymbol{A}_{-F} \triangleq \sum_{i \notin [s] \setminus [k]} \boldsymbol{w}_i \boldsymbol{w}_i^\top \in \mathbb{R}^{n \times n}$ to denote leave-$(s-k)$-out Gram matrix which removes the favored but not label-defining features. Finally, let $\boldsymbol{A}_U \triangleq \sum_{i \notin [s]} \boldsymbol{w}_i \boldsymbol{w}_i^\top \in \mathbb{R}^{n \times n}$ denote the leave-$s$-out matrix which only retains the unfavored features. We will also sometimes write $\boldsymbol{A}_{-s}$ instead of $\boldsymbol{A}_U$ to emphasize that the $s$ favored features have all been removed.

Define the so-called hat matrices by

$$\boldsymbol{H}_k \triangleq \boldsymbol{W}_k^\top \boldsymbol{A}_{-k}^{-1} \boldsymbol{W}_k \in \mathbb{R}^{k \times k} \tag{53}$$

$$\boldsymbol{H}_F \triangleq \boldsymbol{W}_F^\top \boldsymbol{A}_{-F}^{-1} \boldsymbol{W}_F \in \mathbb{R}^{(s-k) \times (s-k)}. \tag{54}$$

These hat matrices appear in the Woodbury inversion formula. For the sake of notational compactness, define

$$\boldsymbol{M}_k \triangleq \boldsymbol{W}_k (\boldsymbol{I}_k + \boldsymbol{H}_k)^{-1} \boldsymbol{W}_k^\top \in \mathbb{R}^{n \times n} \tag{55}$$

$$\boldsymbol{M}_F \triangleq \boldsymbol{W}_F (\boldsymbol{I}_{s-k} + \boldsymbol{H}_F)^{-1} \boldsymbol{W}_F^\top \in \mathbb{R}^{n \times n}. \tag{56}$$

The Woodbury inversion formula yields

$$\boldsymbol{A}^{-1} = (\boldsymbol{W}_k \boldsymbol{W}_k^\top + \boldsymbol{A}_{-k})^{-1} \tag{57}$$

$$= \boldsymbol{A}_{-k}^{-1} - \boldsymbol{A}_{-k}^{-1} \boldsymbol{W}_k (\boldsymbol{I}_k + \boldsymbol{H}_k)^{-1} \boldsymbol{W}_k^\top \boldsymbol{A}_{-k}^{-1} \tag{58}$$

$$= \boldsymbol{A}_{-k}^{-1} - \boldsymbol{A}_{-k}^{-1} \boldsymbol{M}_k \boldsymbol{A}_{-k}^{-1}. \tag{59}$$

Left multiplying (58) by $\boldsymbol{W}_k^\top$ yields

$$\boldsymbol{W}_k^\top \boldsymbol{A}^{-1} = \boldsymbol{W}_k^\top \boldsymbol{A}_{-k}^{-1} - \boldsymbol{H}_k (\boldsymbol{I}_k + \boldsymbol{H}_k)^{-1} \boldsymbol{W}_k^\top \boldsymbol{A}_{-k}^{-1} \tag{60}$$

$$= (\boldsymbol{I}_k - \boldsymbol{H}_k (\boldsymbol{I}_k + \boldsymbol{H}_k)^{-1}) \boldsymbol{W}_k^\top \boldsymbol{A}_{-k}^{-1} \tag{61}$$

$$= (\boldsymbol{I}_k + \boldsymbol{H}_k)^{-1} \boldsymbol{W}_k^\top \boldsymbol{A}_{-k}^{-1}. \tag{62}$$

We can derive completely analogous identities using $\boldsymbol{A}_{-F}^{-1}$ instead of $\boldsymbol{A}_{-k}^{-1}$. The above exposition is summarized by the following lemma.

**Lemma B.2.** *We have*

$$\boldsymbol{W}_k^\top \boldsymbol{A}^{-1} \boldsymbol{\Delta} y = (\boldsymbol{I}_k + \boldsymbol{H}_k)^{-1} \boldsymbol{W}_k^\top \boldsymbol{A}_{-k}^{-1} \boldsymbol{\Delta} y \tag{63}$$

$$\boldsymbol{W}_F^\top \boldsymbol{A}^{-1} \boldsymbol{\Delta} y = (\boldsymbol{I}_{s-k} + \boldsymbol{H}_F)^{-1} \boldsymbol{W}_F^\top \boldsymbol{A}_{-F}^{-1} \boldsymbol{\Delta} y. \tag{64}$$

Lemma B.2 is quite powerful. Indeed, consider the action of the linear operator $\boldsymbol{W}_k^\top \boldsymbol{A}^{-1} : \mathbb{R}^n \to \mathbb{R}^k$ on $\boldsymbol{\Delta} y$. The action is identical to that of the linear operator $\boldsymbol{W}_k^\top \boldsymbol{A}_{-k}^{-1} : \mathbb{R}^n \to \mathbb{R}^k$, up to some invertible transformation. This new linear operator is nice because $\boldsymbol{A}_{-k}^{-1}$ is independent of $\boldsymbol{W}_k$ and $\boldsymbol{\Delta} y$, as it removes all of the label-defining features. Reclaiming independence sets the stage for using our variant of Hanson-Wright.

How does the invertible operator $(\boldsymbol{I}_k + \boldsymbol{H}_k)^{-1}$ act? Our general strategy is to show that $\boldsymbol{H}_k$ is itself close to a scaled identity matrix, i.e. $\boldsymbol{H}_k \approx \nu \boldsymbol{I}_k$ for an appropriately defined $\nu$. Then for any $i \in [k]$, we have that

$$\boldsymbol{w}_i^\top \boldsymbol{A}^{-1} \boldsymbol{\Delta} y \approx (1 + \nu)^{-1} \boldsymbol{w}_i^\top \boldsymbol{A}_{-k}^{-1} \boldsymbol{\Delta} y.$$

Of course, there will be some error in this approximation, as $\boldsymbol{H}_k$ is not *exactly* equal to $\nu \boldsymbol{I}_k$. Nevertheless, we can bound away the error that arises from this approximation.

## B.3 Concentration of spectrum

As foreshadowed in the previous section, we will leverage the fact that the hat matrices such as $\boldsymbol{H}_k$ are close to a scaled identity. To formalize this, we appeal to random matrix theory and show that the spectra of various random matrices are very close to being flat (i.e. all eigenvalues are within $1 + o(1)$ of each other). To that end, we present the following standard characterization of the spectrum of a standard Wishart matrix, which is Equation 2.3 in Rudelson and Vershynin (2010).

**Lemma B.3** (Concentration of spectrum for Wishart matrices). *Let $\boldsymbol{M} \in \mathbb{R}^{M \times m}$ with $M > m$ be a real matrix with iid $N(0,1)$ entries. Then for any $\epsilon \geq 0$, we have with probability at least $1 - 2e^{-\epsilon^2/2}$ that*

$$\sqrt{M} - \sqrt{m} - \epsilon \leq \sigma_{\min}(\boldsymbol{M}) \leq \sigma_{\max}(\boldsymbol{M}) \leq \sqrt{M} + \sqrt{m} + \epsilon. \tag{65}$$

*In other words, the singular values of $\boldsymbol{M}$ satisfy subgaussian concentration.*

Since $\mu_m(\boldsymbol{M}^\top \boldsymbol{M}) = \sigma_{\min}(\boldsymbol{M})^2$ and $\mu_1(\boldsymbol{M}^\top \boldsymbol{M}) = \sigma_{\max}(\boldsymbol{M})^2$, we can conclude that if $m = o(M)$, then for any $\epsilon > 0$ we have

$$M - 2\sqrt{Mm} - \epsilon + o(\sqrt{Mm}) \leq \mu_m(\boldsymbol{M}^\top \boldsymbol{M}) \leq \mu_1(\boldsymbol{M}^\top \boldsymbol{M}) \leq M + 2\sqrt{Mm} + \epsilon + o(\sqrt{Mm}), \tag{66}$$

with probability at least $1 - 2e^{-\epsilon^2/2}$.

On the other hand, consider $\boldsymbol{M}\boldsymbol{M}^\top \in \mathbb{R}^{M \times M}$. Its spectrum is just that of $\boldsymbol{M}^\top \boldsymbol{M} \in \mathbb{R}^{m \times m}$ with an additional $M - m$ zeros corresponding to the fact that $m < M$.

We can use Lemma B.3 to prove concentration of the spectrum of the various matrices introduced in Appendix B.2. Let us summarize some convenient forms of these results; their proofs are deferred to Appendix H.

**Proposition B.4** (Gram matrices have a flat spectrum). *Recall that $\boldsymbol{A}_U = \boldsymbol{A}_{-s} = \sum_{j>s} \lambda_j \boldsymbol{z}_j \boldsymbol{z}_j^\top \in \mathbb{R}^{n \times n}$ is the unfavored Gram matrix and $\boldsymbol{A}_{-k} = \sum_{j>k} \lambda_j \boldsymbol{z}_j \boldsymbol{z}_j^\top \in \mathbb{R}^{n \times n}$ is the leave-k-out Gram matrix.*

*Then the following hold with probability at least $1 - 2e^{-n} - 2e^{-\sqrt{n}}$,*

(a) *For all $i \in [n]$, we have $\mu_i(\boldsymbol{A}_U) = n^p(1 \pm O(n^{-\kappa_7}))$.*

(b) *For all $i \in [s-k]$, we have*

$$\mu_i(\boldsymbol{A}_{-k}) = (1 + \mu^{-1})n^p(1 \pm O(n^{-\kappa_9})), \tag{67}$$

*where $\kappa_9$ is a positive constant. Moreover, for all $i \in [n] \setminus [s-k]$, we have*

$$\mu_i(\boldsymbol{A}_{-k}) = n^p(1 \pm O(n^{-\kappa_7})), \tag{68}$$

*where $\kappa_7$ is a positive constant.*

As a simple corollary, we can obtain the following cruder bounds on the trace and spectral norm of $\boldsymbol{A}_{-k}^{-1}$ and $\boldsymbol{A}_{-s}^{-1}$.

**Corollary B.5** (Trace and spectral norm of $\boldsymbol{A}_{-k}^{-1}$). *In the bi-level model, with probability at least $1 - 2e^{-n}$, we have*

$$\mathrm{Tr}(\boldsymbol{A}_U^{-1}) = n^{1-p}(1 \pm O(n^{-\kappa_7}))\sqrt{\log k} \tag{69}$$

$$\mathrm{Tr}(\boldsymbol{A}_{-k}^{-1}) = n^{1-p}(1 \pm O(n^{-\kappa_3}))\sqrt{\log k} \tag{70}$$

*and*

$$\max\left\{ \left\| \boldsymbol{A}_{-k}^{-1} \right\|_2, \left\| \boldsymbol{A}_U^{-1} \right\|_2 \right\} \leq c_2 n^{-p}, \tag{71}$$

*where $c_2$, $\kappa_7$, and $\kappa_3$ are all positive constants.*

*Proof.* We prove the claim for $\boldsymbol{A}_{-k}^{-1}$; the proof for $\boldsymbol{A}_U^{-1}$ is similar or easier because $\boldsymbol{A}_U^{-1}$ has a flat spectrum (Proposition B.4).

If $q + r < 1$, the upper bound for the spectral norm similarly follows. For the trace bounds, we can apply Proposition B.4, we have

$$\text{Tr}\big(\boldsymbol{A}_{-k}^{-1}\big) = (n - n^r + n^t)n^{-p}(1 \pm O(n^{-\kappa_7})) + (n^r - n^t) \cdot (1 + \mu^{-1})n^{-p}(1 \pm O(n^{-\kappa_9})) \quad (72)$$

$$= n^{1-p}(1 \pm O(n^{-\kappa_1})) \quad (73)$$

where

$$\kappa_1 = \min\{r - 1, 2 - q - 2r\} > 0,$$

as $q + 2r < 2(q + r) < 2$ by assumption.

On the other hand, the claim is obviously true when $q + r > 1$, as the entire spectrum of $\boldsymbol{A}_{-k}^{-1}$ is $(1 \pm O(n^{-\kappa_2}))n^{-p}$ with an appropriately defined positive constant $\kappa_2$. The spectral norm bound follows by defining $c_2$ to be any positive constant greater than 1 which absorbs the $o(1)$ deviation terms in the spectrum.

The proof concludes by setting $\kappa_3 = \min\{\kappa_1, \kappa_2\}$. $\qquad\square$

Finally, we have the following proposition which controls the spectrum of hat matrices such as $\boldsymbol{H}_k \triangleq \boldsymbol{W}_k^\top \boldsymbol{A}_{-k}^{-1} \boldsymbol{W}_k \in \mathbb{R}^{k \times k}$. The intuition is that even though the spectrum of $\boldsymbol{A}_{-k}^{-1}$ may be spiked, the spectrum of $\boldsymbol{W}_k^\top \boldsymbol{A}_{-k}^{-1} \boldsymbol{W}_k$ is ultimately flat because we are taking an extremely low dimensional projection which is unlikely to see significant contribution from the spiked portion of $\boldsymbol{A}_{-k}^{-1}$.

In fact, we can prove a more general statement, which will be useful for us in the proof. Let $\varnothing \neq T \subseteq S \subseteq [s]$; here $T$ and $S$ index nonempty subsets of the $s$ favored features. Then we can define $\boldsymbol{W}_T$ to be the matrix of weighted features in $T$ and the leave-$T$-out Gram matrix

$$\boldsymbol{A}_{-T} \triangleq \sum_{j \notin T} \lambda_j \boldsymbol{z}_j \boldsymbol{z}_j^\top. \quad (74)$$

Now define the $(T, S)$ hat matrix as $\boldsymbol{H}_{T,S} \triangleq \boldsymbol{W}_T^\top \boldsymbol{A}_{-S}^{-1} \boldsymbol{W}_T$. Evidently we have $\boldsymbol{H}_k = \boldsymbol{H}_{[k],[k]}$, so our notion is more general. The full proof is deferred to Appendix H.

**Proposition B.6** (Generalized hat matrices are flat). *Assume we are in the bi-level ensemble Definition 1. For any nonempty $T \subseteq S \subseteq [s]$, with probability at least $1 - 2e^{-\sqrt{n}} - 2e^{-n}$, we have all the eigenvalues tightly controlled:*

$$\mu_i((\boldsymbol{I}_{|T|} + \boldsymbol{H}_{T,S})^{-1}) = \min\{\mu, 1\}(1 \pm c_{T,S}n^{-\kappa_{11}}). \quad (75)$$

*where $c_{T,S}$ and $\kappa_{11}$ are positive constants that depend on $|T|$ and $|S|$.*

## C  Utility bounds: applying the tools

Wishart concentration allows us to tightly bound the hat matrix and pass to studying bilinear forms of the form $\boldsymbol{w}_i^\top \boldsymbol{A}_{-k}^{-1} \boldsymbol{\Delta} y$ rather than $\boldsymbol{w}_i^\top \boldsymbol{A}^{-1} \boldsymbol{\Delta} y$. Since $\boldsymbol{A}_{-k}^{-1}$ is independent of $\boldsymbol{W}_k$ and $\boldsymbol{\Delta} y$, we can condition on $\boldsymbol{A}_{-k}^{-1}$ and then apply Hanson-Wright (Theorem 4.1) to these bilinear forms for every realization of $\boldsymbol{A}_{-k}^{-1}$. In this section, we will explicitly calculate the scaling of the typical value of these bilinear forms using the bi-level ensemble scaling; these will prove to be useful throughout the rest of the paper.

We first state the following proposition which bounds the correlation between the relevant label-defining features and the label vectors; it is a combination of Propositions D.5 and D.6 in (Subramanian et al., 2022).

**Proposition C.1.** *For any distinct $\alpha, \beta \in [k]$, we have*

$$\frac{1}{\sqrt{\pi \ln 2}} \cdot \frac{n}{k} \cdot \sqrt{\ln k} \leq \mathbb{E}[\boldsymbol{z}_\alpha^\top \boldsymbol{y}_\alpha] \leq \sqrt{2} \cdot \frac{n}{k} \cdot \sqrt{\ln k} \quad (76)$$

*and*

$$-\sqrt{2} \cdot \frac{n}{k} \cdot \frac{1}{k-1} \cdot \sqrt{\ln k} \leq \mathbb{E}[\boldsymbol{z}_\alpha^\top \boldsymbol{y}_\beta] \leq -\frac{1}{\sqrt{\pi \ln 2}} \cdot \frac{n}{k} \cdot \frac{1}{k-1} \cdot \sqrt{\ln k} \quad (77)$$

With the above proposition in hand, we can prove the following lemma which gives concentration of the bilinear forms that we study.

**Lemma C.2.** *Let $i \in [d]$ and $\Delta y = \boldsymbol{y}_\alpha - \boldsymbol{y}_\beta$ where $\alpha, \beta \in [k]$ and $\beta \neq \alpha$. Let $\boldsymbol{M} \in \mathbb{R}^{n \times n}$ be a (random) matrix which is independent of $\boldsymbol{z}_i$ and $\Delta y$. Then conditioned on $\boldsymbol{M}$, with probability at least $1 - 1/nk$,*

$$\left| \boldsymbol{z}_i^\top \boldsymbol{M} \Delta y - \mathbb{E}[\boldsymbol{z}_i^\top \boldsymbol{M} \Delta y | \boldsymbol{M}] \right| \leq c_3 \sqrt{\frac{n}{k}} \|\boldsymbol{M}\|_2 \sqrt{\log(nk)},$$

*and the same holds with $\Delta y$ replaced with $\boldsymbol{y}_\alpha$. Here, $c_3$ is an appropriately chosen universal positive constant.*

*Moreover, we have*

(1) *For any distinct $\alpha, \beta \in [k]$, we have*

$$\mathbb{E}[\boldsymbol{z}_\alpha^\top \boldsymbol{M} \Delta y | \boldsymbol{M}] = c_7 \frac{\sqrt{\log k}}{k} \operatorname{tr}(\boldsymbol{M}) = -\mathbb{E}[\boldsymbol{z}_\beta^\top \boldsymbol{M} \Delta y] \tag{78}$$

$$\mathbb{E}[\boldsymbol{z}_\alpha^\top \boldsymbol{M} \boldsymbol{y}_\alpha | \boldsymbol{M}] = c_4 \frac{\sqrt{\log k}}{k} \operatorname{tr}(\boldsymbol{M}), \tag{79}$$

*where $c_7$ and $c_4$ are positive constants.*

(2) *For $i \in [d] \setminus \{\alpha, \beta\}$, we have*

$$\mathbb{E}[\boldsymbol{z}_i^\top \boldsymbol{M} \Delta y | \boldsymbol{M}] = 0. \tag{80}$$

(3) *For $i \in [d] \setminus \{\alpha\}$, we have*

$$\mathbb{E}[\boldsymbol{z}_i^\top \boldsymbol{M} \boldsymbol{y}_\alpha | \boldsymbol{M}] = -c_5 \frac{\sqrt{\log k}}{k(k-1)}, \tag{81}$$

*where $c_5$ is a positive constant.*

*Proof.* Let us check the conditions for our new variant of Hanson-Wright with soft sparsity (Theorem 4.1). We want to apply it to the random vectors $(\boldsymbol{z}_i, \Delta y) = (\boldsymbol{z}_i[j], \Delta y[j])_{j=1}^n$. Some of the hypotheses are immediate by definition. Evidently, $(\boldsymbol{z}_i[j], \Delta y[j])$ are independent across $j$, and are mean zero. Since $\boldsymbol{z}_i[j] \sim N(0,1)$, it is subgaussian with parameter at most $K = 2$. For the bounded and soft sparsity assumption, we clearly have $|\Delta y[j]| \leq 1$ and $\boldsymbol{y}_\alpha[j] \leq 1$ almost surely. Also, since $\Delta y[j]^2 \sim \operatorname{Ber}(\frac{2}{k})$, we have $\mathbb{E}[\Delta y[j]^2] = \frac{2}{k}$. Similarly, $\mathbb{E}[\boldsymbol{y}_\alpha[j]^2] = \frac{1}{k}(1 - \frac{1}{k})^2 + (1 - \frac{1}{k})\frac{1}{k^2} \leq \frac{2}{k}$.

The more complicated condition is the subgaussianity of $\boldsymbol{z}_i[j]$ conditioned on the value of $\Delta y[j]$ or $\boldsymbol{y}_\alpha[j]$. Regardless of whether we're conditioning on $\Delta y$ or $\boldsymbol{y}_\alpha$, it suffices to instead prove that $\boldsymbol{z}_i[j]$ is subgaussian conditioned on whether feature $i$ won the competition for datapoint $j$. First, suppose $i$ won, i.e. $\boldsymbol{y}_i^{\mathsf{oh}}[j] = 1$. Then the Borell-TIS inequality (Adler et al., 2007, Theorem 2.1.1) implies that $\boldsymbol{z}_i[j]$ satisfies a subgaussian tail inequality. By the equivalent conditions for subgaussianity Vershynin (2018, Proposition 2.5.2), it follows that $\boldsymbol{z}_i[j] - \mathbb{E}[\boldsymbol{z}_i[j]|\boldsymbol{y}_i^{\mathsf{oh}}[j] = 1]$ conditionally has subgaussian norm bounded by some absolute constant $K$. If $i$ doesn't win (or doesn't participate in the competition), then Proposition D.2 in Subramanian et al. (2022) implies that $\boldsymbol{z}_i[j] - \mathbb{E}[\boldsymbol{z}_i[j]|\boldsymbol{y}_i^{\mathsf{oh}}[j] = 0]$ conditionally has subgaussian norm bounded by 6.

Finally, since $\boldsymbol{M}$ is independent of $\boldsymbol{z}_i$ and $\Delta y$, we can condition on $\boldsymbol{M}$ and apply Theorem 4.1 to the bilinear form for every realization of $\boldsymbol{M}$.

Hence, we conclude that with probability at least $1 - 1/nk$ we have

$$\left| \boldsymbol{z}_i^\top \boldsymbol{M} \Delta y - \mathbb{E}[\boldsymbol{z}_i^\top \boldsymbol{M} \Delta y | \boldsymbol{M}] \right| \leq c_3 \sqrt{\frac{n}{k}} \|\boldsymbol{M}\|_2 \sqrt{\log(nk)}, \tag{82}$$

where $c_3$ is an appropriately chosen absolute constant based on $K$ and the constant $c$ defined in Theorem 4.1.

Now we can compute $\mathbb{E}[\boldsymbol{z}_i^\top \boldsymbol{M} \Delta y | \boldsymbol{M}]$ to prove the rest of the theorem. If $i = \alpha$, we have

$$\mathbb{E}[\boldsymbol{z}_\alpha^\top \boldsymbol{M} \Delta y | \boldsymbol{M}] = \operatorname{tr}\left( \boldsymbol{M} \mathbb{E}[\Delta y \boldsymbol{z}_\alpha^\top] \right).$$

Let us now compute $\mathbb{E}[\boldsymbol{\Delta} y \boldsymbol{z}_\alpha^\top]$. From Eq. (76) in Proposition C.1, we have $\mathbb{E}[\boldsymbol{y}_\alpha \boldsymbol{z}_\alpha^\top] = c_4 \frac{\sqrt{\log k}}{k} \boldsymbol{I}_n$, where $\frac{1}{\sqrt{\pi \log 2}} \le c_4 \le \sqrt{2}$. Similarly, we have $\mathbb{E}[\boldsymbol{y}_\beta \boldsymbol{z}_\alpha^\top] = -c_5 \frac{\sqrt{\log k}}{k(k-1)} \boldsymbol{I}_n$ where $\frac{1}{\sqrt{\pi \log 2}} \le c_5 \le \sqrt{2}$. It follows that $\mathbb{E}[\boldsymbol{\Delta} y \boldsymbol{z}_\alpha^\top] = \Theta\!\left( \frac{\sqrt{\log k}}{k} \right) \boldsymbol{I}_n$.

For $i \in [d] \setminus \{\alpha, \beta\}$, by symmetry we obtain $\mathbb{E}[\boldsymbol{y}_\alpha \boldsymbol{z}_i^\top] = \mathbb{E}[\boldsymbol{y}_\beta \boldsymbol{z}_i^\top]$. This implies $\mathbb{E}[\boldsymbol{\Delta} y \boldsymbol{z}_i^\top] = \mathbb{E}[\boldsymbol{y}_\alpha \boldsymbol{z}_i^\top] - \mathbb{E}[\boldsymbol{y}_\beta \boldsymbol{z}_i^\top] = 0$, so we obtain

$$\mathbb{E}[\boldsymbol{z}_i^\top \boldsymbol{M} \boldsymbol{\Delta} y | \boldsymbol{M}] = \mathrm{tr}\big( \boldsymbol{M} \mathbb{E}[\boldsymbol{\Delta} y \boldsymbol{z}_i^\top] \big) \tag{83}$$
$$= 0. \tag{84}$$

$\square$

Plugging in the bi-level scaling, we obtain the following corollary.

**Corollary C.3** (Asymptotic concentration of bilinear forms). *In the bi-level model, for any $i \in [k]$, we have with probability at least $1 - O(1/nk)$ that*

$$\big| \boldsymbol{z}_i^\top \boldsymbol{A}_{-k}^{-1} \boldsymbol{\Delta} y - \mathbb{E}[\boldsymbol{z}_i^\top \boldsymbol{A}_{-k}^{-1} \boldsymbol{\Delta} y] \big| \le c_6 n^{\frac{1-t}{2} - p} \sqrt{\log(nk)}.$$

*Moreover, we have*

(1) *For any distinct $\alpha, \beta \in [k]$,*
$$\mathbb{E}[\boldsymbol{z}_\alpha^\top \boldsymbol{A}_{-k}^{-1} \boldsymbol{\Delta} y] = c_7 n^{1-t-p} (1 \pm O(n^{-\kappa_3})) \sqrt{\log k} = -\mathbb{E}[\boldsymbol{z}_\beta^\top \boldsymbol{A}_{-k}^{-1} \boldsymbol{\Delta} y] \tag{85}$$

*The same statements hold (with different constants) if we replace $\boldsymbol{A}_{-k}^{-1}$ with $\boldsymbol{A}_{-s}^{-1}$.*

*Proof.* From Corollary B.5, we have $\big\| \boldsymbol{A}_{-k}^{-1} \big\|_2 \le c_6 n^{-p}$, where $c_6$ is an appropriately chosen universal positive constant based on $c_3$. Recall that $\boldsymbol{A}_{-k}$ is obtained by removing the $k$ label-defining features, so in particular $\boldsymbol{A}_{-k}^{-1}$ is independent of $(\boldsymbol{z}_i, \boldsymbol{\Delta} y)$ for $i \in [k]$. Hence, the conditions for Lemma C.2 are satisfied. Then applying the union bound for the spectral norm bound on $\boldsymbol{A}_{-k}^{-1}$, we see that with probability at least $1 - O(1/nk)$, the deviation term from Hanson-Wright is at most $c_6 n^{\frac{1}{2} - p} \sqrt{\log(nk)}$.

We now turn to calculating the asymptotic scalings for the expectations. From Lemma C.2, we know that $\mathbb{E}[\boldsymbol{z}_\alpha^\top \boldsymbol{A}_{-k}^{-1} \boldsymbol{\Delta} y | \boldsymbol{A}_{-k}^{-1}] = c_7 \frac{\sqrt{\log k}}{k} \mathrm{tr}\big( \boldsymbol{A}_{-k}^{-1} \big)$. Applying the high probability bound on $\mathrm{tr}\big( \boldsymbol{A}_{-k}^{-1} \big)$ from Corollary B.5, we obtain that with probability at least $1 - O(1/nk)$ that

$$-\mathbb{E}[\boldsymbol{z}_\beta^\top \boldsymbol{A}_{-k}^{-1} \boldsymbol{\Delta} y] = \mathbb{E}[\boldsymbol{z}_\alpha^\top \boldsymbol{A}_{-k}^{-1} \boldsymbol{\Delta} y] = c_7 n^{-t} n^{1-p} (1 \pm O(n^{-\kappa_3})) \sqrt{\log k} \tag{86}$$
$$= c_7 n^{1-t-p} (1 \pm O(n^{-\kappa_3})) \sqrt{\log k} \tag{87}$$

where in the second line we have applied Corollary B.5 and $c_7$ is an appropriately chosen positive constant. This proves (85). $\square$

With Corollary C.3 in hand, we are now in a position to do some straightforward calculations and bound some quantities which will pop up in the survival and contamination analysis.

**Proposition C.4** (Worst-case bound based on Hanson-Wright). *Let $T \subseteq [s]$ be a subset of favored features such that $\{\alpha, \beta\} \subseteq T$. Assume that $|T| = n^\tau$ for some $\tau \le r$. Then with probability at least $1 - O(1/nk)$, we have*

$$\big\| \boldsymbol{Z}_T^\top \boldsymbol{A}_{-T}^{-1} \boldsymbol{\Delta} y \big\|_2 \le c_8 (n^{1-t-p} + n^{\frac{1+\tau-t}{2} - p}) \sqrt{\log(nk|T|)}. \tag{88}$$

*Proof.* WLOG, suppose $\alpha = 1$ and $\beta = 2$. By Corollary C.3 we have with probability at least $1 - 1/n$ that

$$\big| \boldsymbol{Z}_T^\top \boldsymbol{A}_{-T}^{-1} \boldsymbol{\Delta} y \big| \le \begin{bmatrix} c_7 n^{1-t-p} \sqrt{\log k} \\ c_7 n^{1-t-p} \sqrt{\log k} \\ c_6 n^{\frac{1-t}{2} - p} \sqrt{\log(nk)} \\ \vdots \\ c_6 n^{\frac{1-t}{2} - p} \sqrt{\log(nk)} \end{bmatrix}. \tag{89}$$

Hence, the norm of this vector is at most

$$\left\|\boldsymbol{Z}_T^\top \boldsymbol{A}_{-T}^{-1} \Delta y\right\|_2 \le 2c_7 n^{1-t-p}\sqrt{\log k} + n^{\frac{\tau}{2}}c_6 n^{\frac{1-t}{2}-p}\sqrt{\log(nk)} \tag{90}$$

$$\le c_8(n^{1-t-p} + n^{\frac{1+\tau-t}{2}-p})\sqrt{\log(nk)}, \tag{91}$$

where $c_8$ is a positive constant.

$\square$

# D Bounding the survival

Recall that the relative survival was defined to be $\lambda_F \widehat{\boldsymbol{h}}_{\alpha,\beta}[\alpha] = \lambda_F \boldsymbol{z}_\alpha^\top \boldsymbol{A}^{-1} \Delta y$. The strategy is to apply our variant of Hanson-Wright to $\boldsymbol{z}_\alpha^\top \boldsymbol{A}^{-1} \Delta y$. Unfortunately, $\boldsymbol{A}^{-1}$ is not independent of $\boldsymbol{z}_\alpha$ or $\Delta y$, so we need to use Woodbury to extract out the independent portions and bound away the dependent portion. As we'll see shortly, the error from the dependent portions can also be controlled using Hanson-Wright. Let us now recall Proposition 4.2 for reference.

**Proposition 4.2** (Bounds on relative survival). *Suppose we are in the bi-level model. With probability at least $1 - O(1/n)$,*

$$\lambda_F \widehat{\boldsymbol{h}}_{\alpha,\beta}[\alpha] = \min\left\{\mu^{-1}, 1\right\}\Theta(n^{-\min\left\{t,\frac{1}{2}\right\}})\sqrt{\log k}.$$

*Proof.* Recall that $\widehat{\boldsymbol{h}}_{\alpha,\beta}[\alpha] = \boldsymbol{z}_\alpha^\top \boldsymbol{A}^{-1} \Delta y$. We first observe that for $i \in [k]$, the dependence between $\boldsymbol{A}^{-1}$ and $\boldsymbol{z}_i$ as well as $\boldsymbol{A}^{-1}$ and $\Delta y$ only comes through the $k$ label defining features. Hence, we can use the Woodbury identity to extract out the independent portions of $\boldsymbol{A}^{-1}$.

Indeed, our "push through" lemma for Woodbury (Lemma B.2) and concentration of the hat matrix (Proposition B.6) implies that with extremely high probability

$$\boldsymbol{Z}_k^\top \boldsymbol{A}^{-1} \Delta y = (\boldsymbol{I}_k + \boldsymbol{H}_k)^{-1} \boldsymbol{Z}_k^\top \boldsymbol{A}_{-k}^{-1} \Delta y \tag{92}$$

$$= \min\left\{\mu, 1\right\} (\boldsymbol{I}_k + \boldsymbol{E}) \boldsymbol{Z}_k^\top \boldsymbol{A}_{-k}^{-1} \Delta y, \tag{93}$$

where $\|\boldsymbol{E}\|_2 = O(n^{-\kappa_{11}})$.

Let $\boldsymbol{u}_\alpha \in \mathbb{R}^k$ denote the $\alpha$th row vector in $\boldsymbol{E}$, and let $\boldsymbol{u}_\alpha^- \in \mathbb{R}^{k-1}$ denote the subvector of $\boldsymbol{u}_\alpha$ without index $\alpha$. By reading off the $\alpha$th row of Eq. (93), we see that

$$\boldsymbol{z}_\alpha^\top \boldsymbol{A}^{-1} \Delta y = \min\left\{\mu, 1\right\} (\boldsymbol{z}_\alpha^\top \boldsymbol{A}_{-k}^{-1} \Delta y + \left\langle \boldsymbol{u}_\alpha, \boldsymbol{Z}_k^\top \boldsymbol{A}_{-k}^{-1} \Delta y \right\rangle) \tag{94}$$

Since $\|\boldsymbol{u}_\alpha\|_2 \le \|\boldsymbol{E}\|_2 = O(n^{-\kappa_{11}})$, it follows from Cauchy-Schwarz that

$$\left|\boldsymbol{z}_\alpha^\top \boldsymbol{A}^{-1} \Delta y - \min\left\{\mu, 1\right\}\boldsymbol{z}_\alpha^\top \boldsymbol{A}_{-k}^{-1} \Delta y\right| \le \min\left\{\mu, 1\right\}O(n^{-\kappa_{11}})\left\|\boldsymbol{Z}_k^\top \boldsymbol{A}_{-k}^{-1} \Delta y\right\|_2. \tag{95}$$

Let us pause for a moment and interpret Eq. (95). The term $\min\left\{\mu, 1\right\}$ is merely capturing the difference in behavior when regression works and fails; if regression works ($q + r < 1$) then it becomes $\mu$, and if regression fails ($q + r > 1$), then it becomes 1. This behavior should be expected: in the regression works case, we expect the effect of interpolation to be a *regularizing* one: the signals are attenuated by a factor of $\mu$. The RHS of Eq. (95) is an error term, capturing how differently $\boldsymbol{z}_\alpha^\top \boldsymbol{A}^{-1} \Delta y$ behaves from the expected behavior $\min\left\{\mu, 1\right\}\boldsymbol{z}_\alpha^\top \boldsymbol{A}_{-k}^{-1} \Delta y$.

Let us now bound the error term. From Proposition C.4 we have with probability at least $1 - O(1/nk)$ that

$$\left\|\boldsymbol{Z}_k^\top \boldsymbol{A}_{-k}^{-1} \Delta y\right\|_2 \le c_8(n^{1-t-p} + n^{\frac{1}{2}-p})\sqrt{\log(nk^2)}. \tag{96}$$

Let us do casework on $t$. For $t < \frac{1}{2}$, we have $\frac{1}{2} - p < 1 - t - p$, so we conclude that the error term is $\min\left\{\mu, 1\right\}O(n^{1-t-p} \cdot n^{-\kappa_4})\sqrt{\log(nk^2)}$, where $\kappa_4$ is a positive constant.

On the other hand, our Hanson-Wright calculations imply (Corollary C.3) that with probability at least $1 - O(1/nk)$ that

$$\left|\boldsymbol{z}_\alpha^\top \boldsymbol{A}_{-k}^{-1} \Delta y - c_7 n^{1-t-p}(1 \pm O(n^{-\kappa_3}))\sqrt{\log k}\right| \le c_6 n^{\frac{1}{2}-p}\sqrt{\log(nk)}.$$

Again, since $t < \frac{1}{2}$, the deviation term is $o(n^{1-t-p})\sqrt{\log k}$.

Hence we conclude that with probability $1 - O(1/nk)$ we have

$$\boldsymbol{z}_\alpha^\top \boldsymbol{A}^{-1} \boldsymbol{\Delta} y = c_7 \min\{\mu, 1\} n^{1-t-p} (1 \pm O(n^{-\kappa_5})) \sqrt{\log k},$$

where $\kappa_5$ is a positive constant.

Completely analogous logic handles the bounds for $\boldsymbol{z}_\beta^\top \boldsymbol{A}^{-1} \boldsymbol{\Delta} y$. Let us now return back to the quantity of interest, $\lambda_F \widehat{\boldsymbol{h}}_{\alpha,\beta}[\alpha]$. We can compute

$$
\begin{aligned}
\lambda_F \boldsymbol{z}_\alpha^\top \boldsymbol{A}^{-1} \boldsymbol{\Delta} y &= n^{p-q-r} \cdot c_7 \min\{\mu, 1\} n^{1-t-p} (1 \pm O(n^{-\kappa_5})) \sqrt{\log k} \\
&= c_7 \mu^{-1} \min\{\mu, 1\} n^{-t} (1 \pm O(n^{-\kappa_5})) \sqrt{\log k} \\
&= c_7 \min\{1, \mu^{-1}\} n^{-t} (1 \pm O(n^{-\kappa_5})) \sqrt{\log k}.
\end{aligned}
$$

On the other hand, if $t \geq \frac{1}{2}$ the error terms all dominate, and we replace $n^{1-t-p}$ with $n^{\frac{1}{2}-p}$ everywhere. We conclude that with probability at least $1 - O(1/nk)$,

$$\left| \boldsymbol{z}_\alpha^\top \boldsymbol{A}^{-1} \boldsymbol{\Delta} y \right| \leq c_9 \min\{\mu, 1\} n^{\frac{1}{2}-p} \sqrt{\log(nk)}, \tag{97}$$

where $c_9$ is a positive constant. Plugging in the scaling for $\lambda_F$ yields the desired result. $\qquad\square$

# E  Bounding the contamination

In this section we give a tight analysis of the contamination term. First, we rewrite the squared contamination term and separate it out into the contamination from the $k-2$ label-defining features which are not $\alpha$ or $\beta$, the rest of the $s-k$ favored features, and the remaining $d-s$ unfavored features. From Eq. (30), we have

$$\mathsf{CN}_{\alpha,\beta}^2 = \sum_{j \in [d] \backslash \{\alpha, \beta\}} \lambda_j^2 (\boldsymbol{z}_j^\top \boldsymbol{A}^{-1} \boldsymbol{\Delta} y)^2 \tag{98}$$

$$= \boldsymbol{\Delta} y^\top \boldsymbol{A}^{-1} \left( \sum_{j \in [d] \backslash \{\alpha, \beta\}} \lambda_j^2 \boldsymbol{z}_j \boldsymbol{z}_j^\top \right) \boldsymbol{A}^{-1} \boldsymbol{\Delta} y \tag{99}$$

$$= \underbrace{\boldsymbol{\Delta} y^\top \boldsymbol{A}^{-1} \left( \sum_{j \in [k] \backslash \{\alpha, \beta\}} \lambda_j^2 \boldsymbol{z}_j \boldsymbol{z}_j^\top \right) \boldsymbol{A}^{-1} \boldsymbol{\Delta} y}_{\triangleq \mathsf{CN}_{\alpha,\beta,L}^2} + \underbrace{\boldsymbol{\Delta} y^\top \boldsymbol{A}^{-1} \left( \sum_{j \in [s] \backslash [k]} \lambda_j^2 \boldsymbol{z}_j \boldsymbol{z}_j^\top \right) \boldsymbol{A}^{-1} \boldsymbol{\Delta} y}_{\triangleq \mathsf{CN}_{\alpha,\beta,F}^2}$$

$$\tag{100}$$

$$+ \underbrace{\boldsymbol{\Delta} y^\top \boldsymbol{A}^{-1} \left( \sum_{j > s} \lambda_j^2 \boldsymbol{z}_j \boldsymbol{z}_j^\top \right) \boldsymbol{A}^{-1} \boldsymbol{\Delta} y}_{\triangleq \mathsf{CN}_{\alpha,\beta,U}^2}. \tag{101}$$

Here, $\mathsf{CN}_{\alpha,\beta,L}$ corresponds to contamination from label defining features, $\mathsf{CN}_{\alpha,\beta,F}$ corresponds to contamination from favored features, and $\mathsf{CN}_{\alpha,\beta,U}$ corresponds to contamination from unfavored features. The reason for separating out the contamination into these three subterms is that we will need slightly different arguments to bound each of them, although Hanson-Wright and Woodbury are central to all of the arguments. In Appendix E.1 we prove the upper bound on $\mathsf{CN}_{\alpha,\beta,L} + \mathsf{CN}_{\alpha,\beta,F}$; in Appendix E.2 we prove the lower bound. Finally, in Appendix E.3 we bound $\mathsf{CN}_{\alpha,\beta,U}$. After putting these bounds together, we will obtain the main bounds on the contamination, which we restate here for reference.

**Proposition 4.3** (Bounds on contamination). *Suppose we are in the bi-level model. Then with probability at least $1 - O(1/n)$, the contamination satisfies*

$$\mathsf{CN}_{\alpha,\beta} = \underbrace{\min\{\mu^{-1}, 1\} \Theta(n^{\frac{r-t-1}{2}})}_{\textit{favored features}} + \underbrace{\Theta(n^{\frac{1-t-p}{2}})}_{\textit{unfavored features}}. \tag{26}$$

## E.1 Upper bounding the contamination from label-defining+favored features

In this section, we upper bound the contamination coming from the $s - 2$ favored features which are not $\alpha$ or $\beta$. This culminates in the following lemma.

**Lemma E.1.** *In the same setting as Proposition 4.3, we have with probability $1 - O(1/nk)$ that*

$$\mathsf{CN}^2_{\alpha,\beta,L} + \mathsf{CN}^2_{\alpha,\beta,F} \leq c_{12}^2 \min\left\{1, \mu^{-2}\right\} n^{r-t-1} \log(nsk)^2,$$

*where $c_{12}$ is a positive constant.*

*Proof.* Let $\boldsymbol{W}_R \in \mathbb{R}^{n \times (s-2)}$ as the weighted feature matrix which includes all of the $s - 2$ favored features aside from $\alpha, \beta$. We can then define $\boldsymbol{A}_{-R} \triangleq \boldsymbol{A} - \boldsymbol{W}_R \boldsymbol{W}_R^\top$ and $\boldsymbol{H}_R = \boldsymbol{W}_R^\top \boldsymbol{A}_{-R}^{-1} \boldsymbol{W}_R$. Using Woodbury, an analogous computation to Lemma B.2 implies that

$$\boldsymbol{W}_R \boldsymbol{A}^{-1} \boldsymbol{\Delta} y = (\boldsymbol{I}_{s-2} + \boldsymbol{H}_R)^{-1} \boldsymbol{W}_R \boldsymbol{A}_{-R}^{-1} \boldsymbol{\Delta} y. \tag{102}$$

The contamination from all of the $s - 2$ favored features that are not $\alpha$ or $\beta$ satisfies

$$\mathsf{CN}^2_{\alpha,\beta,L} + \mathsf{CN}^2_{\alpha,\beta,F} = \lambda_F \boldsymbol{\Delta} y^\top \boldsymbol{A}^{-1} \sum_{j \in [s] \setminus \{\alpha, \beta\}} \boldsymbol{w}_j \boldsymbol{w}_j^\top \boldsymbol{A}^{-1} \boldsymbol{\Delta} y$$

$$= \lambda_F \boldsymbol{\Delta} y^\top \boldsymbol{A}^{-1} \boldsymbol{W}_R \boldsymbol{W}_R^\top \boldsymbol{A}^{-1} \boldsymbol{\Delta} y$$

$$= \lambda_F \boldsymbol{\Delta} y^\top \boldsymbol{A}_{-R}^{-1} \boldsymbol{W}_R (\boldsymbol{I}_{s-2} + \boldsymbol{H}_R)^{-2} \boldsymbol{W}_R^\top \boldsymbol{A}_{-R}^{-1} \boldsymbol{\Delta} y.$$

Since Proposition B.6 implies that $\mu_1((\boldsymbol{I}_{s-2} + \boldsymbol{H}_R)^{-2}) \leq c_{10} \min\left\{\mu^2, 1\right\}$ with extremely high probability where $c_{10}$ is a positive constant, we know the contamination is with extremely high probability upper bounded by the following quadratic form:

$$c_{10} \min\left\{\mu^2, 1\right\} \lambda_F \boldsymbol{\Delta} y^\top \boldsymbol{A}_{-R}^{-1} \boldsymbol{W}_R \boldsymbol{W}_R^\top \boldsymbol{A}_{-R}^{-1} \boldsymbol{\Delta} y \tag{103}$$

$$= c_{10} \min\left\{\mu^2, 1\right\} \lambda_F^2 \sum_{j \in [s] \setminus \{\alpha, \beta\}} \left\langle \boldsymbol{z}_j, \boldsymbol{A}_{-R}^{-1} \boldsymbol{\Delta} y \right\rangle^2. \tag{104}$$

We still cannot apply Hanson-Wright, because $\boldsymbol{A}_{-R}^{-1}$ is not independent of $\boldsymbol{\Delta} y$. However, we can use Woodbury again to take out $\boldsymbol{z}_\alpha, \boldsymbol{z}_\beta$ from $\boldsymbol{A}_{-R}^{-1}$.

Define $\boldsymbol{W}_{\alpha,\beta} = [\boldsymbol{w}_\alpha \quad \boldsymbol{w}_\beta]$ and $\boldsymbol{H}_{\alpha,\beta}^{(s)} = \boldsymbol{W}_{\alpha,\beta}^\top \boldsymbol{A}_{-s}^{-1} \boldsymbol{W}_{\alpha,\beta}$. Then Woodbury implies that

$$\boldsymbol{A}_{-R}^{-1} = \boldsymbol{A}_{-s}^{-1} - \boldsymbol{A}_{-s}^{-1} \boldsymbol{W}_{\alpha,\beta} (\boldsymbol{I}_2 + \boldsymbol{H}_{\alpha,\beta}^{(s)})^{-1} \boldsymbol{W}_{\alpha,\beta}^\top \boldsymbol{A}_{-s}^{-1}. \tag{105}$$

Hence

$$\boldsymbol{z}_j^\top \boldsymbol{A}_{-R}^{-1} \boldsymbol{\Delta} y = \boldsymbol{z}_j^\top \boldsymbol{A}_{-s}^{-1} \boldsymbol{\Delta} y - \boldsymbol{z}_j^\top \boldsymbol{A}_{-s}^{-1} \boldsymbol{W}_{\alpha,\beta} (\boldsymbol{I}_2 + \boldsymbol{H}_{\alpha,\beta}^{(s)})^{-1} \boldsymbol{W}_{\alpha,\beta}^\top \boldsymbol{A}_{-s}^{-1} \boldsymbol{\Delta} y. \tag{106}$$

We will use Hanson-Wright and Cauchy-Schwarz to argue that the second term in Eq. (106) above will be dominated by the first term. Indeed, Corollary C.3 implies that $\boldsymbol{z}_j^\top \boldsymbol{A}_{-s}^{-1} \boldsymbol{\Delta} y \leq c_6 n^{\frac{1-t}{2} - p} \sqrt{\log(nsk)}$ with probability at least $1 - O(1/nsk)$, so it suffices to show that the other term is dominated by $n^{\frac{1-t}{2} - p}$. We will show that its contribution for each $j$ is $\min\left\{1, \mu^{-1}\right\} \tilde{O}(n^{\frac{1}{2} - t - p})$. By Cauchy-Schwarz, the magnitude of the second term of Eq. (106) is at most

$$\left\| (\boldsymbol{I}_2 + \boldsymbol{H}_{\alpha,\beta})^{-1} \right\|_2 \left\| \boldsymbol{W}_{\alpha,\beta}^\top \boldsymbol{A}_{-s}^{-1} \boldsymbol{z}_j \right\|_2 \left\| \boldsymbol{W}_{\alpha,\beta}^\top \boldsymbol{A}_{-s}^{-1} \boldsymbol{\Delta} y \right\|_2 \tag{107}$$

$$\leq c_{11} \lambda_F \min\{\mu, 1\} n^{\frac{1}{2} - p} \sqrt{\log(nsk)} \cdot n^{1-t-p} \sqrt{\log k} \tag{108}$$

$$\leq c_{11} \min\left\{1, \mu^{-1}\right\} n^{\frac{1}{2} - t - p} \log(nsk), \tag{109}$$

where $c_{11}$ is a positive constant. In the second line, we have used Proposition B.6 to upper bound $\left\| (\boldsymbol{I}_2 + \boldsymbol{H}_{\alpha,\beta}^{(s)})^{-1} \right\|_2 \leq O(\min\{\mu, 1\})$ and we have used Theorem B.1 and Proposition B.4 to deduce that that $\left| \boldsymbol{z}_j^\top \boldsymbol{A}_{-s}^{-1} \boldsymbol{z}_a \right| \leq O(n^{\frac{1}{2} - p}) \sqrt{\log(nsk)}$ with probability at least $1 - O(1/nsk)$. Similarly, we used the scaling from Corollary C.3 to deduce that $\left| \boldsymbol{z}_\alpha^\top \boldsymbol{A}_{-s}^{-1} \boldsymbol{\Delta} y \right| \leq O(n^{1-t-p}) \sqrt{\log k}$, and similarly for $\beta$.

Hence $z_j^\top A_{-R}^{-1} \Delta y$ is $O(n^{\frac{1-t}{2}-p}) \log(nsk)$ with probability $1 - O(1/nsk)$. By union bounding over $j$ and plugging our upper bound back into Eq. (104), we conclude that with probability at least $1 - O(1/nk)$

$$\mathsf{CN}_{\alpha,\beta,L}^2 + \mathsf{CN}_{\alpha,\beta,F}^2 \le c_{10} \min\left\{\mu^2, 1\right\} \lambda_F^2 n^r \cdot O(n^{1-t-2p}) \log(nsk)^2 \tag{110}$$

$$= \mu^{-2} \min\left\{\mu^2, 1\right\} O(n^{r-t-1}) \log(nsk)^2 \tag{111}$$

$$\le c_{12}^2 \min\left\{1, \mu^{-2}\right\} n^{r-t-1} \log(nsk)^2, \tag{112}$$

where $c_{12}$ is a positive constant, concluding the proof. $\qquad\square$

### E.2  Lower bounding the contamination from label-defining+favored features

In this section, we upper bound the contamination coming from the $s - 2$ favored features which are not $\alpha$ or $\beta$. This culminates in the following lemma.

**Lemma E.2.** *In the same setting as Proposition 4.3, if $t > 0$, with probability at least $1 - O(1/nk)$, we have*

$$\mathsf{CN}_{\alpha,\beta,L}^2 + \mathsf{CN}_{\alpha,\beta,F}^2 \ge c_{14}^2 \min\left\{1, \mu^{-2}\right\} n^{r-t-1},$$

*where $c_{14}$ is a positive constant.*

*Proof.* Following the beginning of the proof of Lemma E.1 and what we know about the flatness of the spectra of hat matrices from Proposition B.6, we can deduce that there is some positive constant $c_{13}$ such that with extremely high probability

$$\mathsf{CN}_{\alpha,\beta,L}^2 + \mathsf{CN}_{\alpha,\beta,F}^2 \ge c_{13} \min\left\{\mu^2, 1\right\} \lambda_F^2 \sum_{j \in [s]\setminus\{\alpha,\beta\}} \left\langle z_j, A_{-R}^{-1} \Delta y \right\rangle^2. \tag{113}$$

We will further lower bound this by throwing out all label-defining $j$. In other words, the goal now is to lower bound

$$\sum_{j \in [s]\setminus[k]} \left\langle z_j, A_{-R}^{-1} \Delta y \right\rangle^2 = \left\langle Z_F^\top A_{-R}^{-1} \Delta y, Z_F^\top A_{-R}^{-1} \Delta y \right\rangle. \tag{114}$$

The main idea is to use Bernstein's inequality, but unfortunately $A_{-R}^{-1}$ is not independent of $\Delta y$, so we will again resort to Woodbury to take out $z_\alpha$ and $z_\beta$. As in the proof for upper bounding the favored contamination, we have $W_{\alpha,\beta} = [w_\alpha \quad w_\beta]$ and $H_{\alpha,\beta}^{(s)} = W_{\alpha,\beta}^\top A_{-s}^{-1} W_{\alpha,\beta}$. Then we can deduce from another application of Woodbury that

$$z_j^\top A_{-R}^{-1} \Delta y = z_j^\top A_{-s}^{-1} \Delta y - z_j^\top A_{-s}^{-1} W_{\alpha,\beta}(I_2 + H_{\alpha,\beta})^{-1} W_{\alpha,\beta}^\top A_{-s}^{-1} \Delta y. \tag{115}$$

Again, we can argue that with probability $1 - O(1/nsk)$, the second term is upper bounded in magnitude by

$$\min\left\{1, \mu^{-1}\right\} n^{\frac{1}{2}-t-p} \log(nsk) = \min\left\{1, \mu^{-1}\right\} O(n^{\frac{1-t}{2}-p} \cdot n^{-\kappa_6}), \tag{116}$$

where $\kappa_6$ is a positive constant because $t > 0$. Since Hanson-Wright (Corollary C.3) implies that $z_j^\top A_{-s}^{-1} \Delta y = \tilde{O}(n^{\frac{1-t}{2}-p})$, this implies that $z_j^\top A_{-R}^{-1} \Delta y = \tilde{O}(n^{\frac{1-t}{2}-p})$, and similarly for $\beta$. Hence we have

$$(z_j^\top A_{-R}^{-1} \Delta y)^2 = (z_j^\top A_{-s}^{-1} \Delta y + \min\left\{1, \mu^{-1}\right\} O(n^{\frac{1-t}{2}-p-\kappa_6}))^2 \tag{117}$$

$$= (z_j^\top A_{-s}^{-1} \Delta y)^2 + \min\left\{1, \mu^{-1}\right\} O(n^{\frac{1-t}{2}-p-\kappa_6}) \tilde{O}(n^{\frac{1-t}{2}-p}) \tag{118}$$

$$= (z_j^\top A_{-s}^{-1} \Delta y)^2 + \min\left\{1, \mu^{-1}\right\} o(n^{1-t-2p}). \tag{119}$$

We are now in a position to analyze the contribution from the first term of Eq. (119) to Eq. (114): its contribution is $\left\langle Z_F^\top A_{-s}^{-1} \Delta y, Z_F^\top A_{-s}^{-1} \Delta y \right\rangle$. This does have all the independence required to apply Bernstein, because $(A_{-s}^{-1}, \Delta y)$ are independent of $Z_F$. Hence conditioned on $A_{-s}^{-1}$ and $\Delta y$, $\left\langle Z_F^\top A_{-s}^{-1} \Delta y, Z_F^\top A_{-s}^{-1} \Delta y \right\rangle$ is a sum of $s - k$ subexponential variables, and by Lemma 2.7.7 of Vershynin (2018) each of these random variables conditionally has subexponential norm at most $\left\|A_{-s}^{-1} \Delta y\right\|_2^2$ and conditional mean $\left\langle A_{-s}^{-1} \Delta y, A_{-s}^{-1} \Delta y \right\rangle$.

We can use Hanson-Wright (Theorem 4.1) to bound both of these quantities. Indeed, it implies that with probability at least $1 - O(1/nk)$,

$$\left\| \boldsymbol{A}_{-s}^{-1} \boldsymbol{\Delta} y \right\|_2^2 \leq O(n^{1-t-2p}). \tag{120}$$

Let us now compute the Hanson-Wright bound for $\left\langle \boldsymbol{A}_{-s}^{-1} \boldsymbol{\Delta} y, \boldsymbol{A}_{-s}^{-1} \boldsymbol{\Delta} y \right\rangle$. Note that $\boldsymbol{A}_{-s}^{-1}$ is independent of $\boldsymbol{\Delta} y$, so we can condition on $\boldsymbol{A}_{-s}^{-1}$ and conclude that with probability at least $1 - O(1/nk)$

$$\left\langle \boldsymbol{A}_{-s}^{-1} \boldsymbol{\Delta} y, \boldsymbol{A}_{-s}^{-1} \boldsymbol{\Delta} y \right\rangle \geq \mathbb{E}[\left\langle \boldsymbol{A}_{-s}^{-1} \boldsymbol{\Delta} y, \boldsymbol{A}_{-s}^{-1} \boldsymbol{\Delta} y \right\rangle | \boldsymbol{A}_{-s}^{-1}] - O(n^{\frac{1-t}{2}}) \left\| \boldsymbol{A}_{-s}^{-2} \right\|_2 \sqrt{\log(nk)} \tag{121}$$

$$= \mathrm{Tr}\left( \boldsymbol{A}_{-s}^{-2} \mathbb{E}[\boldsymbol{\Delta} y \boldsymbol{\Delta} y^\top] \right) - O(n^{\frac{1-t}{2}}) \left\| \boldsymbol{A}_{-s}^{-2} \right\|_2 \sqrt{\log(nk)} \tag{122}$$

$$= \frac{2}{k} \mathrm{Tr}\left( \boldsymbol{A}_{-s}^{-2} \right) - O(n^{\frac{1-t}{2}}) \left\| \boldsymbol{A}_{-s}^{-2} \right\|_2 \sqrt{\log(nk)}, \tag{123}$$

where we have used the fact that $\boldsymbol{\Delta} y$ is mean zero and $\boldsymbol{\Delta} y[i]^2 \sim \mathsf{Ber}(\frac{2}{k})$.

From Proposition B.4, we obtain the scaling for $\mathrm{Tr}\left( \boldsymbol{A}_{-s}^{-2} \right)$ and $\left\| \boldsymbol{A}_{-s}^{-2} \right\|_2$. This implies that with probability at least $1 - O(1/nk)$

$$\left\langle \boldsymbol{A}_{-s}^{-1} \boldsymbol{\Delta} y, \boldsymbol{A}_{-s}^{-1} \boldsymbol{\Delta} y \right\rangle \geq \Omega(n^{1-t-2p}) - O(n^{\frac{1-t}{2}-2p}) \sqrt{\log(nk)} \tag{124}$$

$$\geq \Omega(n^{1-t-2p}). \tag{125}$$

as $t < 1$.

Bernstein and the union bound implies that with probability at least $1 - O(1/nk)$,

$$\left\langle \boldsymbol{Z}_F^\top \boldsymbol{A}_{-s}^{-1} \boldsymbol{\Delta} y, \boldsymbol{Z}_F^\top \boldsymbol{A}_{-s}^{-1} \boldsymbol{\Delta} y \right\rangle \geq \left( \sum_{j \in [s] \setminus [k]} \Omega(n^{1-t-2p}) \right) - O(n^{\frac{r}{2}+1-t-2p}) \tag{126}$$

$$\geq \Omega(n^{r+1-t-2p}), \tag{127}$$

as $r > 0$.

To wrap up, we will need to upper bound the contribution of the error term in Eq. (119). Its contribution from summing over $j \in [s] \setminus [k]$ is $\min\left\{1, \mu^{-2}\right\} o(n^{r+1-t-2p})$, which is negligible compared to the Bernstein term, which as we just proved is $\Omega(n^{r+1-t-2p})$. Hence $\left\langle \boldsymbol{Z}_F^\top \boldsymbol{A}_{-R}^{-1} \boldsymbol{\Delta} y, \boldsymbol{Z}_F^\top \boldsymbol{A}_{-R}^{-1} \boldsymbol{\Delta} y \right\rangle \geq \Omega(n^{r+1-t-2p})$ with high probability, and inserting this back into our lower bound Eq. (113), we see that

$$\mathsf{CN}_{\alpha,\beta,L}^2 + \mathsf{CN}_{\alpha,\beta,F}^2 \geq c_{13} \min\left\{\mu^2, 1\right\} \lambda_F^2 \Omega(n^{r+1-t-2p}) \tag{128}$$

$$= c_{13} \mu^{-2} \min\left\{\mu^2, 1\right\} \Omega(n^{r-t-1}) \tag{129}$$

$$\geq c_{14}^2 \min\left\{1, \mu^{-2}\right\} n^{r-t-1}, \tag{130}$$

where $c_{14}$ is a positive constant. $\qquad \square$

### E.3 Bounding the unfavored contamination

Finally, we wrap up the section by proving matching upper and lower bounds for the unfavored contamination $\mathsf{CN}_{\alpha,\beta,U}$.

**Lemma E.3** (Bounding unfavored contamination)**.** *In the same setting as Proposition 4.3, if $t > 0$, with probability $1 - O(1/nk)$, the contamination from the unfavored features satisfies*

$$\mathsf{CN}_{\alpha,\beta,U}^2 = c_{15}^2 (1 \pm o(1)) n^{1-t-p},$$

*where $c_{15}$ is a positive constant.*

*On the other hand, if $t = 0$, then with probability $1 - O(1/nk)$, the unfavored contamination satisfies*

$$\mathsf{CN}_{\alpha,\beta,U}^2 \leq c_{16}^2 \min\left\{1, \mu^{-1}\right\} n^{1-t-p} \log(nsk).$$

*Proof.* By Woodbury, we have

$$A^{-1} = A_U^{-1} - A_U^{-1} M_s A_U^{-1}, \tag{131}$$

where

$$M_s \triangleq W_s (I_s + H_s)^{-1} W_s^\top, \tag{132}$$

and $H_s \triangleq W_s^\top A_{-s}^{-1} W_s$.

Now we have

$$\mathsf{CN}_{\alpha,\beta,U}^2 = \Delta y^\top A^{-1} A_U A^{-1} \Delta y \tag{133}$$

$$= \Delta y^\top (A_U^{-1} - A_U^{-1} M_s A_U^{-1}) A_U (A_U^{-1} - A_U^{-1} M_s A_U^{-1}) \Delta y \tag{134}$$

$$= \Delta y^\top (A_U^{-1} - 2 A_U^{-1} M_s A_U^{-1} + A_U^{-1} M_s A_U^{-1} M_s A_U^{-1}) \Delta y. \tag{135}$$

By Theorem 4.1, we have with probability at least $1 - O(1/nk)$

$$\Delta y^\top A_U^{-1} \Delta y = c_{15}^2 (1 \pm o(1)) n^{1-t-p}, \tag{136}$$

where $c_{15}$ is a positive constant.

On the other hand, we have that

$$M_s A_U^{-1} M_s = W_s (I_s + H_s)^{-1} W_s^\top A_U^{-1} W_s (I_s + H_s)^{-1} W_s^\top$$

$$= W_s (I_s + H_s)^{-1} H_s (I_s + H_s)^{-1} W_s^\top$$

$$= W_s (I_s - (I_s + H_s)^{-1}) (I_s + H_s)^{-1} W_s^\top$$

$$= W_s ((I_s + H_s)^{-1} - (I_s + H_s)^{-2}) W_s^\top$$

Due to Proposition B.6, $\mu_i((I_s + H_s)^{-1}) = \min\{\mu, 1\}(1 \pm o(1))$ for all $i$ with very high probability. Hence to handle the error terms that are not $\Delta y^\top A_U^{-1} \Delta y$, it suffices to asymptotically bound $\Delta y^\top A_U^{-1} M_s A_U^{-1} \Delta y$. In turn, we can couple this to the quadratic form $\min\{\mu, 1\}(1 \pm o(1)) \Delta y^\top A_U^{-1} W_s W_s A_U^{-1} \Delta y$. By Proposition C.4, we have with probability at least $1 - O(1/nk)$

$$\min\{\mu, 1\}(1 \pm o(1)) \Delta y^\top A_U^{-1} W_s W_s A_U^{-1} \Delta y \tag{137}$$

$$\leq c_8^2 \lambda_F \min\{\mu, 1\}(1 \pm o(1))(n^{2-2t-2p} + n^{r+1-t-2p}) \log(nsk) \tag{138}$$

$$\leq c_8^2 \min\{1, \mu^{-1}\}(1 \pm o(1))(n^{1-2t-p} + n^{r-t-p}) \log(nsk) \tag{139}$$

For $t > 0$, we claim that the term in Eq. (139) is $o(n^{1-t-p})$, because $1 - 2t - p < 1 - t - p$ and $r - t - p < 1 - t - p$. Hence if $t > 0$ then by union bound we have with probability at least $1 - O(1/nk)$ that

$$\mathsf{CN}_{\alpha,\beta,U}^2 = c_{15}^2 (1 \pm o(1)) n^{1-t-p}, \tag{140}$$

as desired.

On the other hand, if $t = 0$, we only have an issue if $q + r < 1$, so that $\min\{1, \mu^{-1}\} = 1$. In this case, the deviation term $n^{1-2t-p} = n^{1-t-p}$. However, this won't affect the fact that the upper bound on contamination will still be $\tilde{O}(n^{1-t-p})$. More precisely, this bound concludes by arguing that

$$\mathsf{CN}_{\alpha,\beta,U}^2 \leq c_{16}^2 \min\{1, \mu^{-1}\} n^{1-t-p} \log(nsk), \tag{141}$$

where $c_{16}$ is an appropriately defined positive constant.

It turns out we don't have to worry about this edge case at all for the lower bound on $\mathsf{CN}_{\alpha,\beta,U}$, because the stated conditions for misclassification imply that $t > 0$ anyway. This completes the proof of the lemma. $\qquad\square$

# F    Obtaining tight misclassification rate

In this section, we will prove Proposition A.5. Let us restate the main proposition and sketch out its proof more formally.

**Proposition A.5** (Correlation bound). *Assume we are in the bi-level ensemble model (Definition 1), the true data generating process is 1-sparse (Assumption 1), and the number of classes scales with $n$ (i.e. $t > 0$). Then for every $\epsilon > 0$, we have*

$$\Pr\left[\max_{\beta \in [k], \beta \neq \alpha} Z^{(\beta)} > n^{-u}\right] \geq 1 - \Theta\left(\frac{1}{k^{1+o(1)}}\right) - \epsilon \tag{49}$$

*for sufficiently large $n$ and any $u > 0$.*

*Proof sketch.* Note that the $Z^{(\beta)}$'s that must outcompete the decaying survival to contamination ratio are jointly Gaussian, as they are projections of a standard Gaussian vector $\boldsymbol{x}_{\text{test}} \in \mathbb{R}^d$. Hence if we want to study the probability that $\max_\beta Z^{(\beta)}$ outcompetes $n^{-u}$, we have to understand the correlation structure of the $Z^{(\beta)}$'s.

We will argue that for $\beta, \gamma \in [k]$ with $\alpha, \beta, \gamma$ pairwise distinct, the correlation between $Z^{(\beta)}$ and $Z^{(\gamma)}$ is $\frac{1}{2} \pm o(1)$ with high probability. To that end, we want to look at the correlation (inner product) between the vectors $\left\{\lambda_j \widehat{\boldsymbol{h}}_{\alpha,\beta}[j]\right\}$ for $j \notin \{\alpha, \beta\}$ and $\left\{\lambda_j \widehat{\boldsymbol{h}}_{\alpha,\gamma}[j]\right\}$ for $j \notin \{\alpha, \gamma\}$. However, note that by independence of the components of $\boldsymbol{x}_{\text{test}}$ from every other random variable and the fact that they are mean zero, we have

$$\mathbb{E}[\widehat{\boldsymbol{h}}_{\alpha,\beta}[\gamma]\boldsymbol{x}_{\text{test}}[\gamma]\widehat{\boldsymbol{h}}_{\alpha,\gamma}[\beta]\boldsymbol{x}_{\text{test}}[\beta]] = 0.$$

Hence it suffices to look at the correlation for $j \notin \{\alpha, \beta, \gamma\}$.

We assume WLOG that $\alpha = 1, \beta = 2, \gamma = 3$. Let

$$\Lambda_{\alpha,\beta} \triangleq \text{diag}(1 - \mathbf{1}_{j=\alpha} - \mathbf{1}_{j=\beta})_{j \in [d]} \circ \text{diag}(\lambda_j)_{j \in [d]} \in \mathbb{R}^{d \times d}$$

represent the diagonal matrices containing the squared feature weights with indices $\alpha, \beta$ zeroed out. Next, let $\boldsymbol{v}_{\alpha,\beta} \in \mathbb{R}^d$ denote the vector with $\boldsymbol{v}_{\alpha,\beta}[\alpha] = \boldsymbol{v}_{\alpha,\beta}[\beta] = 0$ and $\boldsymbol{v}_{\alpha,\beta}[j] = \lambda_j \widehat{\boldsymbol{h}}_{\alpha,\beta}[j]$ for $j \in [d], j \notin \{\alpha, \beta\}$. Hence $\boldsymbol{v}_{\alpha,\beta} = \Lambda_{\alpha,\beta}^{1/2}(\widehat{\boldsymbol{f}}_\alpha - \widehat{\boldsymbol{f}}_\beta)$. Since $Z^{(\beta)} = \langle \boldsymbol{v}_{\alpha,\beta}, \boldsymbol{x}_{\text{test}} \rangle$, in order to analyze the correlations between $Z^{(\beta)}$ and $Z^{(\gamma)}$, it suffices to analyze $\boldsymbol{v}_{\alpha,\beta}$. Indeed, we will show that the weighted halfspaces $\Lambda_{\alpha,\beta}^{1/2}\widehat{\boldsymbol{f}}_\alpha \in \mathbb{R}^d$ and $\Lambda_{\alpha,\beta}^{1/2}\widehat{\boldsymbol{f}}_\beta \in \mathbb{R}^d$ are asymptotically orthogonal.

In other words, we need to show that

$$\frac{\left\langle \Lambda_{\alpha,\beta}^{1/2}\widehat{\boldsymbol{f}}_\alpha, \Lambda_{\alpha,\beta}^{1/2}\widehat{\boldsymbol{f}}_\beta \right\rangle}{\left\|\Lambda_{\alpha,\beta}^{1/2}\widehat{\boldsymbol{f}}_\alpha\right\|_2 \left\|\Lambda_{\alpha,\beta}^{1/2}\widehat{\boldsymbol{f}}_\beta\right\|_2} = o(1)$$

with probability at least $1 - O(1/nk)$; we can then union bound against all choices of $\beta$. This is the most technically involved part of the proof, and is the content of Proposition F.1.

This in turn will imply (see Lemma F.2) that the maximum (and minimum) correlation between the $\boldsymbol{v}_{\alpha,\beta}$ for different $\beta$ is $\frac{1}{2} \pm o(1)$. Let $(\overline{Z}_\beta)_{\beta \in [k], \beta \neq \alpha}$ be equicorrelated gaussians with correlation $\overline{\rho} = \frac{1}{2} + o(1)$, and $(\underline{Z}_\beta)_{\beta \in [k], \beta \neq \alpha}$ be equicorrelated gaussians with correlation $\underline{\rho} = \frac{1}{2} - o(1)$. By Slepian's lemma, for any $u > 0$, the probability of $\max_\beta Z^{(\beta)}$ losing to $n^{-u}$ is sandwiched as

$$\Pr\left[\max_\beta \underline{Z}_\beta \leq n^{-u}\right] \leq \Pr\left[\max_\beta Z^{(\beta)} \leq n^{-u}\right] \leq \Pr\left[\max_\beta \overline{Z}_\beta \leq n^{-u}\right],$$

where we have adopted the shorthand $\max_\beta$ to denote $\max_{\beta \in [k], \beta \neq \alpha}$.

Theorem 2.1 of Pinasco et al. (2021) shows that jointly gaussian vectors in $\mathbb{R}^k$ with equicorrelation $\rho$ lie in the positive orthant with probability $\Theta(k^{1-1/\rho})$. In particular, applied to $\overline{Z}_\beta$, with correlation $\overline{\rho} = \frac{1}{2} + o(1)$, we find that

$$\Pr\left[\max_\beta \overline{Z}_\beta \leq 0\right] = \Theta(k^{-1+o(1)}),$$

and similarly for $\underline{Z}_\beta$. Anticoncentration for Gaussian maxima (Chernozhukov et al., 2015, Corollary 1) implies that we can transfer over the bound on $\Pr\left[\max_\beta \overline{Z}_\beta \le 0\right]$ to a bound on $\Pr\left[\max_\beta \overline{Z}_\beta \le n^{-u}\right]$ to show that for every $\epsilon > 0$, we have

$$\Theta(k^{-1-o(1)}) - \epsilon \le \Pr\left[\max_\beta Z^{(\beta)} \le n^{-u}\right] \le \Theta(k^{-1+o(1)}) + \epsilon \tag{142}$$

for sufficiently large $n$. Taking the complement of the above event concludes the proof. $\qquad\square$

## F.1 Main results for tight misclassification rates

The main result in this section is the following proposition, which states that the halfspace predictions are asymptotically orthogonal. Its proof is deferred to the subsequent sections.

**Proposition F.1.** *Assume we are in the bi-level ensemble model (Definition 1), the true data generating process is 1-sparse (Assumption 1), and the number of classes scales with $n$ (i.e. $t > 0$).*

*For any distinct $\alpha, \beta \in [k]$, with probability at least $1 - O(1/nk)$, we have*

$$\frac{\left\langle \Lambda_{\alpha,\beta}^{1/2} \widehat{\boldsymbol{f}}_\alpha, \Lambda_{\alpha,\beta}^{1/2} \widehat{\boldsymbol{f}}_\beta \right\rangle}{\left\| \Lambda_{\alpha,\beta}^{1/2} \widehat{\boldsymbol{f}}_\alpha \right\|_2 \left\| \Lambda_{\alpha,\beta}^{1/2} \widehat{\boldsymbol{f}}_\beta \right\|_2} = o(1).$$

Given Proposition F.1, we can show that the $Z^{(\beta)}$ have correlations that approach $\frac{1}{2}$. The intuitive reason that this correleation approaches $\frac{1}{2}$ is that the contribution from $\alpha$ is common. The following lemma formalizes this intuition.

**Lemma F.2** (Correlation of relative differences of almost orthogonal vectors)**.** *Suppose that we have $n$ unit vectors $\boldsymbol{x}_1, \ldots, \boldsymbol{x}_n \in \mathbb{R}^d$ such that $|\langle \boldsymbol{x}_i, \boldsymbol{x}_j \rangle| \le \gamma$ for $\gamma > 0$. Then for any distinct $i, j, k \in [n]$, we have*

$$\left| \frac{\langle \boldsymbol{x}_j - \boldsymbol{x}_i, \boldsymbol{x}_i - \boldsymbol{x}_k \rangle}{\|\boldsymbol{x}_j - \boldsymbol{x}_i\| \|\boldsymbol{x}_i - \boldsymbol{x}_k\|} - \frac{1}{2} \right| \le \frac{2\gamma}{1 - \gamma}.$$

*Proof.* For any $i \ne j$, we have $\|\boldsymbol{x}_i - \boldsymbol{x}_j\|^2 = 2 - 2\langle \boldsymbol{x}_i, \boldsymbol{x}_j \rangle$. Hence we have

$$2 - 2\gamma \le \|\boldsymbol{x}_i - \boldsymbol{x}_j\|^2 \le 2 + 2\gamma.$$

Also

$$\begin{aligned}
2 - 2\gamma &\le \|\boldsymbol{x}_j - \boldsymbol{x}_k\|^2 \\
&= \|\boldsymbol{x}_i - \boldsymbol{x}_j\|^2 + \|\boldsymbol{x}_i - \boldsymbol{x}_k\|^2 - 2\langle \boldsymbol{x}_j - \boldsymbol{x}_i, \boldsymbol{x}_i - \boldsymbol{x}_k \rangle \\
&\le 4 + 4\gamma - 2\langle \boldsymbol{x}_j - \boldsymbol{x}_i, \boldsymbol{x}_i - \boldsymbol{x}_k \rangle.
\end{aligned}$$

Since $\|\boldsymbol{x}_i - \boldsymbol{x}_j\| \ge \sqrt{2 - 2\gamma}$, we can rearrange and obtain that

$$\frac{\langle \boldsymbol{x}_j - \boldsymbol{x}_i, \boldsymbol{x}_i - \boldsymbol{x}_k \rangle}{\|\boldsymbol{x}_j - \boldsymbol{x}_i\| \|\boldsymbol{x}_i - \boldsymbol{x}_k\|} \le \frac{1 + 3\gamma}{2 - 2\gamma}.$$

Similarly we can reverse the inequalities and get

$$2 + 2\gamma \ge 4 - 4\gamma - 2\langle \boldsymbol{x}_j - \boldsymbol{x}_i, \boldsymbol{x}_i - \boldsymbol{x}_k \rangle,$$

so

$$\frac{\langle \boldsymbol{x}_j - \boldsymbol{x}_i, \boldsymbol{x}_i - \boldsymbol{x}_k \rangle}{\|\boldsymbol{x}_j - \boldsymbol{x}_i\| \|\boldsymbol{x}_i - \boldsymbol{x}_k\|} \ge \frac{1 - 3\gamma}{2 + 2\gamma}.$$

$\qquad\square$

Combining Proposition F.1 with Lemma F.2 yields the following formal statement about the correlations between the $Z^{(\beta)}$.

**Lemma F.3** (Asymptotic correlation of relative survivals). *For any distinct $\alpha, \beta, \beta' \in [k]$, under the same assumptions as Proposition F.1, as $n \to \infty$, with probability at least $1 - O(1/n)$, we have*

$$\left| \mathbb{E}[Z^{(\beta)} Z^{(\beta')}] - \frac{1}{2} \right| \leq o(1).$$

*As a consequence, the asymptotic correlation between the relative survivals approaches $\frac{1}{2}$ at a polynomial rate.*

*Proof.* Plugging in the result of Proposition F.1 into Lemma F.2, we obtain the stated result. □

## F.2 Lower bounding the denominator

Let us now begin to prove Proposition F.1. The first step is to bound the denominator of the normalized correlation. Writing out the definitions, we have

$$\left\| \Lambda_{\alpha,\beta}^{1/2} \widehat{\boldsymbol{f}}_\alpha \right\|^2 = \sum_{j \notin \{\alpha,\beta\}} \lambda_j^2 \boldsymbol{y}_\alpha^\top \boldsymbol{A}^{-1} \boldsymbol{z}_j \boldsymbol{z}_j^\top \boldsymbol{A}^{-1} \boldsymbol{y}_\alpha$$

$$= \lambda_F^2 \sum_{j \notin \{\alpha,\beta\}, j \in [s]} \boldsymbol{y}_\alpha^\top \boldsymbol{A}^{-1} \boldsymbol{z}_j \boldsymbol{z}_j^\top \boldsymbol{A}^{-1} \boldsymbol{y}_\alpha + \lambda_U^2 \sum_{j > s} \boldsymbol{y}_\alpha^\top \boldsymbol{A}^{-1} \boldsymbol{z}_j \boldsymbol{z}_j^\top \boldsymbol{A}^{-1} \boldsymbol{y}_\alpha$$

Note that these two terms are respectively analogous to $\mathsf{CN}_{\alpha,\beta,L}^2 + \mathsf{CN}_{\alpha,\beta,F}^2$ and $\mathsf{CN}_{\alpha,\beta,U}^2$. In fact, the proofs of the lower bounds for contamination essentially transfer over verbatim to the lower bounds on the denominator, because Hanson-Wright implies that we can show that $\left\| \boldsymbol{A}_{-s}^{-1} \boldsymbol{y}_\alpha \right\|_2$ concentrates the same way that $\left\| \boldsymbol{A}_{-s}^{-1} \Delta y \right\|_2$ does. In essence, we are able to show the following proposition.

**Proposition F.4** (Lower bound on norm of scaled halfspaces). *Under the same assumptions as Proposition F.1, for any $\alpha, \beta \in [k]$, with $\alpha \neq \beta$, with probability at least $1 - O(1/nk)$, we have*

$$\left\| \Lambda_{\alpha,\beta}^{1/2} \widehat{\boldsymbol{f}}_\alpha \right\|^2 \geq \min\left\{1, \mu^{-2}\right\} \Omega(n^{r-t-1}) + \Omega(n^{1-t-p}).$$

## F.3 Upper bounding the numerator: the unnormalized correlation

We now turn to the more involved part of the bound: proving an upper bound on the numerator. As before, we can bound the split up the numerator into favored and unfavored terms. For each term, we will show that it is dominated by the denominator, in the precise sense that each term is

$$o(\min\left\{1, \mu^{-2}\right\} n^{r-t-1} + n^{1-t-p}).$$

Now, let's look at the numerator, which is the bilinear form

$$\lambda_F^2 \sum_{j \notin \{\alpha,\beta\}, j \in [s]} \boldsymbol{y}_\alpha^\top \boldsymbol{A}^{-1} \boldsymbol{z}_j \boldsymbol{z}_j^\top \boldsymbol{A}^{-1} \boldsymbol{y}_\beta + \lambda_U^2 \sum_{j > s} \boldsymbol{y}_\alpha^\top \boldsymbol{A}^{-1} \boldsymbol{z}_j \boldsymbol{z}_j^\top \boldsymbol{A}^{-1} \boldsymbol{y}_\beta. \tag{143}$$

$$= \underbrace{\lambda_F^2 \left\langle \boldsymbol{Z}_L^\top \boldsymbol{A}^{-1} \boldsymbol{y}_\alpha, \boldsymbol{Z}_L^\top \boldsymbol{A}^{-1} \boldsymbol{y}_\beta \right\rangle}_{\mathsf{cor}_{\alpha,\beta,L}} + \underbrace{\lambda_F^2 \left\langle \boldsymbol{Z}_F^\top \boldsymbol{A}^{-1} \boldsymbol{y}_\alpha, \boldsymbol{Z}_F^\top \boldsymbol{A}^{-1} \boldsymbol{y}_\beta \right\rangle}_{\mathsf{cor}_{\alpha,\beta,F}} + \underbrace{\lambda_U^2 \sum_{j > s} \boldsymbol{y}_\alpha^\top \boldsymbol{A}^{-1} \boldsymbol{z}_j \boldsymbol{z}_j^\top \boldsymbol{A}^{-1} \boldsymbol{y}_\beta}_{\mathsf{cor}_{\alpha,\beta,U}}$$

$$\tag{144}$$

We refer to the the first term as the label defining correlation $\mathsf{cor}_{\alpha,\beta,L}$, the second term as the favored correlation $\mathsf{cor}_{\alpha,\beta,F}$, and the last term as the unfavored correlation $\mathsf{cor}_{\alpha,\beta,U}$. Here, we abuse terminology slightly and refer to these inner products as *correlations*, even though strictly speaking, they are unnormalized.

### F.3.1 Bounding the favored correlation

We now bound the correlation coming from the favored features; we will ultimately show that its contribution is $\min\left\{1, \mu^{-2}\right\} o(n^{r-t-1})$. Recall that $\boldsymbol{W}_R \in \mathbb{R}^{n \times (s-2)}$ is the weighted feature matrix

for the $s-2$ favored features aside from $\alpha$ and $\beta$. Then the label-defining+favored correlation $\mathsf{cor}_{\alpha,\beta,L} + \mathsf{cor}_{\alpha,\beta,F}$ can be written succinctly as

$$\lambda_F^2 \left\langle \boldsymbol{Z}_R^\top \boldsymbol{A}^{-1} \boldsymbol{y}_\alpha, \boldsymbol{Z}_R^\top \boldsymbol{A}^{-1} \boldsymbol{y}_\beta \right\rangle. \tag{145}$$

Why should we be able to bound this better than Cauchy-Schwarz? Intuitively, although there is a mild dependence between $\boldsymbol{y}_\alpha$ and $\boldsymbol{y}_\beta$, it is not strong enough to cause $\boldsymbol{Z}_R^\top \boldsymbol{A}^{-1} \boldsymbol{y}_\alpha$ and $\boldsymbol{Z}_R^\top \boldsymbol{A}^{-1} \boldsymbol{y}_\beta$ to point in the same direction.

To formalize this argument, we will first follow the strategy to bound the favored *contamination*. In particular, using the push-through form of Woodbury (Lemma B.2) we see that

$$\left\langle \boldsymbol{Z}_R^\top \boldsymbol{A}^{-1} \boldsymbol{y}_\alpha, \boldsymbol{Z}_R^\top \boldsymbol{A}^{-1} \boldsymbol{y}_\beta \right\rangle = \boldsymbol{y}_\alpha^\top \boldsymbol{A}_{-R}^{-1} \boldsymbol{Z}_R (\boldsymbol{I}_{s-2} + \boldsymbol{H}_R)^{-2} \boldsymbol{Z}_R^\top \boldsymbol{A}_{-R}^{-1} \boldsymbol{y}_\beta. \tag{146}$$

Now, we can apply Proposition B.6 to replace $(\boldsymbol{I}_{s-2} + \boldsymbol{H}_R)^{-2}$ with $\min\{\mu^2, 1\}(\boldsymbol{I}_{s-2} + \boldsymbol{E})$, where $\|\boldsymbol{E}\|_2 = O(n^{-\kappa_{11}})$ with extremely high probability. Cauchy-Schwarz yields that

$$\left\langle \boldsymbol{Z}_R^\top \boldsymbol{A}^{-1} \boldsymbol{y}_\alpha, \boldsymbol{Z}_R^\top \boldsymbol{A}^{-1} \boldsymbol{y}_\beta \right\rangle \tag{147}$$

$$\leq \min\{\mu^2, 1\} \left\langle \boldsymbol{Z}_R^\top \boldsymbol{A}_{-R}^{-1} \boldsymbol{y}_\alpha, \boldsymbol{Z}_R^\top \boldsymbol{A}_{-R}^{-1} \boldsymbol{y}_\beta \right\rangle \tag{148}$$

$$+ \min\{\mu^2, 1\} \|\boldsymbol{E}\|_2 \left\|\boldsymbol{Z}_R^\top \boldsymbol{A}_{-R}^{-1} \boldsymbol{y}_\beta \right\|_2 \left\|\boldsymbol{Z}_R^\top \boldsymbol{A}_{-R}^{-1} \boldsymbol{y}_\alpha \right\|_2. \tag{149}$$

The term in Eq. (149) can be bounded in the same way that we bounded the favored contamination. Indeed, since we can swap in $\boldsymbol{y}_\alpha$ and $\boldsymbol{y}_\beta$ with $\boldsymbol{\Delta} y$, the argument that proved the bounds on $\left\|\boldsymbol{Z}_R^\top \boldsymbol{A}_{-R}^{-1} \boldsymbol{\Delta} y\right\|_2$ port over immediately. After using the scaling for $\lambda_F$ and the fact that $\|\boldsymbol{E}\|_2 = O(n^{-\kappa_{11}})$, we conclude that this Cauchy-Schwarz error term is at most $\min\{1, \mu^{-2}\} o(n^{r-t-1})$ with probability at least $1 - O(1/nk)$.

Let us now turn to the term in Eq. (148). As in the proof for the lower bound for favored contamination Lemma E.2, to get better concentration than Cauchy-Schwarz, we want to use Bernstein. We can rewrite it suggestively as

$$\sum_{j \in [s] \setminus \{\alpha, \beta\}} (\boldsymbol{z}_j^\top \boldsymbol{A}_{-R}^{-1} \boldsymbol{y}_\alpha)(\boldsymbol{z}_j^\top \boldsymbol{A}_{-R}^{-1} \boldsymbol{y}_\beta) \tag{150}$$

We cannot immediately power through with the calculation, because $\boldsymbol{A}_{-R}^{-1}$ is not independent of $\boldsymbol{y}_\alpha$ or $\boldsymbol{y}_\beta$. The main idea is to again use Woodbury and show that the dependent portions contribute negligibly to $\boldsymbol{z}_j^\top \boldsymbol{A}_{-R}^{-1} \boldsymbol{y}_\alpha$. Therefore the dependent contributions get dominated by the lower bound on the correlation.

As in the proof for bounding the favored contamination, we can further define $\boldsymbol{W}_{\alpha,\beta} = [\boldsymbol{w}_\alpha \quad \boldsymbol{w}_\beta]$ and $\boldsymbol{H}_{\alpha,\beta}^{(s)} = \boldsymbol{W}_{\alpha,\beta}^\top \boldsymbol{A}_{-s}^{-1} \boldsymbol{W}_{\alpha,\beta}$. Then we can deduce from another application of Woodbury that

$$\boldsymbol{z}_j^\top \boldsymbol{A}_{-R}^{-1} \boldsymbol{y}_\alpha = \boldsymbol{z}_j^\top \boldsymbol{A}_{-s}^{-1} \boldsymbol{y}_\alpha - \boldsymbol{z}_j^\top \boldsymbol{A}_{-s}^{-1} \boldsymbol{W}_{\alpha,\beta} (\boldsymbol{I}_2 + \boldsymbol{H}_{\alpha,\beta})^{-1} \boldsymbol{W}_{\alpha,\beta}^\top \boldsymbol{A}_{-s}^{-1} \boldsymbol{y}_\alpha. \tag{151}$$

Again, we can argue that the second term is bounded in magnitude by

$$\min\{1, \mu^{-1}\} n^{\frac{1}{2} - t - p} \log(ns) = \min\{1, \mu^{-1}\} O(n^{\frac{1-t}{2} - p} \cdot n^{-\kappa_6}), \tag{152}$$

because $t > 0$. Since Hanson-Wright (Corollary C.3) implies that $\boldsymbol{z}_j^\top \boldsymbol{A}_{-s}^{-1} \boldsymbol{y}_\alpha = \tilde{O}(n^{\frac{1-t}{2} - p})$, this implies that $\boldsymbol{z}_j^\top \boldsymbol{A}_{-R}^{-1} \boldsymbol{y}_\alpha = \tilde{O}(n^{\frac{1-t}{2} - p})$, and similarly for $\beta$. Hence we have

$$(\boldsymbol{z}_j^\top \boldsymbol{A}_{-R}^{-1} \boldsymbol{y}_\alpha)(\boldsymbol{z}_j^\top \boldsymbol{A}_{-R}^{-1} \boldsymbol{y}_\beta) \tag{153}$$

$$\leq (\boldsymbol{z}_j^\top \boldsymbol{A}_{-s}^{-1} \boldsymbol{y}_\alpha + \min\{1, \mu^{-1}\} O(n^{\frac{1-t}{2} - p - \kappa_6}))(\boldsymbol{z}_j^\top \boldsymbol{A}_{-s}^{-1} \boldsymbol{y}_\beta + \min\{1, \mu^{-1}\} O(n^{\frac{1-t}{2} - p - \kappa_6})) \tag{154}$$

$$\leq (\boldsymbol{z}_j^\top \boldsymbol{A}_{-s}^{-1} \boldsymbol{y}_\alpha)(\boldsymbol{z}_j^\top \boldsymbol{A}_{-s}^{-1} \boldsymbol{y}_\beta) + \min\{1, \mu^{-1}\} O(n^{\frac{1-t}{2} - p - \kappa_6}) \tilde{O}(n^{\frac{1-t}{2} - p}) \tag{155}$$

$$\leq (\boldsymbol{z}_j^\top \boldsymbol{A}_{-s}^{-1} \boldsymbol{y}_\alpha)(\boldsymbol{z}_j^\top \boldsymbol{A}_{-s}^{-1} \boldsymbol{y}_\beta) + \min\{1, \mu^{-1}\} o(n^{1 - t - 2p}). \tag{156}$$

This implies that we can rewrite Eq. (150) as

$$\left( \sum_{j \in [s] \setminus \{\alpha, \beta\}} (\boldsymbol{z}_j^\top \boldsymbol{A}_{-s}^{-1} \boldsymbol{y}_\alpha)(\boldsymbol{z}_j^\top \boldsymbol{A}_{-s}^{-1} \boldsymbol{y}_\beta) \right) \pm \min\{1, \mu^{-1}\} o(n^{r+1 - t - 2p}). \tag{157}$$

Let us argue that the second term in Eq. (157) will be negligible compared to the denominator, which is $\min\{1,\mu^{-2}\}\Omega(n^{r-t-1})$. Tracing back up the stack, we see that its contribution to the favored correlation will be at most

$$\lambda_F^2 \min\{\mu^2,1\}\cdot\min\{1,\mu^{-1}\}o(n^{r+1-t-2p}) \le \mu^{-2}\min\{\mu^2,\mu^{-1}\}o(n^{r-t-1}) \tag{158}$$

$$\le \min\{1,\mu^{-3}\}o(n^{r-t-1}) \tag{159}$$

$$\le \min\{1,\mu^{-2}\}o(n^{r-t-1}). \tag{160}$$

Turning back to the first term, we are now in a position to apply Bernstein. Note that $(\boldsymbol{A}_{-s}^{-1},\boldsymbol{y}_\alpha,\boldsymbol{y}_\beta)$ are independent of $\boldsymbol{Z}_R$. Hence conditioned on $\boldsymbol{A}_{-s}^{-1},\boldsymbol{y}_\alpha$, and $\boldsymbol{y}_\beta$, $\langle\boldsymbol{Z}_R^\top\boldsymbol{A}_{-s}^{-1}\boldsymbol{y}_\alpha,\boldsymbol{Z}_R^\top\boldsymbol{A}_{-s}^{-1}\boldsymbol{y}_\beta\rangle$ is a sum of $s-2$ subexponential variables, and by Lemma 2.7.7 of Vershynin (2018) each of these random variables conditionally has subexponential norm at most $\|\boldsymbol{A}_{-s}^{-1}\boldsymbol{y}_\alpha\|_2\|\boldsymbol{A}_{-s}^{-1}\boldsymbol{y}_\beta\|_2$ and conditional mean $\langle\boldsymbol{A}_{-s}^{-1}\boldsymbol{y}_\alpha,\boldsymbol{A}_{-s}^{-1}\boldsymbol{y}_\beta\rangle$.

We can use Hanson-Wright (Theorem 4.1) to bound both of these quantities. Indeed, it implies that with probability at least $1-O(1/nk)$,

$$\|\boldsymbol{A}_{-s}^{-1}\boldsymbol{y}_\alpha\|_2\|\boldsymbol{A}_{-s}^{-1}\boldsymbol{y}_\beta\|_2 \le O(n^{1-t-2p}). \tag{161}$$

Let us now compute the Hanson-Wright bound for $\langle\boldsymbol{A}_{-s}^{-1}\boldsymbol{y}_\alpha,\boldsymbol{A}_{-s}^{-1}\boldsymbol{y}_\beta\rangle$. Note that $\boldsymbol{A}_{-s}^{-1}$ is independent of $\boldsymbol{y}_\alpha$ and $\boldsymbol{y}_\beta$, so we can condition on $\boldsymbol{A}_{-s}^{-1}$ and conclude that with probability at least $1-O(1/nk)$

$$\langle\boldsymbol{A}_{-s}^{-1}\boldsymbol{y}_\alpha,\boldsymbol{A}_{-s}^{-1}\boldsymbol{y}_\beta\rangle \le \mathbb{E}[\langle\boldsymbol{A}_{-s}^{-1}\boldsymbol{y}_\alpha,\boldsymbol{A}_{-s}^{-1}\boldsymbol{y}_\beta\rangle\mid\boldsymbol{A}_{-s}^{-1}] + c_6 n^{\frac{1-t}{2}}\|\boldsymbol{A}_{-s}^{-2}\|_2\sqrt{\log(nk)}. \tag{162}$$

We can rewrite the expectation as

$$\mathrm{Tr}\big(\boldsymbol{A}_s^{-2}\mathbb{E}[\boldsymbol{y}_\beta\boldsymbol{y}_\alpha^\top]\big). \tag{163}$$

Clearly, $\mathbb{E}[\boldsymbol{y}_\beta\boldsymbol{y}_\alpha^\top]$ is diagonal, and each diagonal entry is equal to . Let $\rho=\frac{1}{k}$. Then since $\boldsymbol{y}_\alpha=1-\rho$ implies $\boldsymbol{y}_\beta=-\rho$ and vice versa, we get

$$\mathbb{E}[\boldsymbol{y}_\alpha[i]\boldsymbol{y}_\beta[i]] = 2(1-\rho)(-\rho)\Pr[\boldsymbol{y}_\alpha[i]=1-\rho] + (-\rho)^2\Pr[\boldsymbol{y}_\alpha[i]=\boldsymbol{y}_\beta[i]=-\rho] \tag{164}$$

$$\le -2\rho^2(1-\rho) + \rho^2\Pr[\boldsymbol{y}_\alpha[i]=-\rho] \tag{165}$$

$$\le -\rho^2(1-\rho). \tag{166}$$

In other words, the expectation is *negative*, so we can neglect it in our upper bound.

On the other hand, the deviation term is with very high probability at most

$$c_6 n^{\frac{1-t}{2}}\|\boldsymbol{A}_{-s}^{-2}\|_2\sqrt{\log(nk)} \le c_6 n^{\frac{1-t}{2}-2p}\sqrt{\log(nk)}. \tag{167}$$

Combining all of our bounds, Bernstein yields

$$\big\langle\boldsymbol{Z}_R^\top\boldsymbol{A}_{-s}^{-1}\boldsymbol{y}_\alpha,\boldsymbol{Z}_R^\top\boldsymbol{A}_{-s}^{-1}\boldsymbol{y}_\beta\big\rangle \le \left(\sum_{j\in[s]\backslash\{\alpha,\beta\}} c_6 n^{\frac{1-t}{2}-2p}\sqrt{\log(nk)}\right) + n^{\frac{r}{2}+1-t-2p} \tag{168}$$

$$\le n^{r+\frac{1-t}{2}-2p}\sqrt{\log(nk)} + n^{\frac{r}{2}-t+1-2p}. \tag{169}$$

Again, let's trace all the way back to Eq. (148) and then the favored correlation bound. We have shown that $\big\langle\boldsymbol{Z}_R^\top\boldsymbol{A}_{-s}^{-1}\boldsymbol{y}_\alpha,\boldsymbol{Z}_R^\top\boldsymbol{A}_{-s}^{-1}\boldsymbol{y}_\beta\big\rangle$'s contribution to the favored correlation is at most

$$c_{17}\lambda_F^2\min\{\mu^2,1\}\big(n^{r+\frac{1-t}{2}-2p}\sqrt{\log(nk)} + n^{\frac{r}{2}-t+1-2p}\big) \tag{170}$$

$$= c_{17}\mu^{-2}\min\{\mu^2,1\}\big(n^{r-\frac{1+t}{2}-1}\sqrt{\log(nk)} + n^{\frac{r}{2}-t-1}\big) \tag{171}$$

$$\le c_{17}\min\{1,\mu^{-2}\}\big(n^{r-\frac{1+t}{2}-1}\sqrt{\log(nk)} + n^{\frac{r}{2}-t-1}\big) \tag{172}$$

$$\le c_{17}\min\{1,\mu^{-2}\}o(n^{r-t-1}), \tag{173}$$

where the last line follows becuase $0<t<r<1$, and $c_{17}$ is a positive constant.

### F.3.2 Bounding the unfavored correlation

Now, let us show that the unfavored correlation $\mathsf{cor}_{\alpha,\beta,U}$ is negligible; more precisely, we'll show that it's $\min\left\{1,\mu^{-2}\right\}o(n^{r-t-1}) + o(n^{1-t-p})$. We can rewrite $\mathsf{cor}_{\alpha,\beta,U}$ as

$$\lambda_U \boldsymbol{y}_\alpha^\top \boldsymbol{A}^{-1} \boldsymbol{A}_U \boldsymbol{A}^{-1} \boldsymbol{y}_\beta,$$

and play the same game with using Woodbury to replace $\boldsymbol{A}^{-1}$ with $\boldsymbol{A}^{-1} - \boldsymbol{A}_U^{-1} \boldsymbol{M}_s \boldsymbol{A}_U^{-1}$, where we recall that

$$\boldsymbol{M}_s \triangleq \boldsymbol{W}_s (\boldsymbol{I}_s + \boldsymbol{H}_s)^{-1} \boldsymbol{W}_s^\top \in \mathbb{R}^{n \times n}.$$

This yields

$$\boldsymbol{y}_\alpha^\top \boldsymbol{A}_U^{-1} \boldsymbol{y}_\beta - 2\boldsymbol{y}_\alpha^\top \boldsymbol{A}_U^{-1} \boldsymbol{M}_s \boldsymbol{A}_U^{-1} \boldsymbol{y}_\beta + \boldsymbol{y}_\alpha^\top \boldsymbol{A}_U^{-1} \boldsymbol{M}_s \boldsymbol{A}_U^{-1} \boldsymbol{M}_s \boldsymbol{A}_U^{-1} \boldsymbol{y}_\beta. \tag{174}$$

Let us first focus on the first term of Eq. (174). Hanson-Wright implies that with probability at least $1 - O(1/n)$,

$$\left| \boldsymbol{y}_\alpha^\top \boldsymbol{A}_U^{-1} \boldsymbol{y}_\beta - \mathbb{E}[\boldsymbol{y}_\alpha^\top \boldsymbol{A}_U^{-1} \boldsymbol{y}_\beta | \boldsymbol{A}_U^{-1}] \right| \leq n^{\frac{1-t}{2}} \left\| \boldsymbol{A}_U^{-1} \right\|_2 \sqrt{\log n}. \tag{175}$$

Also, $\mathbb{E}[\boldsymbol{y}_\alpha^\top \boldsymbol{A}_U^{-1} \boldsymbol{y}_\beta | \boldsymbol{A}_U^{-1}] = \mathrm{Tr}\left(\boldsymbol{A}_U^{-1} \mathbb{E}[\boldsymbol{y}_\beta \boldsymbol{y}_\alpha^\top]\right) = \Theta(n^{1-2t-p})$ with high probability, and $\left\| \boldsymbol{A}_U^{-1} \right\|_2 \leq n^{-p}$ with extremely high probability. Hence we see that $\boldsymbol{y}_\alpha^\top \boldsymbol{A}_U^{-1} \boldsymbol{y}_\beta \leq O(n^{\frac{1-t}{2}-p}) \leq o(n^{1-t-p})$ as $t > 0$.

Next, let's turn to the second and third terms of Eq. (174). We claim that only the second term will be relevant to bound asymptotically, and moreover that they are both $\min\left\{1,\mu^{-2}\right\}o(n^{r-t-1})$. Since

$$\boldsymbol{M}_s \boldsymbol{A}_U^{-1} \boldsymbol{M}_s = \boldsymbol{W}_s (\boldsymbol{I}_s + \boldsymbol{H}_s)^{-1} \boldsymbol{H}_s (\boldsymbol{I}_s + \boldsymbol{H}_s)^{-1} \boldsymbol{W}_s^\top$$
$$= \boldsymbol{W}_s ((\boldsymbol{I}_s + \boldsymbol{H}_s)^{-1} - (\boldsymbol{I}_s + \boldsymbol{H}_s)^{-2}) \boldsymbol{W}_s^\top,$$

the second and third term can be rewritten as

$$-2\boldsymbol{y}_\alpha^\top \boldsymbol{A}_U^{-1} \boldsymbol{W}_s (\boldsymbol{I}_s + \boldsymbol{H}_s)^{-1} \boldsymbol{W}_s^\top \boldsymbol{A}_U^{-1} \boldsymbol{y}_\beta$$
$$+ \boldsymbol{y}_\alpha^\top \boldsymbol{A}_U^{-1} \boldsymbol{W}_s ((\boldsymbol{I}_s + \boldsymbol{H}_s)^{-1} - (\boldsymbol{I}_s + \boldsymbol{H}_s)^{-2}) \boldsymbol{W}_s^\top \boldsymbol{A}_U^{-1} \boldsymbol{y}_\beta.$$

As we are going to use Hanson-Wright to bound the entries of $\boldsymbol{Z}_s^\top \boldsymbol{A}_U^{-1} \boldsymbol{y}_\alpha$, it follows that only the second term of Eq. (174) is relevant asymptotically.

To bound the second term, we will use Cauchy-Schwarz. We see that

$$\boldsymbol{y}_\alpha^\top \boldsymbol{A}_U^{-1} \boldsymbol{M}_s \boldsymbol{A}_U^{-1} \boldsymbol{y}_\beta \leq \lambda_F \left\| (\boldsymbol{I}_s + \boldsymbol{H}_s)^{-1} \right\|_2 \left\| \boldsymbol{Z}_s^\top \boldsymbol{A}_U^{-1} \boldsymbol{y}_\alpha \right\|_2 \left\| \boldsymbol{Z}_s^\top \boldsymbol{A}_U^{-1} \boldsymbol{y}_\beta \right\|_2 \tag{176}$$
$$\leq \lambda_F \min\left\{\mu, 1\right\} O(n^{r-t+1-2p}) \log(ns) \tag{177}$$
$$\leq \mu^{-1} \min\left\{\mu, 1\right\} O(n^{r-t-p}) \log(ns) \tag{178}$$
$$\leq \min\left\{1, \mu^{-1}\right\} O(n^{r-t-p}) \log(ns), \tag{179}$$

where in the second line we have used Proposition B.6.

Now, note that if regression works, this yields an upper bound of $O(n^{r-t-p}) \log(ns)$. But since $p > 1$, this is $o(n^{r-t-1})$, which means this contribution is dominated by the denominator.

On the other hand, if regression fails, then the upper bound is now $\mu^{-1} O(n^{r-t-p}) \log(ns)$, which we claim is $o(\mu^{-2} n^{r-t-1})$. Indeed, from the definition of the bi-level ensemble Definition 1, we have $p > q + r$, so

$$\min\left\{1, \mu^{-1}\right\} n^{r-t-p} \leq \mu^{-1} n^{r-t-1} \cdot n^{1-p}$$
$$\leq \mu^{-1} o(n^{r-t-1} \cdot n^{1-q-r})$$
$$= \mu^{-2} o(n^{r-t-1}),$$

as desired.

Let us now go back to Eq. (174) and combine our two bounds. Since $\lambda_U = O(1)$, we have just shown that

$$\mathsf{cor}_{\alpha,\beta,U} \leq \min\left\{1, \mu^{-2}\right\} o(n^{r-t-1}) + o(n^{1-t-p}), \tag{180}$$

as desired.

# G  A new variant of the Hanson-Wright inequality with soft sparsity

In this section, we prove Theorem 4.1. First, we outline a high level idea of the proof. The starting point of the proof is to explicitly decompose the quadratic form into diagonal and off-diagonal terms

$$\boldsymbol{x}^\top \boldsymbol{M} \boldsymbol{y} - \mathbb{E}[\boldsymbol{x}^\top \boldsymbol{M} \boldsymbol{y}] = \sum_{i,j} m_{ij} X_i Y_j - \sum_i m_{ii} \mathbb{E}[X_i Y_i] \tag{181}$$

$$= \underbrace{\sum_i m_{ii}(X_i Y_i - \mathbb{E}[X_i Y_i])}_{\triangleq S_{\text{diag}}} + \underbrace{\sum_{i \neq j} m_{ij} X_i Y_j}_{\triangleq S_{\text{offdiag}}} \tag{182}$$

where in the first line we have used the fact that for $i \neq j$, $X_i$ and $Y_j$ are independent and mean zero to conclude that $\mathbb{E}[X_i Y_j] = 0$.

We can start with the upper tail inequality $\mathbb{P}[\boldsymbol{x}^\top \boldsymbol{M} \boldsymbol{y} - \mathbb{E}[\boldsymbol{x}^\top \boldsymbol{M} \boldsymbol{y}] > t]$ and conclude the lower tail inequality by replacing $\boldsymbol{M}$ with $-\boldsymbol{M}$. To bound $S_{\text{diag}}$ and $S_{\text{offdiag}}$, we will proceed by explicitly bounding the MGF and applying Chernoff's inequality.

## G.1  Diagonal terms

For the diagonal terms, we want to bound the MGF of $S_{\text{diag}} = \sum_i m_{ii}(X_i Y_i - \mathbb{E}[X_i Y_i])$. For $\lambda^2 < \frac{1}{2C_1 K^2 \max_i m_{ii}^2}$, we obtain

$$\mathbb{E} \exp(\lambda S_{\text{diag}}) = \prod_{i=1}^n \mathbb{E}_{X_i, Y_i} \exp(\lambda m_{ii}(X_i Y_i - \mathbb{E}[X_i Y_i])) \tag{183}$$

$$\leq \prod_{i=1}^n \mathbb{E}_{Y_i} \mathbb{E}_{X_i}[\exp(\lambda m_{ii} Y_i(X_i - \mathbb{E}[X_i|Y_i]))|Y_i] \tag{184}$$

$$\leq \prod_{i=1}^n \mathbb{E}_{Y_i} \exp\big(C_1 \lambda^2 m_{ii}^2 K^2 Y_i^2\big) \tag{185}$$

where we have applied Jensen's inequality in the second line and the subgaussian assumption on $X_i$ conditioned on $Y_i$ in the last line. Here, $C_1$ is a universal positive constant relating the equivalent formulations of subgaussianity Vershynin (2018). Continuing with our calculation, we have

$$\mathbb{E} \exp(\lambda S_{\text{diag}}) \leq \prod_{i=1}^n \mathbb{E}_{Y_i}[1 + 2C_1 \lambda^2 m_{ii}^2 K^2 Y_i^2] \tag{186}$$

$$\leq \prod_{i=1}^n (1 + 2C_1 \pi \lambda^2 K^2 m_{ii}^2) \tag{187}$$

$$\leq \exp\left(2C_1 \pi \lambda^2 K^2 \sum_{i=1}^n m_{ii}^2\right). \tag{188}$$

where in the first line we have used the inequality $\exp(x) \leq 1 + 2x$ valid for $x \leq \frac{1}{2}$, in the second line we have used the soft sparsity assumption on $Y_i$, and in the last line we have used the inequality $1 + x \leq \exp(x)$, valid for all $x$.

Now Markov's inequality yields for $\epsilon > 0$ that

$$\Pr[S_{\text{diag}} > \epsilon] \leq \frac{\mathbb{E} \exp(\lambda S_{\text{diag}})}{\exp(\lambda \epsilon)} \tag{189}$$

$$\leq \exp\left(-\lambda \epsilon + 2\pi C_1 K^2 \lambda^2 \sum_{i=1}^n m_{ii}^2\right), \tag{190}$$

and optimizing $\lambda$ in the region $\lambda^2 \leq \frac{1}{2C_1 K^2 \max_i m_{ii}^2}$ yields

$$\lambda = \min\left\{\frac{\epsilon}{2C_1 K^2 \pi \sum_{i=1}^n m_{ii}^2}, \frac{1}{2C_1 K \max_i |m_{ii}|}\right\}. \tag{191}$$

Plugging in this value of $\lambda$ into the Markov calculation yields the desired upper tail bound. We can repeat the argument with $-M$ to get the lower tail bound. A union bound completes the proof.

### G.2 Offdiagonal terms

Following Rudelson and Vershynin (2013), for the offdiagonal terms we can decouple the terms in the sum. More precisely, the terms in $S_{\text{offdiag}}$ involving indices $i$ and $j$ are precisely $m_{ij}X_iY_j + m_{ji}Y_iX_j$. The issue is that $Y_i$ can be correlated with $X_i$, which complicates the behavior of this random variable. Decoupling ensures that for any $j \in [n]$ we will have exactly one term which involves either $X_j$ or $Y_j$, so in particular we will regain independence of the terms, allowing us to bound the MGF more easily.

Let $\{\delta_i\}_{i \in [n]}$ denote iid Bernoulli's with parameter $1/2$, which are independent of all other random variables.

Let

$$S_\delta \triangleq \sum_{i \neq j} m_{ij}\delta_i(1 - \delta_j)X_iY_j.$$

Since $\mathbb{E}[\delta_i(1 - \delta_j)] = \frac{1}{4}$, we have

$$S_{\text{offdiag}} = 4\mathbb{E}_\delta[S_\delta],$$

Hence, Jensen's inequality yields

$$\mathbb{E}_{\boldsymbol{x},\boldsymbol{y}} \exp(\lambda S_{\text{offdiag}}) \leq \mathbb{E}_{\boldsymbol{x},\boldsymbol{y},\delta} \exp(4\lambda S_\delta),$$

where we have used the independence of $\delta$ and all other random variables. It follows that it suffices to upper bound the MGF of $S_\delta$.

Define the random set $\Lambda_\delta = \{i \in [n] : \delta_i = 1\}$ to denote the indices selected by $\delta$. For a vector $\boldsymbol{u} \in \mathbb{R}^n$ we also introduce the shorthand $\boldsymbol{u}_{\Lambda_\delta}$ to denote the subvector of $\boldsymbol{u}$ where $\delta_i = 1$ and $\boldsymbol{u}_{\Lambda_\delta^c}$ to denote the subvector of $\boldsymbol{u}$ where $\delta_i = 0$.

Hence, we can rewrite $S_\delta \triangleq \sum_{i \in \Lambda_\delta, j \in \Lambda_\delta^c} m_{ij}X_iY_j$. For $|\lambda| \leq \frac{1}{2C_1K\|M\|_2}$, we have

$$\mathbb{E}\exp(\lambda S_{\text{offdiag}}) \leq \mathbb{E}\exp(4\lambda S_\delta) \tag{192}$$

$$\leq \mathbb{E}_\delta \prod_{i \in \Lambda_\delta, j \in \Lambda_\delta^c} \mathbb{E}_{\boldsymbol{x}_{\Lambda_\delta}, \boldsymbol{y}_{\Lambda_\delta^c}}[\exp(\lambda m_{ij}X_iY_j)] \tag{193}$$

Now we can use the fact that the $X_i$ and $Y_j$ are mean zero and independent because $i \in \Lambda_\delta$ and $j \in \Lambda_\delta^c$, to show that

$$\prod_{i \in \Lambda_\delta, j \in \Lambda_\delta^c} \mathbb{E}_{\boldsymbol{x}_{\Lambda_\delta}, \boldsymbol{y}_{\Lambda_\delta^c}}[\exp(\lambda m_{ij}X_iY_j)] \leq \prod_{i \in \Lambda_\delta, j \in \Lambda_\delta^c} \mathbb{E}_{\boldsymbol{y}_{\Lambda_\delta^c}}[\exp(C_1\lambda^2K^2m_{ij}^2Y_j^2)] \tag{194}$$

$$\leq \prod_{i \in \Lambda_\delta, j \in \Lambda_\delta^c} \mathbb{E}_{\boldsymbol{y}_{\Lambda_\delta^c}}[1 + 2C_1\lambda^2K^2m_{ij}^2Y_j^2)] \tag{195}$$

$$\leq \prod_{i \in \Lambda_\delta, j \in \Lambda_\delta^c} (1 + 2\pi C_1\lambda^2K^2m_{ij}^2Y_j^2) \tag{196}$$

$$\leq \prod_{i \in \Lambda_\delta, j \in \Lambda_\delta^c} \exp(2\pi C_1\lambda^2K^2m_{ij}^2Y_j^2) \tag{197}$$

$$\leq \exp\left(2\pi C_1\lambda^2K^2\|M\|_F^2\right). \tag{198}$$

In the first line, we have used the subgaussianity of $X_i$; in the second line, we have used the assumption on $\lambda$, in the third line, we have used the variance bound on $Y_j$.

Again, we can apply Markov's inequality and to see that for $\epsilon > 0$,

$$\Pr[S_{\text{diag}} > \epsilon] \leq \frac{\mathbb{E}\exp(\lambda S_{\text{diag}})}{\exp(\lambda\epsilon)} \tag{199}$$

$$\leq \exp\left(-\lambda\epsilon + 2\pi C_1K^2\lambda^2\|M\|_F^2\right), \tag{200}$$

Picking

$$\lambda = \min\left\{\frac{\epsilon}{2C_1 K^2 \pi \|\boldsymbol{M}\|_F^2}, \frac{1}{2C_1 K \|\boldsymbol{M}\|_2}\right\} \tag{201}$$

yields the desired result.

## H Proofs of main lemmas for concentration of spectrum

The goal of this section is ultimately to prove Proposition B.6, which asserts that for valid $(T, S)$, the hat matrix $\boldsymbol{H}_{T,S}$ is a flat matrix whose spectrum is $\min\{\mu, 1\}(1 + o(1))$ with extremely high probability. First, let us recall some notation. For any $\varnothing \neq T \subseteq S \subseteq [s]$, we can define the $(T, S)$ hat matrix as $\boldsymbol{H}_{T,S} \triangleq \boldsymbol{W}_T^\top \boldsymbol{A}_{-S}^{-1} \boldsymbol{W}_T$. Here, $\boldsymbol{W}_T$ is the $n \times |T|$ matrix of weighted features in $T$, and $\boldsymbol{A}_{-T} = \boldsymbol{A} - \boldsymbol{W}_T \boldsymbol{W}_T^\top$ is the leave-$T$-out Gram matrix.

First, Wishart concentration applied to $\boldsymbol{W}_T^\top \boldsymbol{W}_T$ yields the following result.

**Lemma H.1.** *Recall that $\mu \triangleq n^{q+r-1}$ and $\boldsymbol{W}_T^\top \boldsymbol{W}_T \in \mathbb{R}^{|T| \times |T|}$. For any nonempty $T \subseteq [s]$, with probability at least $1 - 2e^{-\sqrt{n}}$ we have that for all $i \in [|T|]$,*

$$\mu_i(\boldsymbol{W}_T^\top \boldsymbol{W}_T) = \left(1 \pm c_T \sqrt{\frac{|T|}{n}}\right)\mu^{-1} n^p \tag{202}$$

*Proof.* We can apply Lemma B.3 with $\boldsymbol{M} = \boldsymbol{Z}_T \in \mathbb{R}^{n \times |T|}$, with $M = n$, $m = |T| = o(n)$, and $\epsilon = n^{\frac{1}{4}} = o(\sqrt{n|T|})$. Hence we have

$$n - 2\sqrt{n|T|} + o(\sqrt{n|T|}) \leq \mu_{|T|}(\boldsymbol{Z}_T^\top \boldsymbol{Z}_T) \leq \mu_1(\boldsymbol{Z}_T^\top \boldsymbol{Z}_T) \leq n + 2\sqrt{n|T|} + o(\sqrt{n|T|}).$$

Pluagging in the scaling $\lambda_F = n^{p-q-r}$ and dividing through by $n$ yields the desired result. Here, we define $c_T$ to be an appropriately defined positive constant which only depends on $|T|$ (as the favored features are identically distributed). $\qquad\square$

Next, we can use Wishart concentration to bound the spectrum of $\boldsymbol{A}_U$.

**Lemma H.2** (Concentration of spectrum for unfavored Gram matrix)**.** *Throughout this theorem, assume we are in the bi-level model (Definition 1). Define $\mu \triangleq n^{q+r-1}$. Recall $\boldsymbol{A}_U = \lambda_U \sum_{j>s} \boldsymbol{z}_j \boldsymbol{z}_j^\top \in \mathbb{R}^{n \times n}$. With probability at least $1 - 2e^{-n}$, for $i \in [n]$ we have*

$$\mu_i(\boldsymbol{A}_U) = (1 \pm c_{18} n^{\kappa_7}) n^p, \tag{203}$$

*where $c_{18}$ and $\kappa_7$ are positive constants. In other words, the spectrum of the unfavored Gram matrix $\boldsymbol{A}_U$ is flat.*

*Proof.* Note that $\boldsymbol{A}_U = \lambda_U \sum_{j>s} \boldsymbol{z}_j \boldsymbol{z}_j^\top$. Under the bi-level model, $\lambda_U = 1 + o(1)$. Now we can apply Lemma B.3 with $\boldsymbol{M} = \sum_{j>s} \boldsymbol{z}_j \boldsymbol{z}_j^\top$, $M = d - s = n^p - n^r$, $m = n = o(d)$, and $\epsilon = \sqrt{2n}$ to conclude that with probability at least $1 - 2e^{-n}$, we have

$$d - 2\sqrt{dn} + n - \sqrt{2n} \leq \mu_n(\sum_{j>s} \boldsymbol{z}_j \boldsymbol{z}_j^\top) \leq \mu_1(\sum_{j>s} \boldsymbol{z}_j \boldsymbol{z}_j^\top) \leq d + 2\sqrt{dn} + n + \sqrt{2n}. \tag{204}$$

We can obtain the spectrum of $\boldsymbol{A}_U$ by multiplying through by

$$\lambda_U = \frac{(1-a)d}{d-s} \tag{205}$$
$$= 1 + n^{\max\{-q, r-p\}} + o(n^{\max\{-q, r-p\}}), \tag{206}$$

where in the last line we have used the power series expansion for $\frac{1}{1-x} = 1 + x + o(x)$. Preserving only first order terms for $\lambda_U$ and the spectrum of $\sum_{j>s} \boldsymbol{z}_j \boldsymbol{z}_j^\top$ in Eq. (204) yields

$$\mu_i(\boldsymbol{A}_U) = (1 \pm c_{18} n^{\max\{\frac{1-p}{2}, r-p, -q\}}) n^p. \tag{207}$$

In fact, we know $\frac{1-p}{2} > 1 - p > r - p$, since $r < 1$ and $p > 1$. This means we can neglect the $r - p$ term in the max, define $\kappa_7 = \min\left\{\frac{p-1}{2}, -q\right\} > 0$ and $c_{18}$ to be an appropriately defined positive constant. □

Since $\boldsymbol{A}_{-T} = \boldsymbol{A}_U + \boldsymbol{W}_{[s]\setminus T}\boldsymbol{W}_{[s]\setminus T}^\top$, we can apply Lemmas H.1 and H.2 to control the spectrum of $\boldsymbol{A}_{-T}$. We show that there is a (potentially) spiked portion of the spectrum corresponding to the $s - |T|$ favored features which were not taken out, whereas the rest of the $n - s + |T|$ eigenvalues are flat.

**Lemma H.3.** *Recall that $\boldsymbol{A}_{-T} \in \mathbb{R}^{n\times n}$. For any nonempty $T \subseteq [s]$, with probability at least $1 - 2e^{-\sqrt{n}} - 2e^{-n}$, we have that for all $i \in [s - |T|]$,*

$$\mu_i(\boldsymbol{A}_{-T}) = \left(1 \pm c_T\sqrt{\frac{|T|}{n}}\right)\mu^{-1}n^p + \left(1 \pm c_{18}n^{-\kappa_7}\right)n^p. \tag{208}$$

*For all $i \in [n] \setminus [s - |T|]$, we have*

$$\mu_i(\boldsymbol{A}_{-T}) = \left(1 \pm c_{18}n^{-\kappa_7}\right)n^p. \tag{209}$$

*Proof.* We can write

$$\boldsymbol{A}_{-T} = \boldsymbol{W}_{[s]\setminus T}\boldsymbol{W}_{[s]\setminus T}^\top + \boldsymbol{A}_U. \tag{210}$$

Weyl's inequality (Horn and Johnson, 2012, Corollary 4.3.15) implies that for any $i \in [n]$, we have

$$\mu_i(\boldsymbol{W}_{[s]\setminus T}\boldsymbol{W}_{[s]\setminus T}^\top) + \mu_n(\boldsymbol{A}_U) \leq \mu_i(\boldsymbol{A}_{-T}) \leq \mu_i(\boldsymbol{W}_{[s]\setminus T}\boldsymbol{W}_{[s]\setminus T}^\top) + \mu_1(\boldsymbol{A}_U). \tag{211}$$

Then applying Lemmas H.1 and H.2, for $i \in [s - |T|]$ we conclude that

$$\mu_i(\boldsymbol{A}_{-T}) = \left(1 \pm c_T\sqrt{\frac{|T|}{n}}\right)\mu^{-1}n^p + \left(1 \pm c_{18}n^{-\kappa_7}\right)n^p.. \tag{212}$$

which proves Eq. (208).

For $i > s - |T|$, applying Lemma H.2 and the fact that $\mu_i(\boldsymbol{W}_{[s]\setminus T}\boldsymbol{W}_{[s]\setminus T}^\top) = 0$ to Eq. (211) yields

$$\mu_i(\boldsymbol{A}_{-T}) = \left(1 \pm c_{18}n^{-\kappa_7}\right)n^p. \tag{213}$$

which proves Eq. (209). □

By inverting the bounds proved above, we can also control the spectrum of $\boldsymbol{A}_{-T}^{-1}$.

**Corollary H.4.** *Recall that $\boldsymbol{A}_{-T} \in \mathbb{R}^{n\times n}$. For any nonempty $T \subseteq [s]$, with probability at least $1 - 2e^{-\sqrt{n}} - 2e^{-n}$, we have that for all $i \in [n - s + |T|]$,*

$$\mu_i(\boldsymbol{A}_{-T}^{-1}) = (1 \pm c_{19}n^{-\kappa_7})n^{-p} \tag{214}$$

*For all $i \in [n] \setminus [n - s + |T|]$, we have*

$$\mu_i(\boldsymbol{A}_{-T}^{-1}) = \min\{\mu, 1\}(1 \pm c_{20}n^{-\kappa_8})n^{-p}. \tag{215}$$

*where $\kappa_8$ is a positive constant depending on $|T|$.*

*Proof.* By inverting the bounds in Lemma H.3, using the fact that $\mu_i(\boldsymbol{A}_{-T}^{-1}) = \frac{1}{\mu_{n-i+1}(\boldsymbol{A}_{-T})}$ we see that for $i \in [n - s + |T|]$,

$$\mu_i(\boldsymbol{A}_{-T}^{-1}) = \frac{1}{1 \pm c_{18}n^{-\kappa_7}}n^{-p} \tag{216}$$

$$= (1 \pm c_{19}n^{-\kappa_7})n^{-p}, \tag{217}$$

where we have used the power series expansion $\frac{1}{1-x} = 1 + x + o(x^2)$ and $c_{19}$ is a positive constant.

On the other hand, for $i > n - s + |T|$, we get

$$\mu_i(\boldsymbol{A}_{-T}^{-1}) = \frac{1}{\left(1 \pm c_T \sqrt{\frac{|T|}{n}}\right)\mu^{-1} + (1 \pm c_{18} n^{-\kappa_7})} n^{-p} \tag{218}$$

$$= \min\{\mu, 1\}(1 \pm c_{20} n^{-\kappa_8}) n^{-p}, \tag{219}$$

where $c_{20}$ and $\kappa_8$ are positive constants defined as follows. If $q + r < 1$, i.e. regression works, then $\mu^{-1} = \omega(1)$, so the denominator becomes $\mu^{-1}\left(1 \pm c_T \sqrt{\frac{|T|}{n}} + \mu(1 \pm c_{18} n^{-\kappa_7})\right)$. Then, since $|T| \le s = n^r$, we see that we can pick

$$\kappa_8 = \min\left\{\frac{1-r}{2}, 1 - q - r\right\}.$$

On the other hand, if $q + r > 1$, i.e. regression fails, then $\mu^{-1} = o(1)$, and so we can define

$$\kappa_8 = \min\{\kappa_7, q + r - 1\}.$$

Hence to cover both cases we can pick

$$\kappa_8 = \min\left\{\frac{1-r}{2}, \kappa_7, |1 - q - r|\right\}.$$

The choice of $c_{20}$ is picked by again using the power series expansion for $\frac{1}{1-x}$. $\qquad\square$

Note that Corollary H.4 immediately implies Proposition B.4, with $\kappa_9$ defined based on picking $T = [k]$. We are now in a position to prove that the generalized hat matrices $\boldsymbol{H}_{T,S}$, and hence the Woodbury terms $(\boldsymbol{I}_{|T|} + \boldsymbol{H}_{T,S})^{-1}$ have a flat spectrum as well.

**Proposition B.6** (Generalized hat matrices are flat). *Assume we are in the bi-level ensemble Definition 1. For any nonempty $T \subseteq S \subseteq [s]$, with probability at least $1 - 2e^{-\sqrt{n}} - 2e^{-n}$, we have all the eigenvalues tightly controlled:*

$$\mu_i((\boldsymbol{I}_{|T|} + \boldsymbol{H}_{T,S})^{-1}) = \min\{\mu, 1\}(1 \pm c_{T,S} n^{-\kappa_{11}}). \tag{75}$$

*where $c_{T,S}$ and $\kappa_{11}$ are positive constants that depend on $|T|$ and $|S|$.*

*Proof.* We seek to control the spectrum of the hat matrix $\boldsymbol{H}_{T,S} = \boldsymbol{W}_T^\top \boldsymbol{A}_{-S}^{-1} \boldsymbol{W}_T$. We cannot directly use naive eigenvalue bounds to bound the minimum and maximum eigenvalue, as this does not rule out the possibility that $\boldsymbol{H}_{T,S}$ has a spike. Instead, we control the spectrum from first principles.

**The spectrum of $\boldsymbol{H}_{T,S}$ is flat:** Since $\boldsymbol{A}_{-S}^{-1}$ is symmetric, it has an eigendecomposition $\boldsymbol{V}\boldsymbol{D}\boldsymbol{V}^\top$, where $\boldsymbol{V}$ is an orthogonal matrix. Because $\boldsymbol{W}_T$ is a weighted subset of only the (equally) favored features, its law is rotationally invariant. Furthermore, since $\boldsymbol{A}_{-S}^{-1}$ is independent of $\boldsymbol{W}_T$ (as $T \subseteq S$), we can absorb the rotation $\boldsymbol{V}$ into $\boldsymbol{W}_T$ to reduce to the case where $\boldsymbol{A}_{-S}^{-1} = \boldsymbol{D}$. Here, we have

$$\boldsymbol{D} \triangleq \begin{bmatrix} \boldsymbol{D}_{\text{flat}} & \\ & \boldsymbol{D}_{\text{spiked}} \end{bmatrix} \in \mathbb{R}^{n \times n} = \begin{bmatrix} \mu_1(\boldsymbol{A}_{-S}^{-1}) & & \\ & \ddots & \\ & & \mu_n(\boldsymbol{A}_{-S}^{-1}) \end{bmatrix}, \tag{220}$$

where $\boldsymbol{D}_{\text{flat}} \in \mathbb{R}^{(n-s+|T|)\times(n-s+|T|)}$ and $\boldsymbol{D}_{\text{spiked}} \in \mathbb{R}^{(s-|T|)\times(s-|T|)}$ correspond to the flat and (downwards) spiked portions of the spectrum of $\boldsymbol{A}_{-S}^{-1}$. We can also correspondingly decompose

$$\boldsymbol{Z}_T = \begin{bmatrix} \boldsymbol{B}_T \\ \boldsymbol{C}_T \end{bmatrix}, \tag{221}$$

where $\boldsymbol{B}_T \in \mathbb{R}^{(n-s+|T|)\times|T|}$ and $\boldsymbol{C}_T \in \mathbb{R}^{(s-|T|)\times|T|}$. Note that each entry of these matrices are i.i.d. $N(0,1)$ variables.

By direct computation we have

$$\boldsymbol{Z}_T^\top \boldsymbol{D} \boldsymbol{Z}_T = \boldsymbol{B}_T^\top \boldsymbol{D}_{\mathsf{flat}} \boldsymbol{B}_T + \boldsymbol{C}_T^\top \boldsymbol{D}_{\mathsf{spiked}} \boldsymbol{C}_T \tag{222}$$

We thus have by using standard eigenvalue inequalities that

$$\mu_{|T|}(\boldsymbol{Z}_T^\top \boldsymbol{D} \boldsymbol{Z}_T) \geq \mu_{|T|}(\boldsymbol{B}_T^\top \boldsymbol{D}_{\mathsf{flat}} \boldsymbol{B}_T) + \mu_{|T|}(\boldsymbol{C}_T^\top \boldsymbol{D}_{\mathsf{spiked}} \boldsymbol{C}_T) \tag{223}$$

$$\geq \mu_{|T|}(\boldsymbol{B}_T^\top \boldsymbol{B}_T)\mu_{n-s+|T|}(\boldsymbol{A}_{-S}^{-1}) + \mu_{|T|}(\boldsymbol{C}_T^\top \boldsymbol{C}_T)\mu_n(\boldsymbol{A}_{-S}^{-1}) \tag{224}$$

$$\geq \mu_{|T|}(\boldsymbol{B}_T^\top \boldsymbol{B}_T)\mu_{n-s+|T|}(\boldsymbol{A}_{-S}^{-1}), \tag{225}$$

where in the last line we have used $\mu_n(\boldsymbol{A}_{-S}^{-1}) \geq 0$.

Since $n > s$, we have $n - s + |T| > |T|$, so we can apply Wishart concentration (Lemma B.3) to $\boldsymbol{B}_T^\top \boldsymbol{B}_T$ to obtain that with probability at least $1 - 2e^{-\sqrt{n}}$ we have

$$\mu_{|T|}(\boldsymbol{B}_T^\top \boldsymbol{B}_T) \geq n - s + |T| - 2\sqrt{(n-s+|T|)|T|} + o(\sqrt{(n-s+|T|)|T|}) \tag{226}$$

$$\geq n\left(1 - n^{r-1} - c_{21}\sqrt{\tfrac{|T|}{n}}\right), \tag{227}$$

where $c_{21}$ is a positive constant.

On the other hand, we can deduce that

$$\mu_1(\boldsymbol{Z}_T^\top \boldsymbol{D} \boldsymbol{Z}_T) \leq \mu_1(\boldsymbol{Z}_T^\top \boldsymbol{Z}_T)\mu_1(\boldsymbol{A}_{-S}^{-1}). \tag{228}$$

Lemma H.1 implies that with probability at least $1 - 2e^{-\sqrt{n}}$

$$\mu_1(\boldsymbol{Z}_T^\top \boldsymbol{Z}_T) \leq n\left(1 + c_T\sqrt{\tfrac{|T|}{n}}\right) \tag{229}$$

Similarly, Corollary H.4 implies that with probability at least $1 - 2e^{-n} - 2e^{-\sqrt{n}}$, $\mu_1(\boldsymbol{A}_{-S}^{-1})$ and $\mu_{n-s+|T|}(\boldsymbol{A}_{-S}^{-1})$ are both $(1 \pm c_{20}n^{\kappa_8})n^{-p}$. Since $|T| \leq s = n^r$, Eqs. (227) and (229) together demonstrate that for all $i \in [|T|]$,

$$\mu_i(\boldsymbol{Z}_T^\top \boldsymbol{D} \boldsymbol{Z}_T) = n^{1-p}(1 \pm c_{22}n^{-\kappa_{10}}). \tag{230}$$

Here, $c_{22}$ and $\kappa_{10}$ are positive constants defined as follows. Since $|T| \leq s = n^r$. Then $\kappa_{10} = \min\left\{1 - r, \frac{1-r}{2}, \kappa_8\right\}$, and $c_{22}$ is a constant chosen appropriately based on $c_{20}$ and $c_{21}$. Plugging in the scaling $\lambda_F = n^{p-q-r}$, we conclude that with extremely high probability, for all $i \in [|T|]$,

$$\mu_i(\boldsymbol{H}_{T,S}) = \mu^{-1}(1 \pm c_{22}n^{-\kappa_{10}}). \tag{231}$$

From here, it is easy to compute the spectrum of $(\boldsymbol{I}_{|T|} + \boldsymbol{H}_{T,S})^{-1}$. Indeed, reading off our result from Eq. (231) yields

$$\mu_i((\boldsymbol{I}_{|T|} + \boldsymbol{H}_{T,S})^{-1}) = \frac{1}{1 + \mu_{n-i+1}(\boldsymbol{H}_{T,S})} \tag{232}$$

$$= \min\{\mu, 1\}(1 \pm c_{T,S}n^{-\kappa_{11}}). \tag{233}$$

Here, the positive constant $c_{T,S}$ is picked appropriately and $\kappa_{11} = \min\{\kappa_{10}, |1-q-r|\} > 0$. This completes the proof. $\qquad\square$

# I  Miscellaneous lemmas

**Lemma I.1** (Coupling of quadratic forms). *Let $\boldsymbol{B} \in \mathbb{R}^{n \times m}$ be an arbitrary real matrix and $\boldsymbol{M} \in \mathbb{R}^{n \times n}$ be a PSD matrix. Then for any vector $\boldsymbol{x} \in \mathbb{R}^m$, we have*

$$\lambda_n(\boldsymbol{M})\boldsymbol{x}^\top \boldsymbol{B}^\top \boldsymbol{B} \boldsymbol{x} \leq \boldsymbol{x}^\top \boldsymbol{B}^\top \boldsymbol{M} \boldsymbol{B} \boldsymbol{x} \leq \lambda_1(\boldsymbol{M})\boldsymbol{x}^\top \boldsymbol{B}^\top \boldsymbol{B} \boldsymbol{x}. \tag{234}$$

*Proof.* For any PSD matrix $\boldsymbol{C}$, the matrix $\boldsymbol{B}^\top \boldsymbol{C} \boldsymbol{B}$ is PSD. In particular, $\boldsymbol{C}$ has a unique square root $\boldsymbol{C}^{1/2} \in \mathbb{R}^{n \times n}$ with $\boldsymbol{C}^{1/2}\boldsymbol{C}^{1/2} = (\boldsymbol{C}^{1/2})^\top \boldsymbol{C}^{1/2} = \boldsymbol{C}$. We thus have

$$\boldsymbol{x}^\top \boldsymbol{B}^\top \boldsymbol{C} \boldsymbol{B} \boldsymbol{x} = \boldsymbol{x}^\top \boldsymbol{B}^\top \boldsymbol{C}^{1/2}\boldsymbol{C}^{1/2}\boldsymbol{B} \boldsymbol{x} \tag{235}$$

$$= \left\|\boldsymbol{C}^{1/2}\boldsymbol{B}\boldsymbol{x}\right\|_2^2 \geq 0. \tag{236}$$

Hence

$$\lambda_1(\boldsymbol{M})\boldsymbol{x}^\top\boldsymbol{B}^\top\boldsymbol{B}\boldsymbol{x} - \boldsymbol{x}^\top\boldsymbol{B}^\top\boldsymbol{M}\boldsymbol{B}\boldsymbol{x} = \boldsymbol{x}^\top\boldsymbol{B}^\top(\lambda_1(\boldsymbol{M})\boldsymbol{I}_n - \boldsymbol{M})\boldsymbol{B}\boldsymbol{x}. \tag{237}$$

Since $\boldsymbol{M} \preceq \lambda_1(\boldsymbol{M})\boldsymbol{I}_n$ by definition, $\lambda_1(\boldsymbol{M})\boldsymbol{I}_n - \boldsymbol{M}$ is a PSD matrix. Hence by applying Eq. (236), we conclude that

$$\lambda_1(\boldsymbol{M})\boldsymbol{x}^\top\boldsymbol{B}^\top\boldsymbol{B}\boldsymbol{x} - \boldsymbol{x}^\top\boldsymbol{B}^\top\boldsymbol{M}\boldsymbol{B}\boldsymbol{x} \geq 0, \tag{238}$$

which gives the upper bound in Eq. (234).

Similarly, $\boldsymbol{M} \succeq \lambda_n(\boldsymbol{M})\boldsymbol{I}_n$, so an analogous argument

$$\lambda_n(\boldsymbol{M})\boldsymbol{x}^\top\boldsymbol{B}^\top\boldsymbol{B}\boldsymbol{x} - \boldsymbol{x}^\top\boldsymbol{B}^\top\boldsymbol{M}\boldsymbol{B}\boldsymbol{x} \leq 0, \tag{239}$$

which gives the lower bound in Eq. (234). $\square$

Next, we prove the elementary anti-concentration result that we will need.

**Proposition I.2** (Gaussian anticoncentration). *Let $\boldsymbol{x} \sim N(0, \boldsymbol{I}_d)$ be a standard Gaussian vector, and let $\boldsymbol{v} \in \mathbb{R}^d$ be arbitrary deterministic vector. Then*

$$\Pr[|\langle\boldsymbol{x}, \boldsymbol{v}\rangle| \leq \epsilon] \leq \frac{2\epsilon}{\sqrt{2\pi}\|v\|_2}.$$

*Proof.* Note that $\langle\boldsymbol{x}, \boldsymbol{v}\rangle$ is a linear projection of a standard multivariate Gaussian, so it is itself a one-dimensional Gaussian. It is also clearly zero mean, and its variance is just give by the squared norm of $\boldsymbol{v}$. So $\langle\boldsymbol{x}, \boldsymbol{v}\rangle \sim N(0, \|\boldsymbol{v}\|_2^2)$. Now we have

$$\Pr[|\langle\boldsymbol{x}, \boldsymbol{v}\rangle| \leq \epsilon] = \frac{1}{\sqrt{2\pi}\|\boldsymbol{v}\|_2}\int_{-\epsilon}^{\epsilon}\exp\left(-\frac{x^2}{\|\boldsymbol{v}\|_2^2}\right)dx \leq \frac{2\epsilon}{\sqrt{2\pi}\|\boldsymbol{v}\|_2}.$$

$\square$

# J   Comparison to the straightforward non-interpolative scheme

In this section, we quickly give calculations for how well a straightforward non-interpolating scheme for learning classifiers can work asymptotically. However, a similar analysis using the tools developed to prove our main results should give a rigorous proof of the below derivation.

This scheme simply uses the sum/average of all positive training examples of a class as the vector we take an inner-product with to generate scores for classifying test points. For $m \in [k]$, define

$$\widehat{\boldsymbol{f}}_m = \sum_{i:\ell_i=m} \boldsymbol{x}_i^w. \tag{240}$$

To understand how well this will do asymptotically, it is easy to see that the for the true label-defining direction, the positive exemplars in the bi-level model will be tightly concentrating around $\sqrt{2\log k}\sqrt{\lambda_F}$ which, keeping only the polynomial-order scaling, will be like $n^{\frac{p-q-r}{2}}$. There will be roughly $\frac{n}{k} = n^{1-t}$ positive examples for every class with high probability. For simplicity, let us just look at $m = 1$ and consider $\frac{k}{n}\widehat{\boldsymbol{f}}_1 = n^{t-1}\widehat{\boldsymbol{f}}_1$. We see

$$n^{t-1}\widehat{\boldsymbol{f}}_1[1] = \Theta(n^{\frac{p-q-r}{2}}). \tag{241}$$

For the other directions that are not true-label defining, we will just have random Gaussians. The favored directions will be Gaussian with variance $\lambda_F = n^{p-q-r}$ while the unfavored directions will essentially be Gaussian with unit variance. By averaging over $n^{1-t}$ examples, those variances will be reduced by that factor. This means that for the $s = n^r$ favored directions, the variance of the average will be $n^{p-q-r-(1-t)}$ each and for the essentially $n^p$ unfavored directions, the variance of the average will be $n^{t-1}$ each.

On a test point, we are going to take the inner product of $n^{t-1}\widehat{\boldsymbol{f}}_m$ with an independent random draw of $\boldsymbol{x}_{\text{test}}^w$. For classification to succeed, we need this inner product to be dominated by the true

$m$-th feature-defining direction. When that happens, the correct label will win the comparison. One can easily see that the contribution from the true feature-defining direction will be a Gaussian with mean 0 and variance $\lambda_F \cdot (n^{\frac{p-q-r}{2}})^2 = \lambda_F^2 = n^{2p-2q-2r}$. Meanwhile, the $s$ favored features will have their scaled variances sum up in the score to give a total variance of $n^r \cdot \lambda_F \cdot n^{p-q-r-(1-t)} = n^{2p-2q-r-(1-t)}$. And finally, the unfavored features will also have their variances sum up in the score to give a total variance of $n^p \cdot 1 \cdot n^{t-1} = n^{p+t-1}$.

For the true-feature-defining direction to dominate the contamination from other favored directions, we need

$$2p - 2q - 2r > 2p - 2q - r - (1 - t) \tag{242}$$

which immediately gives the condition $t < 1 - r$.

For the true-feature-defining direction to dominate the contamination from other unfavored directions, we need

$$2p - 2q - 2r > p + t - 1 \tag{243}$$

which gives the condition $t < p + 1 - 2(q + r)$.

Here, there is no difference between regimes in which regression works or does not work. The condition for classification to asymptotically succeed is $t < \min(1 - r, p + 1 - 2(q + r))$.

Notice that when MNI regression does not work $q + r > 1$, this is identical to the tight characterization given for MNI classification in (13). But in the regime where MNI regression *does* work $q + r < 1$, this is different. For MNI classification, (13) tells us that we require $t < \min(1 - r, p - 1)$. Consider $q = 0.1, r = 0.5$ and $p = 1.1$. MNI classification can only allow $t < 0.1$. Meanwhile, the non-interpolating average-of-positive-examples classifier will work as long as $t < 0.5$. This demonstrates the potential for significant suboptimality (in terms of the number of distinct classes that can be learned) of MNI classifiers in this regime of benign overfitting for regression.

## K   Alternative framing of bi-level ensemble

In this section we provide an alternative exposition for the bi-level ensemble, which may be more intuitive for some readers.

The high level conceptual picture of the setup is as follows. The learner simply observes high dimensional jointly Gaussian zero-mean features with some unknown covariance $\Sigma \in \mathbb{R}^{d \times d}$. It then performs min-norm interpolation of (essentially) one-hot-encoded labels to learn the score functions for the $k$ different classes. These score functions are used at test-time to do multiclass classification.

Note that the learning algorithm has no knowledge of $\Sigma$, nor does it even know that the features are jointly Gaussian — all it has is the training data. For analysis purposes, we parameterize $\Sigma$. In the spirit of spiked covariance models, where a low-dimensional subspace has higher variance, we study the case that the eigenvalues of $\Sigma$ follow the simplified bi-level model parameterized by $(p, q, r)$.

The bi-level model stipulates that (i) the number of features is $d = n^p$ and (ii) there are two discrete variance levels for the Gaussian features. The higher variance component resides in a low-dimensional subspace of dimension $s = n^r$. For this bi-level model, we are able to prove sharp phase-transition style results telling us when successful generalization will happen. Here, the number of classes also matters, and that is where the parameter $t$ enters.

To simplify notation for analysis in the paper, Subramanian et al. (2022) just assume that $\Sigma$ is diagonal to begin with, instead of explicitly rotating coordinates to the eigenbasis of a general $\Sigma$. Because Gaussianity is preserved under rotations and min-norm interpolation only cares about norms, this transformation is without loss of generality. We reiterate that the learner is agnostic to all of these choices and does not know $\Sigma$ at any point.

