# OpenReview forum: "Precise asymptotic generalization for multiclass classification with overparameterized linear models"
_NeurIPS.cc/2023/Conference — NeurIPS 2023 spotlight_

### Official Review · Reviewer_qo1L · 2023-07-04

**Soundness:** 3 good
**Presentation:** 3 good
**Contribution:** 4 excellent
**Rating:** 7
**Confidence:** 1

**Summary:**

The paper studies the asymptotic generalization error behavior of an overparameterized linear model and under the Gaussian covariates bi-level model. In this setup, the number of data points, features, and classes all grow together. The authors manage to fully characterize the regimes of the generalization error, which is surprisingly “polarized”. This solves the conjecture posed by Subramanian et al. (2022).

**Strengths:**

Both the achieved theoretical result and also the used technical tools (like the newly established Hanson-Wright inequality) are highly novel and strong. Even though I am not familiar with the literature, I estimate that the work is quite valuable.

**Weaknesses:**


- The considered assumptions for the distribution of features are very simplistic. Both independence and having identical Gaussian distributions are very restrictive; which highly influences the practicality of the results.

-  The paper studies the generalization error of linear models; which are far from the deep neural networks. In this sense, there is still a big gap in the practical aspects of the paper (and also the previous literature on this). However, it is totally understandable that these are the first steps toward that goal.

- The paper lacks providing the needed intuitions about the obtained results.

**Questions:**


-  It would be useful to provide more intuition about the considered “bi-level ensemble” model. In this model, it seems that while we are considering $n^p$ dimensions, the ``effective dimensions’’ is the favored ones; meaning that the features in the rest of the dimensions (and their relative magnitude) somehow either add a “useful noise injection” or become dominant with respect to the “signal” which makes the prediction impossible. This might be inexact, but this is just to give an example of what kind of intuition I refer to.

-  Similarly, it would be extremely useful to discuss the different regimes of the theorems. Why having $r$ close to 1 would fail the learner? Why having a small $p-(q+r)$ would do so?

-   Finally, the scope of the paper remains a bit narrow (and purely technical) and the authors could probably use the intuitions derived from their results to discuss some potential understanding that these results could give about some more realistic setups. This is shortly done in the discussion section, but it would be appreciated to be extended.



**Limitations:**

As discussed above, and I guess the authors would also agree, the considered model is quite restrictive.

---

> ### Author Rebuttal · Authors · 2023-08-09
>
> We thank the reviewer for their positive review and detailed feedback.
> > The considered assumptions for the distribution of features are very simplistic. Both independence and having identical Gaussian distributions are very restrictive; which highly influences the practicality of the results.
> - Because our paper focuses on resolving the conjecture of Subramanian et al., we chose to adopt the same model and notation for ease of comparison.
> - With respect to independence, we agree that it is a strong simplifying assumption. However, it is not essential for every part of the argument. In several places, we use union bound, so independence is not really needed. Furthermore, in the Gaussian world, independence is assumed mainly just for exposition. As long as we assume the labels are being determined by a subspace of the features, we can do a basis change for the sake of analysis to make the covariance diagonal (and the learning algorithm need not know about this basis change).
> - Generally speaking, we made these simplifying assumptions for the sake of tractability. Of course, there is a need to move beyond flat covariances, but it requires more work. As we explain in the response to reviewer V1nE, we think that many of the same results hold if we allow the covariance to be heterogeneous along the non-label defining directions, but the situation likely becomes much more complex if the label-defining directions are allowed to be heterogeneous.
> - We think that going from Gaussian to subgaussian distributions will be fine if we use an appropriate vector notion of subgaussianity that allows us to perform basis changes. One place where the Gaussianity assumption is used explicitly is the margin computation, where we needed to use the fact that the gap between the max and second max decays at a logarithmic rate. It is certainly of interest to precisely identify where else we actually need the Gaussian assumption and where we can relax it.
>
> > The paper studies the generalization error of linear models; which are far from the deep neural networks. In this sense, there is still a big gap in the practical aspects of the paper (and also the previous literature on this). However, it is totally understandable that these are the first steps toward that goal.
> - We agree that like many other papers in the field, our work is just one of the first steps towards a more robust theory for deep networks. However, in finetuning settings where a nonlinear network is massively overparameterized with respect to the finetuning data, one can use the NTK approximation to study the finetuning behavior with overparameterized generalized linear models. Here, the pretraining of the model creates an implicit nonlinear lifting of the inputs into a high dimensional space, where every parameter of the model corresponds to one nonlinear feature in the lifting (via the first-order Taylor expansion of the model with respect to the finetunable parameters). It is an interesting question to determine whether real systems actually have the type of behavior predicted by our results. (In fact, the nature of our results point to the need for work in this direction because simply looking at the empirical eigenvalues of the training data might not reveal the existence of the underlying spiked structure that helps generalization work.)
>
> > The paper lacks providing the needed intuitions about the obtained results.
> - We will try to add some more intuition along the lines of your suggestion below.
>
> > It would be useful to provide more intuition about the considered “bi-level ensemble” model. In this model, it seems that while we are considering dimensions, the "effective dimensions" is the favored ones; meaning that the features in the rest of the dimensions (and their relative magnitude) somehow either add a “useful noise injection” or become dominant with respect to the “signal” which makes the prediction impossible. This might be inexact, but this is just to give an example of what kind of intuition I refer to.
> - Thanks for the suggestion about discussing the theorem. Your intuitions about the bi-level model are essentially correct. Below we answer your questions more directly.
> - The intuition for $ r \approx 1 $ being a failure mode is that the model is effectively severely overparameterized, even after restricting to the favored features. In particular, since there are $n^t$ classes, the failure condition $t + r > 1$ precisely captures the right sense of bad overparameterization restricted to the subspace of $n^r$ favored features.
> - The other condition about $ p - (q+r) $  is about the feature weighting for favored features: if the favoring is too small, classification will not succeed because the noise/contamination from the unfavored features will dominate the signal. We will include this discussion in the revision.
> - The paper's discussion of the "just average positive exemplars" (non-interpolating) classifier should also help give a complementary intuition for the failure conditions.
>
> > Finally, the scope of the paper remains a bit narrow (and purely technical) and the authors could probably use the intuitions derived from their results to discuss some potential understanding that these results could give about some more realistic setups. This is shortly done in the discussion section, but it would be appreciated to be extended.
> - We agree that the main contribution of our paper is technical. However, one consequence of the technical work is that it justifies the style of heuristic calculation that led to the conjecture in the first place (carried out in the appendix of Subramanian et al.). We believe that a similar type of heuristic reasoning can be more widely applied in more realistic settings, with tools such as our sharpened Hanson-Wright inequality potentially being useful to justify those calculations.

---

> > ### Comment · Area_Chair_ZMoe · 2023-08-21
> >
> > To the authors: your response has been read and is being considered.

---

### Official Review · Reviewer_V1nE · 2023-07-06

**Soundness:** 3 good
**Presentation:** 3 good
**Contribution:** 3 good
**Rating:** 8
**Confidence:** 3

**Summary:**

The paper titled "Asymptotic Generalization of Overparameterized Linear Models for Multiclass Classification under Gaussian Covariates Bi-level Model" presents a study on the asymptotic generalization of overparameterized linear models for multiclass classification under the Gaussian covariates bi-level model. The authors provide an asymptotic characterization of the generalization of a linear model for multiclass classification in an idealized Gaussian setting where a) the number of data points, b) the dimension, and c) the number of classes diverge while their ratio remains fixed. An interesting result, in particular, is that the min-norm interpolating classifier can be suboptimal in this regime.

**Strengths:**

The paper present many strong analytical results.

In particular, the authors have successfully resolved a conjecture posed in a previous works  (Subramanian et al. '22,) and established new lower bounds that demonstrate the misclassification rate either goes to 0 or 1 asymptotically.  The paper also introduces a new variant of the Hanson-Wright inequality, a tool used in high-dimensional probability, that is particularly useful for multiclass problems with sparse labels. Sparse labels refer to situations where only a small number of classes are represented in the dataset.


**Weaknesses:**



**Questions:**

How realistic is the  Bi-level feature weighting model ? I am thinking instead of the source/capacity setting where I would expect a more power-law-like behavior. How different would the conclusions be?

**Limitations:**

The paper is theoretical in nature and its limitations are stated in the theorems

---

> ### Author Rebuttal · Authors · 2023-08-09
>
> We thank the reviewer for their positive review and feedback.
> > How realistic is the Bi-level feature weighting model ? I am thinking instead of the source/capacity setting where I would expect a more power-law-like behavior. How different would the conclusions be?
>
> - Because our paper focuses on resolving the conjecture of Subramanian et al., we chose to adopt the same model and notation for ease of comparison.
> - You are correct that in applications for spiked covariance models, one typically sees power-law like behavior. We expect that the bi-level model can be relaxed to allow for constant deviations in the weightings for the $s-k$ favored features that are not label defining, and a power law decay for the $d-s$ unfavored features. The former change would likely only affect constants in certain areas of the argument that do not crucially depend on the exact constants involved, whereas the latter would likely just change the effective degree of overparameterization (à la Bartlett et al.). In this type of more relaxed bi-level setup, we expect an analogous result to hold, but ironing out these details is definitely an important next step.
> - However, even allowing for constant fluctuations in weighting for the label-defining features can lead to many subtleties. Even constant fluctuations in label defining weightings manifest as polynomial variations in how many examples of each class there are. Hence the heterogeneity between label-defining directions would likely lead to much messier condition for generalization.
> - On that note, if one is able to analyze the heterogeneity in label-defining features, one would naturally also capture a setup with imbalanced classes, which is more representative of real-life applications. We leave this as an interesting direction for future work.

---

### Official Review · Reviewer_S1af · 2023-07-09

**Soundness:** 4 excellent
**Presentation:** 4 excellent
**Contribution:** 4 excellent
**Rating:** 8
**Confidence:** 1

**Summary:**

This is a theoretical paper that gives insight into how and when an overparametrized linear classification model, for multi-class classification, can be successfully generalized.    In particular, they look at multiclass classification under the Gaussian covariates bi-level model introduced by Subramamian et al. in 2022, and fully resolve a conjecture from that paper.  The key to their analysis is a new variant of the Hanson-Wright inequality.

**Strengths:**

This paper builds on previous work to give tight bounds on the regions where generalization is possible, improving significantly on previous partial results.  It also makes rigorous previous analyses based on heuristic calculations.
This is a very well-written paper.
Based on the proof sketch given in the main paper, the technical level of the proofs seems high, involving both the use of existing techniques as well as the proof of a new version of the Hanson-Wright inequality,.

**Weaknesses:**

No significant weaknesses were noted.

**Questions:**


None.

**Limitations:**


It would be helpful to comment on the limitations of the bi-level model studied.

---

> ### Author Rebuttal · Authors · 2023-08-09
>
> We thank the reviewer for their positive review.
>
> > It would be helpful to comment on the limitations of the bi-level model studied.
> - As mentioned in the general comment, because the main contribution of our paper is resolving the conjecture of Subramanian et al., we chose to adopt the same model and notation for ease of comparison.
> - Certainly, the bi-level model, being a simplified caricature model, has its own limitations which make it a little unrealistic. For example, the bi-level structure implies that the training dataset has balanced classes, which is often unrealistic in real world scenarios with lots of classes. Relaxing some of these assumptions is an interesting direction for future work. See also our rebuttal to Reviewer V1nE for a more in depth discussion of how one might be able to relax the assumptions of the bi-level model.

---

> > ### Comment · Area_Chair_ZMoe · 2023-08-21
> >
> > To the authors: your response has been read and is being considered.

---

### Official Review · Reviewer_ggP8 · 2023-07-21

**Soundness:** 4 excellent
**Presentation:** 4 excellent
**Contribution:** 4 excellent
**Rating:** 8
**Confidence:** 4

**Summary:**

In their main result, Theorem 3.2, the authors establish Conjecture 3.1 which is a conjecture posed in 2022 describing the asymptotic misclassification probability of the bi-level ensemble model (Definition 1) under a sparsity assumption (Assumption 1).  They provide a rigorous and tight analysis, with very clear explanations both in-text and in the appendix.

**Strengths:**

The paper is clearly written and explained wonderfully, the proofs are detailed, and the contributions are interesting.  The proofs are sound and clearly articulated.


Minor point, the appendix is also very neatly organized which makes proofreading nice.

**Update:** My questions and concerns have been addressed.

**Weaknesses:**

Overall the paper is rigerous and clear, so I do not really have any significant weakness worth reporting; only a few minor comments.

* Only minor comments:*

 - In equations (36)-(40) could you explicitly write the polylogarithmic terms in the denominator, or at the very least, precisely defined them after the equation environments.

- Could you formally state that all r.v. are defined on the same probability space, at the beginning of the paper; for rigor.

- Very minor point, above equation (219), I guess $\boldsymbol{A}_{-\boldsymbol{S}}^{-1}$ is *block diagonal* not diagonal.

**Questions:**

There are two little details, which I didn't fully follow, so let me ask:

- Perhaps a silly question, but in equation (182), are you using a Brascamp–Lieb inequality (or something simpler which I'm possibly missing)?

- In equation (240) do you mean $n^{t-1}\hat{\boldsymbol{f}}_1[1]$ is $\Theta(n^{(p-q-2)/2})$?

---

> ### Author Rebuttal · Authors · 2023-08-09
>
> We thank the reviewer for their positive feedback and suggestions.
>
> > In equations (36)-(40) could you explicitly write the polylogarithmic terms in the denominator, or at the very least, precisely defined them after the equation environments.
> - Yes, we will include the explicit polylog factors. For clarity, all of these terms are at most $\log(nsk)$.
>
> > Could you formally state that all r.v. are defined on the same probability space, at the beginning of the paper; for rigor.
> - We will add this to the paper as requested. The probability space for training can be viewed as being $n \times d$ iid standard Gaussians, and extended for each test point by another $d$ iid standard Gaussians. All random variables for us are either linear combinations of the underlying iid Gaussians or functions of those random variables. There is no other randomness in the setup.
>
> > Very minor point, above equation (219), I guess $\boldsymbol{A}\_{-S}^{-1}$ is block diagonal not diagonal.
>
> - We see how there might be some confusion with the wording of the argument here. We will explicitly add the definition of the matrices into the notation table. We will also replace the text leading up to (219) with the following:
>
> Since $\boldsymbol{A}\_{-S}^{-1}$ is symmetric, it has an eigendecomposition $\boldsymbol{V} \boldsymbol{D} \boldsymbol{V}^\top$, where $\boldsymbol{V}$ is an orthogonal matrix. Because $\boldsymbol{W}\_T$ is a weighted subset of only the (equally) favored features, its law is rotationally invariant. Furthermore, since $\boldsymbol{A}\_{-S}^{-1}$ is independent of $\boldsymbol{W}\_T$ (as $T \subseteq S$), we can absorb the rotation $\boldsymbol{V}$ into $\boldsymbol{W}\_T$ to reduce to the case where $\boldsymbol{A}\_{-S}^{-1} = \boldsymbol{D}$..
>
> > Perhaps a silly question, but in equation (182), are you using a Brascamp–Lieb inequality (or something simpler which I'm possibly missing)?
> - There is a typo here (thank you so much for pointing it out); there should be an expectation on the LHS: it should be $$\mathbb{E} \exp(\lambda S_{\mathrm{diag}}) = \prod_{i=1}^n \mathbb{E}\_{X\_i, Y\_i} \exp(\lambda m\_{ii} (X\_iY\_i - \mathbb{E}[X\_iY\_i])),$$ which is just using independence of the $(X\_i, Y\_i)$ pairs. After that, we apply Jensen’s inequality to get to the next line.
>
> > In equation (240) do you mean $n^{t-1}\hat{\boldsymbol{f}}_1[1]$ is $\Theta(n^{(p-q-2)/2})$?
> - Yes, that is correct, we will fix the notation in the revision.

---

> > ### Comment · Reviewer_ggP8 · 2023-08-16
> > **Happy with edits**
> >
> > Dear authors,
> >
> >
> > Thanks you very much for the clear response and very nice paper.
> >
> > Goodluck :)

---

### Official Review · Reviewer_PSb6 · 2023-07-27

**Soundness:** 3 good
**Presentation:** 3 good
**Contribution:** 2 fair
**Rating:** 7
**Confidence:** 2

**Summary:**

In this paper, the authors analyze the generalization of the linear multiclass classifiers in the overparametrized regime under the bi-level model with Gaussian covariates. In particular they prove a conjecture made in a previous paper about the region (characterized by the paramters of the bi-level model) under which the generalization error can go to zero. In fact, they establish a `0-1` law for generalization error: depending upon the parameter regime the probability of error either converges to zero or one. In the process of proving  this result the authors also establish a generalization of the usual Hanson Wight inequality for quadratic forms to exploit "soft sparsity".

Overall, I feel that this paper contains a technically rigorous analysis of the overparametrized classification problem somewhat restrictive model assumptions.

**Strengths:**

 The paper is well written, and while the material is quite technical, the authors do make an effort to provide sufficient context and explanations to motivate them. In particular, the detailed overview of the argument used in proving the main result (Theorem 3.2) is quite helpful in getting the idea used in the proof.

**Weaknesses:**

1.  While this paper does not seem to have any obvious flaws, I feel that the model analyzed in this paper is a bit too restrictive. In a footnote on page 4, the authors mention that "such models are widely used to study learning even beyond this particular thread of work". It would be very helpful, if the authors include a detailed discussion of at least one such practical application, which is naturally modeled by the bi-level model studied in this paper.
2. Besides the model, even the construction of the classifier makes some strong assumptions. In particular, the classifier is constructed after reweighting the features with weights $(\lambda_i)_{i \geq 1}$ that  are tuned to the specific bi-level model.  Since in most practical problems, there is at least some level of misspecification, I feel that this reduces the practical utility of the results in this paper.

**Questions:**

1. Can you include a discussion of some practical tasks where the bi-level model arises naturally?

2. Regarding the second point in the weaknesses section, can you discuss (at least informally) the effect of model misspecification on the generalization guarantees. More generally, do you think that the same performance guarantees as Theorem 3.2 (i.e., probability of error converging to zero for the same parameter ranges) can be achieved adaptively without using the knowledge of model parameters ($p, q, r, t$) to construct the classifier? Or there is a price to pay for achieving zero generalization adaptively?

**Limitations:**

I think that the strong model assumptions used in this paper might reduce the practical utility of the results of this paper.

---

> ### Author Rebuttal · Authors · 2023-08-09
>
> We thank the reviewer for their comments and suggestions.
> > While this paper does not seem to have any obvious flaws, I feel that the model analyzed in this paper is a bit too restrictive. In a footnote on page 4, the authors mention that "such models are widely used to study learning even beyond this particular thread of work". It would be very helpful, if the authors include a detailed discussion of at least one such practical application, which is naturally modeled by the bi-level model studied in this paper.
> - Please see the general comment for an overview and motivation of the bi-level model. To repeat the key takeaway here, the bi-level model is a stylized version of the spiked covariance model.
> - Spiked covariance models have been extensively studied in the statistics literature (see e.g. [1], [2]), and are a common assumption for many statistical applications. Some concrete theoretical models related to the spiked covariance model include the sparse/nonnegative/tensor PCA problems and stochastic block model. For real life applications, some examples (lifted from [1]) include climate studies and functional data analysis.
>
> >Besides the model, even the construction of the classifier makes some strong assumptions. In particular, the classifier is constructed after reweighting the features with weights $(\lambda\_i)\_{i \ge 1}$ that are tuned to the specific bi-level model. Since in most practical problems, there is at least some level of misspecification, I feel that this reduces the practical utility of the results in this paper.
>
> - As mentioned above, we tried to stay as consistent as possible with the notation and setup of Subramanian et al., who used this explicit reweighting procedure in the exposition of the setup so that their analysis would be notationally simpler. We would like to emphasize that this explicit reweighting is done purely for analysis purposes, and in reality the actual classifier does *not* do any reweighting of features. The classifier observes jointly Gaussian features with labels during training and does min-norm interpolation to learn the score functions. At test time, the test vector is fed into the $k$ score functions to produce the labeling by looking for the highest score. The classifier has no dependence on the $\lambda_i$, or $p, q, r, t$. It just sees the training data.
> - In the real world, misspecification is absolutely a real concern, so the reviewer’s concern about the role of misspecification is certainly warranted. So, an interesting question is to what extent is this classifier robust to distribution shift at test time? Roughly speaking, our intuition is that with small enough (non-adversarial) shifts, the margin that the classifier currently leverages to succeed would allow the classifier to still generalize.  This is because our analysis is largely tied to the training distribution (establishing appropriate concentration, etc.).  Quantifying this robustness is an interesting future direction and is in fact related to the fact that the min-norm interpolator generalizes for multi-label classification as well. We did not discuss the connection to misspecification in the paper at all, but if you think it is important please let us know and we will be sure to mention it in the revision.
>
> > Can you include a discussion of some practical tasks where the bi-level model arises naturally?
> - See the above discussion about spiked covariance models, and we will add some into the revision as well. Certainly, the bi-level model, being a simplified caricature model, has its own limitations which make it a little unrealistic. For example, the bi-level structure implies that the training dataset has balanced classes, which is often unrealistic in real world scenarios with lots of classes. Relaxing some of these assumptions is an interesting direction for future work.
> - The bi-level model is a tractable linear proxy for the more widely accepted manifold hypothesis about real-world high-dimensional data.
>
> > Regarding the second point in the weaknesses section, can you discuss (at least informally) the effect of model misspecification on the generalization guarantees. More generally, do you think that the same performance guarantees as Theorem 3.2 (i.e., probability of error converging to zero for the same parameter ranges) can be achieved adaptively without using the knowledge of model parameters $(p, q, r, t)$ to construct the classifier? Or there is a price to pay for achieving zero generalization adaptively?
>
> - The model parameters $(p, q, r, t)$ are only used to parameterize the data generating assumptions (the bi-level model), and the min-norm-trained classifier itself does not need to know any of these parameters to learn the score functions. Hence, the classifier defined in the paper will achieve the stated performance guarantees.
> - Of course, it receives $d = n^p$ dimensional features and knows that there are $k = n^t$ classes, but it is hard to think of a situation where a learning algorithm would not know these ahead of time. On the other hand, the classifier definitely does not need to know $q$ or $r$. One of the interesting results of our paper is that multiclass classification can succeed even in the regime $q+r>1$, where the empirical covariance is completely flat, despite the spike being present in the true covariance. Put differently, although the data approximately lies on a low dimensional manifold, you can’t see it from the empirical covariance given the limited amount of data available, so that it is naively impossible to recover $q$ and $r$.
>
> [1] Johnstone, I. M. (2001). On the distribution of the largest eigenvalue in principal components analysis. Annals of Statistics, 29(2), 295-327.
>
> [2] Donoho, David L., Matan Gavish, and Iain M. Johnstone. "Optimal shrinkage of eigenvalues in the spiked covariance model." Annals of Statistics 46.4 (2018): 1742.

---

> ### Comment · Reviewer_PSb6 · 2023-08-18
> **Reply to the rebuttal**
>
> Thank you for the clarifications about the bi-level model and model-misspecification. I am happy to update my score to 7.

---

### Official Review · Reviewer_BQHA · 2023-07-27

**Soundness:** 3 good
**Presentation:** 3 good
**Contribution:** 3 good
**Rating:** 6
**Confidence:** 1

**Summary:**

By resolving the conjecture posed by Subramanian et al. (2022), the authors address the asymptotic generalization for overparameterized minimum-norm interpolation (MNI) linear multi-class classifiers under two assumptions: (1) features are Gaussian vectors and labels are generated from $1$-sparse noiseless model; (2) the scaling follows bi-level ensemble. This paper can be seen as a completion to the previous result of Subramanian et al. (2022) since it captures the generalization for both the regime where the regressor fails and regime where it works. The technical contribution appears to be a new variant of the Hanson-Wright inequality.

**Strengths:**

A sound completion to the previous theoretical result.


**Weaknesses:**

The setting of bi-level ensemble seems a bit weird to me, especially the eq (12). Maybe the authors can offer more explainations.

**Questions:**

Line 165: "it suffices for the maximum value of the LHS ......'' Is it the maximum instead of minimum here?

**Limitations:**

The authors adequately addressed the limitations.

---

> ### Author Rebuttal · Authors · 2023-08-09
>
> We thank the reviewer for their comments and suggestions.
>
> > The setting of bi-level ensemble seems a bit weird to me, especially the eq (12). Maybe the authors can offer more explainations.
>
> - See the general comment for an overview of the bi-level model. The bi-level model is a set of simplifying assumptions to more general spiked covariance models which distills out a minimal set of assumptions for theoretical tractability while being able to call out interesting phenomena.
> - The choice of the feature weights in eq (12) reflects a relatively natural normalization of the covariance of the feature vector so that $\mathrm{tr}(\Sigma) = d = n^p$. For example, this kind of trace would arise if each feature on its own had unit variance but there was some interesting correlation structure among the features. (Of course, we do all our analysis in the underlying rotated eigenbasis coordinates for $\Sigma$ for notational simplicity.)
>
> > Line 165: "it suffices for the maximum value of the LHS ......'' Is it the maximum instead of minimum here?
> - This is indeed a typo and we have fixed this in the revised version.

---

> > ### Comment · Reviewer_BQHA · 2023-08-16
> >
> > Dear  authors,
> >
> > Thank you for the clarifications on eq(12) and it makes more sense to me now.

---

### Author Rebuttal · Authors · 2023-08-09

We thank all of the reviewers for their comments and feedback. Below, we highlight some high level takeaways which address a set of questions shared across multiple reviewers.
## Bi-level model
- Because the main contribution of our paper is fully resolving the conjecture of Subramanian et al. from NeurIPS 2022, we chose to adopt the same model and notation in the paper for ease of comparison.
-  The high level conceptual picture of the setup is as follows: The learner simply observes jointly Gaussian zero-mean features with some **unknown** covariance structure $\Sigma$ in a very high dimensional space, and performs min-norm interpolation of essentially one-hot-encoded labels to learn the score functions. These score functions are used at test-time to do multi-class classification.
- Note that the learning algorithm has no knowledge of $\Sigma$, nor does it even know that the features are jointly Gaussian. *All the learning algorithm has is the training data.*
- For analysis purposes, $\Sigma$ is parameterized. In the spirit of spiked covariance models, where a low-dimensional subspace has higher variance, we study the case that the *eigenvalues* of $\Sigma$ follow the simplified bi-level model parameterized by $(p, q, r)$. The bi-level model stipulates that there are two discrete variance levels, the higher of which lies in the low-dimensional subspace. For this bi-level model, we are able to prove sharp phase-transition style results telling us when successful generalization will happen. Here, the number of classes also matters, and that is where the parameter $t$ enters.
- This bi-level model is a stylized linear version of the well-known manifold hypothesis, which stipulates that real-world high dimensional data actually approximately lie on a low rank manifold that is unknown to the learning algorithm.
- To simplify notation in the paper, Subramanian, et. al. just assume $\Sigma$ is diagonal to begin with instead of explicitly rotating coordinates to the eigenbasis of a general $\Sigma$. Because min-norm interpolation only cares about norms, this is without loss of generality.

The camera-ready version will make this setup clearer.

---

### Decision · Program_Chairs · 2023-09-21

**Decision:**

Accept (spotlight)

**Comment:**

Some very nice and surprising results on a topic of wide interest.

We noted that many of the reviewers had low confidence.  The authors are encouraged to write for a wider audience.